# LOCAL GEOMETRY ATTENTION FOR TIME SERIES FORECASTING UNDER REALISTIC CORRUPTIONS

**Dongbin Kim[1],   Youngjoo Park[1],   Woojin Jeong[1],   Jaewook Lee[1*]**

[1]Seoul National University

{dongbin413,youngjoo0913,jwj7955,jaewook}@snu.ac.kr

## ABSTRACT

Transformers have demonstrated strong performance in time series forecasting, yet they often fail to capture the intrinsic structure of temporal data, making them susceptible to real-world noise and anomalies. Unlike in vision or language, the local geometry of temporal patterns is a critical feature in time series forecasting, but it is frequently disrupted by corruptions. In this work, we address this gap with two key contributions. First, we propose Local Geometry Attention (LGA), a novel attention mechanism theoretically grounded in local Gaussian process theory. LGA adapts to the intrinsic data geometry by learning query-specific distance metrics, enabling it to model complex temporal dependencies and enhance resilience to noise. Second, we introduce TSRBench, the first comprehensive benchmark for evaluating forecasting robustness under realistic, statistically-grounded corruptions. Experiments on TSRBench show that LGA significantly reduces performance degradation, consistently outperforming both Transformer and linear model. These results establish a foundation for developing robust time series models that can be deployed in real-world applications where data quality is not guaranteed. Our code is available at: https://github.com/dongbeank/LGA.

## 1  INTRODUCTION

Transformer architectures have revolutionized deep learning across various domains since their introduction (Vaswani et al., 2017). Their success in natural language processing (Devlin et al., 2019) and computer vision (Dosovitskiy et al., 2020) has extended to time series analysis, where models like PatchTST (Nie et al., 2023) have set new performance benchmarks.

However, time series data have unique characteristics that challenge standard Transformers. Unlike text or images, time series often exhibit complex temporal dependencies and non-uniform local data distributions, creating a structured "attention geometry" (Si et al., 2024; Lavin & Ahmad, 2015). Standard attention mechanisms, which treat all inputs uniformly, may fail to adapt to these local statistical variations, seasonal patterns, and anomalies, leading to suboptimal performance and a lack of robustness (Schmidl et al., 2022; Cheng et al., 2024).

Furthermore, while robustness to input corruptions is a well-established evaluation standard in other fields, with benchmarks like ImageNet-C (Hendrycks & Dietterich, 2019), a comparable framework for time series forecasting is notably absent. This gap is critical, as real-world time series are frequently contaminated by issues like sensor failures and transmission noise, yet existing research has largely focused on synthetic adversarial attacks rather than realistic data degradation (Liu et al., 2023; Cheng et al., 2024). These limitations highlight a dual need: attention mechanisms that adapt to local temporal structure and principled benchmarks to assess their robustness.

To address these challenges, we make two primary contributions:

- We propose **Local Geometry Attention (LGA)**, a novel attention mechanism designed to adapt to the intrinsic data geometry of time series. Theoretically grounded in local Gaussian process theory, LGA learns query-specific distance metrics to compute attention

---

*Corresponding author

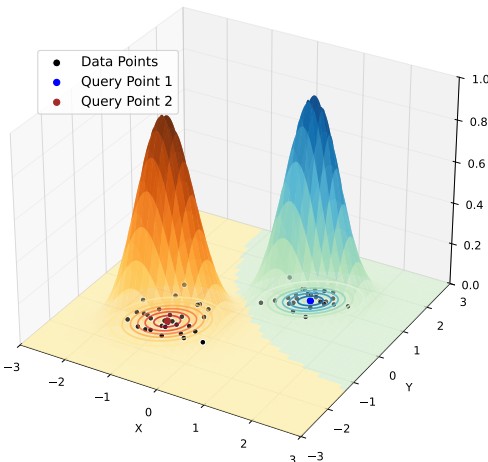 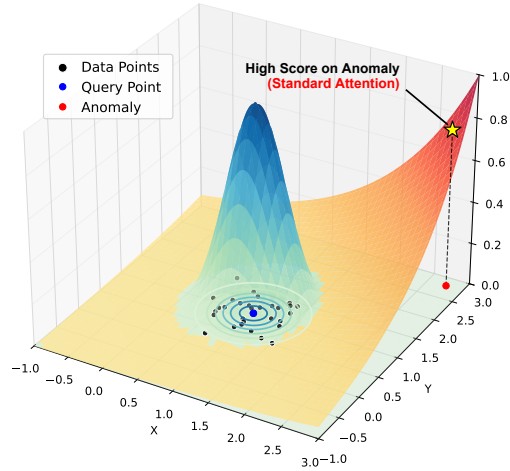

(a) Attention score surfaces of LGA for two distinct query points

(b) Attention score surfaces of LGA and Standard Attention for a query point, with an anomaly

Figure 1: The core principles of Local Geometry Attention (LGA) illustrated on a 2D toy dataset. In (a), for two different data distributions, the attention score surface generated from each query point (Query Point 1 and Query Point 2) correctly captures the unique local geometry of its respective cluster. (b) demonstrates robustness of LGA: while standard attention (red surface) is distracted by the anomaly and assigns it a high score, LGA (blue surface) correctly identifies it as an anomaly and maintains focus on the main data distribution.

    scores, enabling it to model complex local data structures without imposing strong global assumptions.

- We introduce **TSRBench**, a comprehensive benchmark for evaluating time series forecasting robustness. TSRBench provides statistically grounded methods for injecting two canonical corruption types—*spikes* and *level shifts*—with controllable severity levels, addressing a critical gap in principled robustness evaluation for time series.

- We demonstrate practical effectiveness and robustness of LGA through extensive experiments. As illustrated in Figure 1, LGA successfully identifies distinct data distributions and ignores anomalies, unlike standard attention. Our full empirical evaluation on the TSRBench benchmark further shows that LGA consistently and significantly mitigates performance degradation under various realistic corruptions.

We validate our approach through extensive experiments on standard forecasting datasets. The results demonstrate that LGA significantly mitigates performance degradation under corruption compared to existing baselines, highlighting the importance of designing both robust attention mechanisms and principled evaluation tools for time series forecasting.

## 2 RELATED WORK

**Theoretical Approaches for Attention.** Transformers (Vaswani et al., 2017) have become the dominant architecture in natural language processing (Devlin et al., 2019; Brown et al., 2020; Raffel et al., 2020) and vision (Dosovitskiy et al., 2021; Liu et al., 2021; Touvron et al., 2021), and have recently been extended to time series analysis (Wu et al., 2021; Zhang & Yan, 2023; Nie et al., 2023). Attention is a fundamental component of Transformer architectures, and recent studies have offered several theoretical interpretations of its structure. Bui et al. (2024) interpret attention as cross-covariance between correlated Gaussian processes to enable asymmetric uncertainty-aware attention. Similarly, Chen & Li (2023) proposes Sparse Gaussian process Attention (SGPA), which replaces the standard dot-product with a symmetric kernel, allowing Bayesian inference via the GP posterior. Han et al. (2023) propose robust kernel density estimation (RKDE), which mitigates the influence of outlier keys in the computation of attention scores. They employ the Median-of-Means (MoM)

principle (Jerrum et al., 1986; Humbert et al., 2022) into RKDE to further improve computational efficiency. Nielsen et al. (2024) constructs hyper-ellipsoidal neighborhoods around queries to increase attention weights in contextually important directions. However, these approaches either rely on global kernel assumptions or do not explicitly capture the local geometric structure of the data.

**Robustness benchmarks under Realistic Corruptions.** Robustness benchmarks are critical for evaluating model performance under contaminated inputs. In computer vision, ImageNet-C (Hendrycks & Dietterich, 2019) set a standard by testing models against various common corruptions at multiple severity levels. Similar efforts exist in natural language processing (McCoy et al., 2020; Nie et al., 2020). In contrast, the time series domain has largely focused on robustness against synthetic adversarial attacks (Liu et al., 2023; Lin et al., 2024), which target model-specific vulnerabilities but may not reflect naturally occurring data degradation.

Recent studies, however, emphasize the importance of evaluating models against realistic corruptions that mirror real-world phenomena (Cheng et al., 2024). Real-world time series often exhibit both minor point-wise disturbances and structured anomalies that signal meaningful, event-driven changes, such as random spikes from sensor limitations or sustained level shifts from hardware malfunctions (Si et al., 2024; Schmidl et al., 2022). Despite the need, a standardized benchmark for such realistic corruptions in time series remains a significant gap, underscoring the need for a systematic framework to evaluate model robustness under diverse operational conditions.

## 3 LOCAL GEOMETRY ATTENTION

We propose **Local Geometry Attention (LGA)**, a novel approach that adapts to the intrinsic geometric structure of time series data. Unlike standard attention mechanisms that use dot product similarity in Euclidean space, LGA computes attention scores using query-specific distance metrics derived from Gaussian process theory. This approach captures the local geometric structure of the data manifold, enabling more effective attention mechanisms that reflect the inherent geometry of periodic temporal patterns, as illustrated in Figure 2, where similar periodic patterns naturally form clusters.

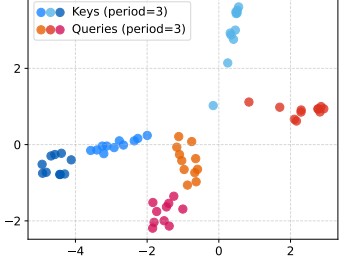

Figure 2: Key-query embeddings in PatchTST showing natural clustering of periodic patterns.

In this section, we build LGA from the ground up. We start by developing its theoretical foundation in three parts: we first establish a local kernel-covariance formulation to capture data geometry in Section 3.1; we then connect this to local Gaussian processes to estimate data density in Section 3.2; and finally, we derive the geometry-aware attention scoring function in Section 3.3. With the theory established, Section 3.4 then presents the practical implementation, detailing how this theoretically-grounded function is efficiently approximated for use in modern architectures.

### 3.1 LOCAL GEOMETRY ESTIMATION VIA KERNEL-COVARIANCE

To capture the local geometric structure of time series data, we employ local Gaussian processes instead of traditional global Gaussian processes. We first establish a local kernel-covariance formulation that naturally extends to a local Gaussian process framework, providing a principled foundation for modeling the intrinsic geometry of time series and informing our attention mechanism.

**Local Kernel–covariance.** First, we define linear mappings that project an input $x \in \mathbb{R}^n$ into key and query representations, $\mathbf{k} \in \mathbb{R}^d$ and $\mathbf{q} \in \mathbb{R}^d$, respectively. For a set of inputs $\{x_1, \ldots, x_T\}$ and a target point $x_*$, we obtain key vectors $\{\mathbf{k}_1, \ldots, \mathbf{k}_T\}$ and a query vector $\mathbf{q}_*$.

We then define a feature mapping $\phi : \mathbb{R}^d \to \mathcal{W}$ based on the difference between keys and the query: $\phi(\mathbf{k}_i) = \mathbf{k}_i - \mathbf{q}_*$. The design matrix $\mathbf{\Phi}$ is constructed as:

$$\mathbf{\Phi}(x_*) = [\phi(\mathbf{k}_1), \cdots, \phi(\mathbf{k}_T)]^\top .$$

The local kernel-covariance matrix for the target point $x_*$ is formed as a weighted sum of outer products:

$$\Sigma(x_*) = \mathbf{\Phi}(x_*)^\top \mathbf{W}(x_*)\mathbf{\Phi}(x_*) = \sum_{i=1}^{T} \omega_i(x_*)\,(\mathbf{k}_i - \mathbf{q}_*)(\mathbf{k}_i - \mathbf{q}_*)^\top. \tag{1}$$

Here, $\mathbf{W}(x_*)$ is a $T \times T$ diagonal matrix with weights $\omega_i(x_*)$ on its diagonal. These weights are computed using a kernel function $K$, which measures the similarity between each key $\mathbf{k}_i$ and the query $\mathbf{q}_*$:

$$\omega_i(x_*) = \frac{K(\mathbf{k}_i, \mathbf{q}_*)}{\sum_{j=1}^{T} K(\mathbf{k}_j, \mathbf{q}_*)}.$$

In our implementation, we use a Gaussian kernel for $K$:

$$K(\mathbf{k}_i, \mathbf{q}_*) = \exp\left(-\|\mathbf{k}_i - \mathbf{q}_*\|^2/2h^2\right),$$

where $h$ is the bandwidth parameter. More generally, $K$ can be any kernel function, such as the compactly supported tri-cube or Epanechnikov kernels. Our local kernel-covariance matrix, as defined in Equation (1), thus employs decaying kernel weights to approximate the inverse metric tensor on the data manifold. As demonstrated by Berry & Sauer (2016), this formulation effectively captures the local geometry of the underlying manifold.

## 3.2 Density Estimation with Local Gaussian Processes

Now consider a local Gaussian process regression model where all observed outputs are zero ($y_i = 0$ for all $i$), based on the local kernel covariance centered at a target point $x_*$. The model takes the form:

$$y = f(x) + \varepsilon, \qquad \text{where} \quad f(x) \sim \mathcal{GP}(0, k(x, x')) \tag{2}$$

Here, the noise term $\varepsilon$ is assumed to be i.i.d. Gaussian, $\varepsilon \sim \mathcal{N}(0, \sigma^2)$, and $k(x, x')$ is the GP covariance kernel.

For a target point $x_*$ (which maps to query $\mathbf{q}_*$), the local GP model is established. The predictive distribution for the latent function value at a new point $x$ (which maps to key $\mathbf{k}$) has a zero mean. Reusing the feature map $\phi(\mathbf{k}) = \mathbf{k} - \mathbf{q}_*$, the predictive variance is given by:

$$\sigma_{\mathbf{q}_*}^2(\mathbf{k}) = \phi(\mathbf{k})^\top G(x_*)\phi(\mathbf{k}) = (\mathbf{k} - \mathbf{q}_*)^\top G(x_*)(\mathbf{k} - \mathbf{q}_*) \tag{3}$$

where the matrix $G(x_*)$ is defined as:

$$G(x_*) = \sigma^2[\Sigma(x_*) + \sigma^2 I]^{-1}. \tag{4}$$

Crucially, the predictive variance is smaller in regions dense with data points (keys) and larger in sparse regions (Williams & Rasmussen, 2006; Kim & Lee, 2007). This means the negative predictive variance, $-\sigma_{\mathbf{q}_*}^2(\mathbf{k})$, can serve as a surrogate for the data density around $\mathbf{q}_*$ as experienced by $\mathbf{k}$. This insight allows us to use the variance function as a principled, data-driven way to measure the similarity between points on the data manifold. We provide a formal theoretical justification for this approach in the Appendix A.2. Building on this connection, we next reformulate the attention mechanism itself.

## 3.3 Geometry-Aware Attention Scoring

Building upon this local Gaussian process framework, we reformulate the attention score using the negative predictive variance. The similarity score between a query $\mathbf{q}$ and a key $\mathbf{k}$ is defined as:

$$\text{score}(\mathbf{q}, \mathbf{k}) = -(\mathbf{k} - \mathbf{q})^\top G(\mathbf{q})(\mathbf{k} - \mathbf{q}) \tag{5}$$

where $G(\mathbf{q})$ is the local geometry matrix estimated at the query's location. This score function computes the negative squared Mahalanobis distance, where the metric is adapted to the local data geometry. The softmax function then produces the final attention weights.

This geometry-aware scoring function forms the core of our **Local Geometry Attention (LGA)** mechanism. By leveraging the local geometric information encoded in the matrix $G(\mathbf{q})$, our approach moves beyond simple Euclidean similarity to better capture relationships between time series elements along the data manifold. This leads to more effective attention allocation for complex temporal patterns, as demonstrated in Figure 3.

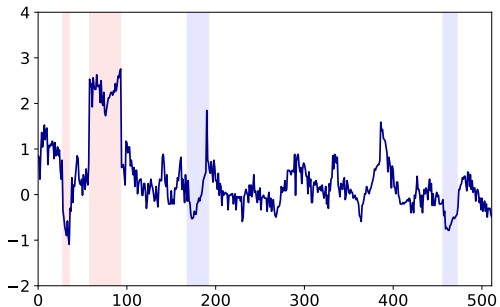

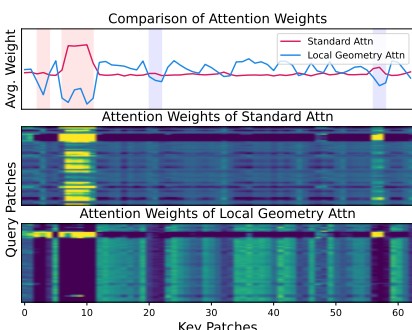

(a) An input time series with both natural (blue-shaded) and artificial (red-shaded) noisy regions

(b) Comparison of attention weights between standard attention and LGA

Figure 3: Effectiveness of LGA on a real-world time series with corruptions. The input series in (a) is corrupted with noise. (b) shows that while standard attention is distracted by these noisy regions, LGA successfully diminishes their influence. This robust handling of noise enables LGA to achieve a superior MSE of 0.533, compared to 0.616 from standard attention.

**Connection to Riemannian Geometry.** The proposed geometry-aware attention scoring in Equation (5) also has a strong theoretical interpretation within the framework of Riemannian geometry. On a data manifold $\mathcal{M}$ equipped with a Riemannian metric tensor $G$, the distance between two points $\mathbf{q}, \mathbf{k} \in \mathcal{M}$ is given by the geodesic distance, $\mathrm{dist}(\mathbf{q}, \mathbf{k}) = \inf_\gamma \int_0^1 \sqrt{\gamma'(t)^\top G(\gamma(t))\gamma'(t)} \, dt$, where the infimum is taken over all smooth paths $\gamma$ connecting $\mathbf{q}$ and $\mathbf{k}$. While computationally intractable, for a key $\mathbf{k}$ in a small neighborhood of a query $\mathbf{q}$, the squared geodesic distance can be approximated by a first-order Taylor expansion:

$$\mathrm{dist}(\mathbf{q}, \mathbf{k})^2 \approx (\mathbf{k} - \mathbf{q})^\top G(\mathbf{q})(\mathbf{k} - \mathbf{q}).$$

This expression reveals that the local Riemannian geometry induces a Mahalanobis distance. Consequently, our proposed attention score, $\mathrm{score}(\mathbf{q}, \mathbf{k}) = -(\mathbf{k} - \mathbf{q})^\top G(\mathbf{q})(\mathbf{k} - \mathbf{q})$, can be interpreted as the negative squared geodesic distance approximation. In this view, the matrix $G(\mathbf{q})$ derived from our local Gaussian process framework in Equation (4) serves as an empirical estimate of the local Riemannian metric tensor at the query point, allowing the attention mechanism to adapt to the intrinsic curvature of the data manifold.

This formulation provides a theoretically sound, geometry-aware attention mechanism. However, the direct computation of the metric tensor $G(\mathbf{q})$ for every query, which requires access to the full set of keys as per Equation (4), is computationally prohibitive for large-scale models. Addressing this computational challenge requires an efficient implementation.

### 3.4 Implementation of Local Geometry Attention

To make Local Geometry Attention (LGA) computationally feasible, we train a small network, $f_\theta$, to directly approximate the metric tensor $G(\mathbf{q})$ from a given query vector $\mathbf{q}$:

$$G(\mathbf{q}) \approx f_\theta(\mathbf{q}) \tag{6}$$

This is motivated by the insight that a position of query on the data manifold implicitly defines its local geometric structure. As universal function approximators, neural networks are well-suited to learn this mapping. For each attention head, we employ a separate network $f_{\theta_h}$ to predict its corresponding metric tensor.

To ensure the network $f_\theta$ generalizes well, we train it on two sets of query vectors: (1) a subset $\mathcal{S}_\mathrm{real}$ randomly sampled from the actual query vectors that appear during training, and (2) a set $\mathcal{S}_\mathrm{gen}$ of randomly generated vectors designed to explore a broader region of the representation space. For efficiency, we approximate the target metric tensor $G_\mathrm{true}$ (computed via Equation (4)) as a diagonal matrix. This is a practical trade-off that assumes independence between local feature dimensions to ensure computational tractability.

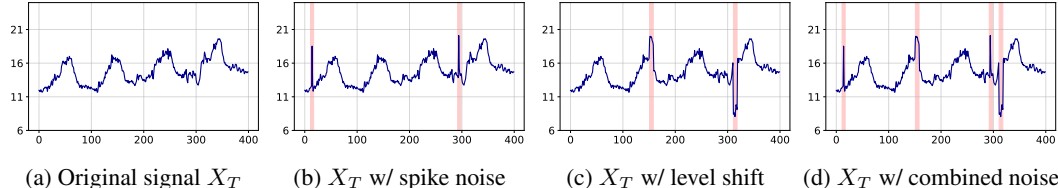

(a) Original signal $X_T$    (b) $X_T$ w/ spike noise    (c) $X_T$ w/ level shift    (d) $X_T$ w/ combined noise

Figure 4: Visualization of synthetic corruptions on the ETTm1 dataset. (a) Original time series without any corruption. (b) Time series with spike corruptions. (c) Time series with level shift corruptions. (d) Time series with a combination of corruptions, demonstrating how our TSRBench creates challenging test cases for time series models.

The networks are then trained to minimize the mean squared error between their prediction and the true metric tensor. The total loss is a weighted sum over all layers $L$ and heads $H$:

$$\mathcal{L}_G = \frac{1}{L \cdot H} \sum_{l=1}^{L} \sum_{h=1}^{H} \left[ \frac{1}{|\mathcal{S}_{\text{real}}^{h,l}|} \sum_{\mathbf{q} \in \mathcal{S}_{\text{real}}^{h,l}} \mathcal{L}(G_{\text{true}}(\mathbf{q}), f_{\theta_h^l}(\mathbf{q})) + \frac{1}{|\mathcal{S}_{\text{gen}}|} \sum_{\mathbf{q} \in \mathcal{S}_{\text{gen}}} \mathcal{L}(G_{\text{true}}(\mathbf{q}), f_{\theta_h^l}(\mathbf{q})) \right] \quad (7)$$

This pre-learning strategy decouples the expensive geometry estimation from training and inference of the model. Consequently, the local geometric structure can be rapidly approximated, making LGA practical for modern deep learning models. The complete algorithm is detailed in Appendix B.

## 4 CORRUPTED TIME SERIES BENCHMARK: TSRBENCH

To address the gap in standardized robustness evaluation for time series, we introduce **TSRBench**, a new benchmark designed to assess model performance against realistic, statistically-grounded corruptions. Unlike adversarial attacks, TSRBench focuses on corruptions reflecting naturally occurring data degradation, such as sensor limitations or external shocks (Si et al., 2024; Schmidl et al., 2022).

While evaluation on real-world datasets with genuine anomalies is ideal, it presents challenges for the specific task of *forecasting robustness*. A fair evaluation requires the ground truth of the test set to be clean, which necessitates precise, time-aligned anomaly labels to exclude corrupted periods from the loss calculation. However, most real-world forecasting benchmarks lack such labels, making it impossible to reliably isolate the effects of input corruptions on future predictions. Our TSRBench addresses this critical gap by providing a controlled environment where corruptions are systematically injected into the input, while the ground truth for the forecasting horizon remains clean. This enables a principled and reproducible assessment of model robustness that is difficult to achieve otherwise.

Our benchmark introduces two canonical corruption types: *spikes* and *level shifts* (Wang et al., 2021; Lavin & Ahmad, 2015), which model phenomena like transient bursts and sustained deviations. This provides a more realistic test than common augmentation like jittering (Iglesias et al., 2023). To support a systematic evaluation, we adopt statistically grounded noise injection methods with controllable severity levels (Siffer et al., 2017; Wunderlich & Sklar, 2023).

**Local Corruption Functions.** Given an original time series $X \in \mathbb{R}^T$, we generate two types of corruptions: spikes, $\varepsilon^{\text{spike}}$, and level shifts, $\varepsilon^{\text{shift}}$. The timing of these corruption events is governed by a Poisson process with rate $\lambda$, reflecting the random arrival of discrete perturbations.

Each corruption type is defined by an amplitude parameter $h$ and duration parameters ($d_1, d_2$, or $d$). For a corruption event starting at time $\tau$, the spike function $\varepsilon_\tau^{\text{spike}}(s)$ simulates an asymmetric exponential spike, while the level shift function $\varepsilon_\tau^{\text{shift}}(s)$ models a flat shift. These are defined as follows:

$$\varepsilon_\tau^{\text{spike}}(s) = \begin{cases} h_\tau^{\text{spike}} \cdot \exp\left(-\frac{\ln(\beta)}{d_1}(s - \tau - d_1)\right) & \text{if } \tau \leq s < \tau + d_1 \\ h_\tau^{\text{spike}} \cdot \exp\left(\frac{\ln(\beta)}{d_2}(s - \tau - d_1)\right) & \text{if } \tau + d_1 \leq s \leq \tau + d_1 + d_2 \end{cases} \tag{8}$$

$$\varepsilon_\tau^{\text{shift}}(s) = h_\tau^{\text{shift}} \quad \text{for } \tau \leq s < \tau + d$$

Here, the duration parameters $(d_1, d_2, d)$ are sampled from a geometric distribution with parameter $p$, and the sharpness is fixed at $\beta = 10^{-4}$. Crucially, the corruption amplitudes $(h_\tau^{\text{spike}}, h_\tau^{\text{shift}})$ are not arbitrary. They are calibrated based on time-varying statistical thresholds determined by a significance level $q$. This process, which uses the DSPOT algorithm (Siffer et al., 2017) to model extreme values, ensures that the injected corruptions represent statistically significant but realistic deviations from the normal behavior of signal. The complete generation algorithm is detailed in the Appendix F.1 (Algorithm 2).

Each final corrupted signal is formed by summing the noise instances from all events:

$$\varepsilon^{\text{spike}}(t) = \sum_{\tau \in \mathcal{T}} \varepsilon_\tau^{\text{spike}}(t), \quad \text{and} \quad \varepsilon^{\text{shift}}(t) = \sum_{\tau \in \mathcal{T}} \varepsilon_\tau^{\text{shift}}(t) \tag{9}$$

where $\mathcal{T}$ is the set of starting times for corruption events. Figure 4 visualizes examples of these corruptions.

Table 1: Parameter settings defining the five corruption severity levels. From level 1 to 5, the expected corruption frequency ($\lambda$) and duration ($p$) increase, while the amplitude significance level ($q$) decreases. They collectively intensifying the corruption.

| Params. | Description | 1 | 2 | 3 | 4 | 5 |
|---|---|---|---|---|---|---|
| $\lambda$ | Expected frequency of corruptions | 0.002 | 0.004 | 0.004 | 0.008 | 0.008 |
| $p$ | Expected duration of each corruption | 6 | 9 | 12 | 12 | 15 |
| $q$ | Significance level of amplitude | 0.0016 | 0.0016 | 0.0004 | 0.0004 | 0.0001 |

**Corruption Severity Levels.** To comprehensively evaluate model robustness, we designed five severity levels that represent gradually increasing data degradation. The corrupted time series is generated by adding the synthesized corruptions to the original signal as described.

We control the severity of corruptions at each level using a parameter triplet $(\lambda, p, q)$, which respectively regulate the frequency, expected duration, and amplitude significance of the anomalies. The specific values for each level, summarized in Table 1, were established through extensive experimentation to create a progressive scale of difficulty, from Level 1 (mild) to Level 5 (frequent, prolonged, and high-magnitude corruption). We conducted extensive experiments with various parameter configurations to determine these settings, which are presented in Appendix F.2.

## 5 EXPERIMENTS

We empirically evaluate the robustness of LGA using our proposed TSRBench on six standard forecasting datasets: Weather[1], Electricity[2], and ETT (Zhou et al., 2021) (ETTh1, ETTh2, ETTm1, ETTm2). We integrate LGA into the PatchTST architecture (Nie et al., 2023), referred to as PatchLGA, by replacing its standard Scaled-Dot Product (SDP) attention. We compare PatchLGA against the original PatchTST and other strong baselines, including TimeMixer (Wang et al., 2024), CATS (Kim et al., 2024), and iTransformer (Liu et al., 2024), to assess its effectiveness across various model types.

### 5.1 LONG-TERM TIME SERIES FORECASTING RESULTS UNDER REALISTIC CORRUPTIONS

We evaluate forecasting models on TSRBench across six corruption levels (0-5) with an input length of 512. Table 10 summarizes the results for combined spike and level shift corruptions, averaged

---

[1] https://www.bgc-jena.mpg.de/wetter/
[2] https://archive.ics.uci.edu/ml/datasets/ElectricityLoadDiagrams20112014

Table 2: Performance comparison of PatchLGA, PatchTST, and TimeMixer across different datasets under combined corruptions. Results show average performance across forecasting horizons {96, 192, 336, 720}. We report MSE and MAE at varying severity levels 0-5 (0 = original data). Lower values are better, with the best results highlighted in **bold**.

| Dataset | | ETTm1 | | | | | | ETTm2 | | | | | | Weather | | | | |
|---|---|---|---|---|---|---|---|---|---|---|---|---|---|---|---|---|---|---|
| Model | | **PatchLGA** | | PatchTST | | TimeMixer | | **PatchLGA** | | PatchTST | | TimeMixer | | **PatchLGA** | | PatchTST | | TimeMixer |
| Metric | | MSE | MAE | MSE | MAE | MSE | MAE | MSE | MAE | MSE | MAE | MSE | MAE | MSE | MAE | MSE | MAE | MSE MAE |
| Severity 0 | | **0.351** | **0.379** | 0.352 | 0.382 | 0.360 | 0.390 | 0.257 | **0.316** | **0.256** | 0.317 | 0.259 | 0.322 | 0.227 | 0.265 | **0.225** | **0.264** | 0.227 0.266 |
| 1 | | **0.359** | **0.385** | **0.359** | 0.389 | 0.366 | 0.396 | **0.264** | **0.326** | **0.264** | 0.328 | 0.265 | 0.330 | **0.238** | **0.284** | 0.239 | **0.284** | 0.242 0.288 |
| 2 | | **0.372** | **0.397** | 0.375 | 0.402 | 0.385 | 0.409 | **0.276** | **0.335** | 0.277 | 0.338 | 0.279 | 0.341 | **0.264** | **0.313** | 0.266 | 0.314 | 0.275 0.320 |
| 3 | | **0.519** | **0.468** | 0.614 | 0.507 | 0.594 | 0.499 | **0.304** | **0.355** | 0.308 | 0.360 | 0.310 | 0.364 | **0.301** | **0.338** | 0.306 | 0.339 | 0.326 0.350 |
| 4 | | **0.617** | **0.526** | 0.695 | 0.558 | 0.716 | 0.560 | **0.356** | **0.395** | 0.361 | 0.400 | 0.373 | 0.409 | **0.361** | **0.389** | 0.369 | 0.384 | 0.441 0.421 |
| 5 | | **0.734** | **0.577** | 0.839 | 0.613 | 0.837 | 0.613 | **0.421** | **0.428** | 0.431 | 0.434 | 0.459 | 0.450 | **0.454** | **0.423** | 0.491 | **0.423** | 0.576 0.464 |
| Dataset | | ETTh1 | | | | | | ETTh2 | | | | | | Electricity | | | | |
| Model | | **PatchLGA** | | PatchTST | | TimeMixer | | **PatchLGA** | | PatchTST | | TimeMixer | | **PatchLGA** | | PatchTST | | TimeMixer |
| Metric | | MSE | MAE | MSE | MAE | MSE | MAE | MSE | MAE | MSE | MAE | MSE | MAE | MSE | MAE | MSE | MAE | MSE MAE |
| Severity 0 | | **0.415** | **0.428** | **0.415** | 0.429 | 0.436 | 0.445 | **0.337** | **0.385** | 0.343 | 0.387 | 0.351 | 0.393 | 0.161 | 0.255 | **0.160** | **0.254** | 0.165 0.259 |
| 1 | | **0.416** | **0.430** | **0.416** | **0.430** | 0.437 | 0.447 | **0.338** | **0.387** | 0.343 | 0.389 | 0.352 | 0.394 | **0.168** | **0.264** | 0.169 | 0.265 | 0.172 0.268 |
| 2 | | **0.420** | **0.435** | 0.421 | 0.436 | 0.441 | 0.451 | **0.345** | **0.397** | 0.349 | **0.397** | 0.363 | 0.406 | **0.179** | **0.276** | 0.184 | 0.279 | 0.184 0.280 |
| 3 | | **0.471** | **0.465** | 0.474 | 0.468 | 0.498 | 0.483 | **0.358** | **0.407** | 0.362 | **0.407** | 0.384 | 0.420 | **0.193** | **0.288** | 0.201 | 0.293 | 0.201 0.293 |
| 4 | | **0.547** | **0.510** | 0.550 | 0.513 | 0.572 | 0.526 | **0.382** | **0.425** | 0.391 | 0.427 | 0.416 | 0.443 | **0.226** | **0.317** | 0.232 | 0.321 | 0.237 0.323 |
| 5 | | **0.712** | **0.580** | 0.724 | 0.588 | 0.733 | 0.596 | **0.404** | **0.436** | 0.427 | 0.445 | 0.459 | 0.465 | **0.284** | **0.352** | 0.290 | 0.358 | 0.312 0.361 |

over four forecasting horizons. First, the results at Severity 0 serve as a crucial baseline to ensure that our proposed robust attention mechanism does not degrade performance on the original, clean data. This confirms that integrating LGA does not compromise the inherent forecasting capability of model on clean data. As corruption severity increases, PatchLGA consistently shows the most robust performance. Its advantage becomes more pronounced at higher severity levels; for example, at level 5, PatchLGA achieves a 12.3% MSE reduction on ETTm1 and is 21.2% lower than TimeMixer on the large Weather dataset. These results confirm the effectiveness of modeling local geometry for enhancing robustness, especially when data quality cannot be guaranteed.

## 5.2 COMPARISON WITH ALTERNATIVE ROBUST ATTENTION METHODS

We evaluated robust attention mechanisms on time series forecasting with corruptions. Our proposed LGA consistently outperforms both standard attention and other robust attention variants including MoM (Han et al., 2023) and Elliptical attention (Nielsen et al., 2024).

Table 3: Averaged MSE comparison on ETTm1 for LGA, SDP, MoM, and Elliptical attention. Lower values indicate better performance, with the best result for each severity level highlighted in **bold**.

| Model | | PatchTST | | |
|---|---|---|---|---|
| Severity | SDP | MoM | Ellip. | LGA |
| 1 | **0.359** | 0.375 | 0.360 | **0.359** |
| 2 | 0.375 | 0.397 | 0.374 | **0.372** |
| 3 | 0.614 | 0.670 | 0.722 | **0.519** |
| 4 | 0.695 | 0.871 | 0.755 | **0.617** |
| 5 | 0.839 | 1.016 | 0.880 | **0.734** |

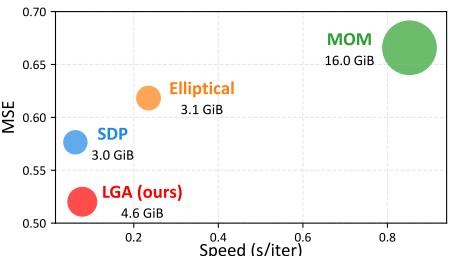

Figure 5: Comparison of MSE, training speed, and memory usage. Attention closer to the lower-left corner achieves better trade-offs between accuracy and efficiency.

As shown in Table 3, while robust attention mechanisms successful in vision and language tasks underperform compared to standard attention when applied to corrupted time series data, our LGA specifically designed for temporal structure exhibits the smallest performance degradation under corruption. Unlike Elliptical attention, which trades speed for memory efficiency without robustness gains (Fig. 5), LGA effectively leverages local geometry in time series while seeing a moderate increase in memory usage, yet maintaining a training speed comparable to standard attention.

## 5.3 Robustness Analysis Across Attention Mechanisms

We conducted experiments to evaluate the effectiveness of LGA across various types of attention strategies. In time series transformers, two primary attention approaches exist: temporal attention, which focuses on relationships between time steps within each variable, and channel attention, which emphasizes relationships between different variables at each time step.

Table 4: Comparison of attention mechanisms on ETTm1 under 5 severity levels. Values show MSE averaged across forecasting horizons {96, 192, 336, 720}. **Bold** values indicate the better performance for each experiment.

| Model | | | | | | | | | | | | | | | | | | |
|---|---|---|---|---|---|---|---|---|---|---|---|---|---|---|---|---|---|---|
| | PatchTST | | | | | | CATS | | | | | | iTransformer | | | | | |
| Noise Type | Combined | | Level Shift | | Spike | | Combined | | Level Shift | | Spike | | Combined | | Level Shift | | Spike | |
| Attn Type | LGA | SDP | LGA | SDP | LGA | SDP | LGA | SDP | LGA | SDP | LGA | SDP | LGA | SDP | LGA | SDP | LGA | SDP |
| Severity 1 | **0.359** | 0.359 | **0.359** | 0.359 | **0.352** | 0.352 | **0.345** | **0.345** | **0.344** | 0.345 | 0.340 | **0.338** | **0.425** | 0.429 | **0.423** | 0.427 | **0.411** | 0.413 |
| Severity 2 | **0.372** | 0.375 | **0.370** | 0.374 | **0.353** | 0.353 | **0.362** | 0.364 | **0.359** | 0.361 | 0.342 | **0.340** | **0.459** | 0.465 | **0.455** | 0.462 | **0.413** | 0.414 |
| Severity 3 | **0.519** | 0.615 | **0.506** | 0.608 | **0.362** | 0.364 | **0.662** | 0.779 | **0.634** | 0.765 | **0.349** | 0.355 | **0.783** | 0.801 | **0.756** | 0.771 | **0.434** | 0.440 |
| Severity 4 | **0.617** | 0.695 | **0.602** | 0.691 | **0.367** | 0.369 | **0.972** | 0.986 | **0.923** | 0.965 | **0.357** | 0.366 | **1.000** | 1.027 | **0.949** | 0.970 | **0.453** | 0.465 |
| Severity 5 | **0.734** | 0.839 | **0.718** | 0.848 | **0.372** | 0.374 | **1.037** | 1.102 | **1.003** | 1.084 | **0.358** | 0.366 | **1.266** | 1.309 | **1.202** | 1.229 | **0.469** | 0.485 |

As shown in Table 4, LGA enhances robustness across diverse attention architectures. PatchTST, with its temporal self-attention, consistently achieves the most significant performance gains. While CATS shows a notable peak improvement of up to 17.1%, its gains are less consistent, likely because its cross-attention operates on linearly embedded noisy inputs. iTransformer displays modest but stable improvements because it applies a linear embedding to the entire time series, which disrupts the local geometry created by temporal periodic patterns. These results validate that LGA is a broadly applicable technique for improving model robustness, demonstrating its benefit for temporal self-attention, cross-attention, and channel-wise attention.

## 5.4 Impact of Input Length on Robustness under Realistic Corruptions

To investigate the relationship between temporal context and model robustness, we evaluated PatchLGA, PatchTST, and TimeMixer by varying input sequence lengths under corrupted conditions. The results in Figure 6 demonstrate superior capability of PatchLGA to leverage long historical contexts for accurate forecasting in the presence of noise.

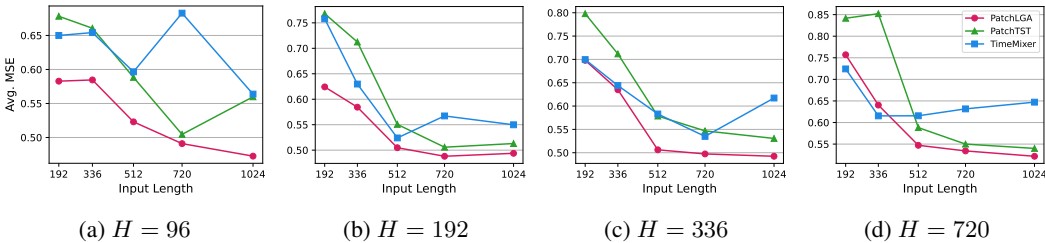

(a) $H = 96$    (b) $H = 192$    (c) $H = 336$    (d) $H = 720$

Figure 6: Performance comparison of different models on ETTm1 across varying input lengths under corruption. The y-axis shows average MSE (lower is better) for forecasting different horizons $H$, while the x-axis represents input sequence length.

Across all forecasting horizons, PatchLGA (red line) shows a consistent and significant reduction in MSE as the input sequence length increases, achieving the lowest MSE in nearly all experimental settings. This highlights its strong benefit in utilizing extended temporal data to mitigate the impact of corruptions. In contrast, while PatchTST (green line) also benefits from longer sequences, its performance remains inferior to PatchLGA. The linear model, TimeMixer (blue line), fails to capitalize on increased context, with its performance stagnating or degrading. In conclusion, these results provide compelling evidence that PatchLGA is effectively designed to enhance forecasting accuracy by effectively processing long, noisy input sequences, underscoring its suitability for real-world applications.

## 5.5 QUALITATIVE ANALYSIS OF ATTENTION GEOMETRY

To intuitive understand how LGA handles corrupted data, we visualize the attention score distribution in the query-key space. We utilize Principal Component Analysis (PCA) to project the key and query vectors from a representative attention head into 2D space. The experiment was conducted on the ETTm1 dataset under Severity Level 5 corruption.

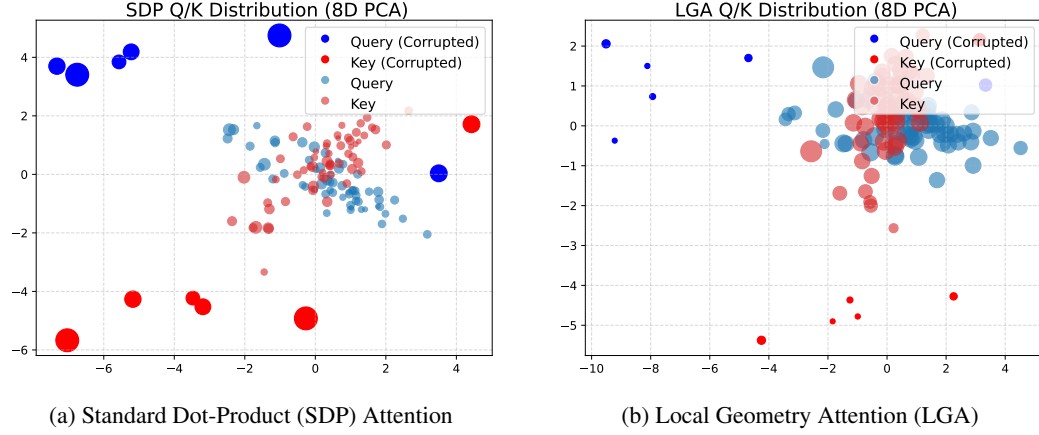

(a) Standard Dot-Product (SDP) Attention  (b) Local Geometry Attention (LGA)

Figure 7: PCA visualization of attention scores under corruption. Point size indicates attention magnitude, and dark points denote corrupted regions. Unlike **(a)** SDP, which assigns high weights to noise, **(b)** LGA effectively suppresses these off-manifold anomalies.

As observed in Figure 7, SDP attention (a) is prone to distraction by outliers, assigning them large attention weights. In contrast, LGA (b) successfully suppresses these outliers, assigning them minimal attention weights due to their distance from the high-density data manifold. Consequently, by filtering out anomalies, LGA ensures the model remains focused on intrinsic temporal patterns, leading to significantly more robust and accurate forecasting under severe corruption.

## 5.6 COMPUTATIONAL EFFICIENCY OF LOCAL GEOMETRY ATTENTION

While LGA introduces operations for metric prediction and Mahalanobis scoring, its practical computational overhead remains marginal. Because the Feed-Forward Network (FFN) governs the primary computational budget of a Transformer block, the relative cost of the attention module is significantly offset. As shown in Table 7, replacing SDP attention with LGA increases the theoretical Floating Point Operations (FLOPs) of a full block by only 11.7% during inference and 21.6% during training. This confirms

Table 5: Relative overhead of LGA compared to SDP (Full Transformer Block).

| Method | Inference | Training |
|---|---|---|
| SDP | 24.78 G | 73.99 G |
| **LGA (Ours)** | 27.67 G | 89.96 G |
| Overhead | +11.7% | +21.6% |

that LGA effectively captures local data geometry and provides robust modeling with minimal impact on computational efficiency. A comprehensive hardware-agnostic analysis and further algorithmic optimizations are detailed in Appendix B.2.

## 6 CONCLUSION

We introduced Local Geometry Attention (LGA), an attention mechanism grounded in local Gaussian process theory that adapts to the intrinsic geometry of time series data. To validate its effectiveness, we also developed TSRBench, the first standardized benchmark for evaluating models against realistic corruptions. Our experiments confirm that geometry-aware approach of LGA provides a substantial robustness advantage over strong baselines, especially under severe conditions. While our efficient matrix approximation can be explored further, this work provides a powerful new framework and a critical evaluation tool for developing forecasting models ready for real-world deployment.

ACKNOWLEDGMENTS

This research was supported by Basic Science Research Program through the National Research Foundation of Korea(NRF) funded by the Ministry of Education(No. RS-2025-25420430). This research was also partly supported by the Institute of Information & communications Technology Planning & Evaluation (IITP) grant funded by the Korea government (MSIT) (No. RS-2022-II220984, Development of Artificial Intelligence Technology for Personalized Plug-and-Play Explanation and Verification of Explanation), and the National Research Foundation of Korea (NRF) grant funded by the Korean government (MSIT) (No. RS-2024-00338859)

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

# A   Mathematical Supplement to Section 3.2

This supplement provides a detailed theoretical foundation for our proposed Local Geometry Attention (LGA). We begin by framing our approach within the context of locally weighted regression with generalized basis functions, which naturally leads to the local kernel-covariance matrix. We then present a theoretical bound on the predictive variance of our Local Gaussian Process model, justifying its use as a reliable surrogate for data density.

## A.1   Local Basis Function Regression

Locally weighted regression, or loess, is a non-parametric method that fits simple models to localized subsets of data. This approach is highly effective for ordered data, like time series, as it can capture complex patterns without assuming a global model structure. A prime example is the Seasonal-Trend decomposition based on Loess (STL) algorithm, widely used for its robustness in time series decomposition.

Our work extends this framework by using the difference between key and query embeddings as generalized, data-driven basis functions. For each target point $x_*$ (which maps to a query vector $\mathbf{q}_*$), we solve a separate weighted least squares problem:

$$\min_{\beta(\mathbf{q}_*)} \sum_{i=1}^{T} K(\mathbf{k}_i, \mathbf{q}_*) \left[ y_i - (\mathbf{k}_i - \mathbf{q}_*)^\top \beta(\mathbf{q}_*) \right]^2, \tag{10}$$

where $\{\mathbf{k}_i\}_{i=1}^{T}$ are the key vectors corresponding to the input data. The term $(\mathbf{k}_i - \mathbf{q}_*)$ serves as our basis function, and the weight $K(\mathbf{k}_i, \mathbf{q}_*)$ is determined by a Gaussian kernel:

$$K(\mathbf{k}_i, \mathbf{q}_*) = \exp\left( -\|\mathbf{k}_i - \mathbf{q}_*\|^2 / 2h^2 \right). \tag{11}$$

Let $\mathbf{B}(\mathbf{q}_*)$ be the $T \times d$ regression matrix whose $i$-th row is $(\mathbf{k}_i - \mathbf{q}_*)^\top$, and let $\mathbf{W}(x_*)$ be the $T \times T$ diagonal matrix of normalized weights $\omega_i(x_*)$:

$$\omega_i(x_*) = \frac{K(\mathbf{k}_i, \mathbf{q}_*)}{\sum_{j=1}^{T} K(\mathbf{k}_j, \mathbf{q}_*)}. \tag{12}$$

The solution to the weighted least squares problem is $\hat{\beta}(\mathbf{q}_*) = \Sigma(x_*)^{-1} \mathbf{B}(\mathbf{q}_*)^\top \mathbf{W}(x_*) \mathbf{y}$, where $\Sigma(x_*)$ is the local kernel-covariance matrix identical to Equation (1):

$$\Sigma(x_*) = \mathbf{B}(\mathbf{q}_*)^\top \mathbf{W}(x_*) \mathbf{B}(\mathbf{q}_*) = \sum_{i=1}^{T} \omega_i(x_*)(\mathbf{k}_i - \mathbf{q}_*)(\mathbf{k}_i - \mathbf{q}_*)^\top.$$

The covariance of the estimated parameters $\hat{\beta}(\mathbf{q}_*)$, assuming observation noise with covariance $\Sigma_{obs}$, is given by $\Sigma^\beta = (\mathbf{B}^\top \mathbf{W} \mathbf{B})^{-1} (\mathbf{B}^\top \mathbf{W} \Sigma_{obs} \mathbf{W}^\top \mathbf{B})(\mathbf{B}^\top \mathbf{W}^\top \mathbf{B})^{-1}$. If we make a simplifying assumption that the weights are chosen as the inverse of the observation covariance, $\mathbf{W} \approx \Sigma_{obs}^{-1}$, this expression simplifies to $\Sigma^\beta \approx (\mathbf{B}^\top \mathbf{W} \mathbf{B})^{-1} = \Sigma(x_*)^{-1}$. This connection motivates using the local kernel-covariance matrix $\Sigma(x_*)$ within a probabilistic framework, which we term a "Local Gaussian Process" as it effectively constructs a GP conditioned on a localized data neighborhood.

## A.2   Theoretical Bound on Predictive Variance

**Theorem 1.** *Let $\mathbb{P}$ be a fixed but unknown probability distribution over a space $\mathcal{H}$, with no atomic components and support contained within the unit ball. For the predictive variance function (3) of our Local Gaussian Process, let $\hat{\sigma}^2 = \max_{\mathbf{k} \in \{\mathbf{k}_i\}} \sigma_{\mathbf{q}_*}^2(\mathbf{k})$ denote the maximum predictive variance over a data sequence of size $T$. Then, for any $\varepsilon > 0$, with probability at least $1 - \delta$, the following inequality holds:*

$$\mathbb{P}\left( \mathbf{k} : \sigma_{\mathbf{q}_*}^2(\mathbf{k}) > \hat{\sigma}^2 + 2\varepsilon \right) \leq \frac{2}{T} \left( \rho + \log\left( \frac{T^2}{2\delta} \right) \right),$$

*for some $\rho = \mathcal{O}(\log(T))$.*

*Proof.* The derivation is based on the method in Kim & Lee (2007), adapted to our framework. The predictive variance is given by:

$$\sigma^2_{\mathbf{q}_*}(\mathbf{k}) = (\mathbf{k} - \mathbf{q}_*)^\top G(x_*)(\mathbf{k} - \mathbf{q}_*), \tag{13}$$

where $G(x_*) = \sigma^2[\Sigma(x_*) + \sigma^2 I]^{-1}$. We define a weighted feature mapping $\psi(\mathbf{k}_i) = \sqrt{\omega_i(x_*)}(\mathbf{k}_i - \mathbf{q}_*)$ and a design matrix $\mathbf{\Psi}$ whose $i$-th row is $\psi(\mathbf{k}_i)^\top$. The local kernel-covariance matrix is then $\Sigma(x_*) = \mathbf{\Psi}^\top \mathbf{\Psi}$.

Using the Sherman-Woodbury-Morrison formula, the predictive variance can be expressed as:

$$\sigma^2_{\mathbf{q}_*}(\mathbf{k}) = \tilde{k}(\mathbf{k}, \mathbf{k}) - \tilde{\mathbf{k}}_{\mathbf{k}}^\top (\tilde{\mathbf{K}} + \sigma^2 \mathbf{I})^{-1} \tilde{\mathbf{k}}_{\mathbf{k}},$$

where the weighted kernel is $\tilde{k}(\mathbf{k}, \mathbf{k}') = \psi(\mathbf{k})^\top \psi(\mathbf{k}')$, $\tilde{\mathbf{K}}$ is the Gram matrix with entries $\tilde{k}(\mathbf{k}_i, \mathbf{k}_j)$, and $\tilde{\mathbf{k}}_{\mathbf{k}}$ is a vector with entries $\tilde{k}(\mathbf{k}_i, \mathbf{k})$.

The function $h(\mathbf{k}) := \tilde{\mathbf{k}}_{\mathbf{k}}^\top (\tilde{\mathbf{K}} + \sigma^2 \mathbf{I})^{-1} \tilde{\mathbf{k}}_{\mathbf{k}}$ can be shown to be linear in a transformed feature space. Let $\mathbf{C} = \tilde{\mathbf{K}} + \sigma^2 \mathbf{I}$ and $\mathbf{w} = \mathbf{\Psi}^\top \mathbf{C}^{-1} \mathbf{\Psi}$. Then,

$$h(\mathbf{k}) = \psi(\mathbf{k})^\top \mathbf{w} \psi(\mathbf{k}) = \mathrm{tr}\left(\psi(\mathbf{k})\psi(\mathbf{k})^\top \mathbf{w}\right).$$

By defining a new feature map $\Psi_2(\mathbf{k}) = \mathrm{Vec}(\psi(\mathbf{k})\psi(\mathbf{k})^\top)$, we can write $h(\mathbf{k}) = \mathbf{w}^\top \Psi_2(\mathbf{k})$, making it linear. Thus, $\sigma^2_{\mathbf{q}_*}(\mathbf{k}) = \tilde{k}(\mathbf{k}, \mathbf{k}) - h(\mathbf{k})$ is also linear in this feature space (assuming $\tilde{k}(\mathbf{k}, \mathbf{k})$ is constant or slowly varying).

Applying the theoretical results from Smola & Schölkopf (1998); Schölkopf et al. (2001), we obtain that, with probability at least $1 - \delta$, and letting $\hat{\sigma}^2_m = \min_{\mathbf{k} \in \{\mathbf{k}_i\}} h(\mathbf{k}) = \tilde{k} - \hat{\sigma}^2$, for all $\varepsilon > 0$, we have:

$$\mathbb{P}\left\{\mathbf{k} : \sigma^2_{\mathbf{q}_*}(\mathbf{k}) > \hat{\sigma}^2 + 2\varepsilon\right\} \le \frac{2}{T}\left(\rho + \log\left(\frac{T^2}{2\delta}\right)\right),$$

where

$$\rho = \frac{c_1 \log(c_2 \hat{\varepsilon}^2 T)}{\hat{\varepsilon}^2} + \mathcal{D}_{\hat{\varepsilon}} \log\left(e\left(\frac{(2T-1)\hat{\varepsilon}}{\mathcal{D}_{\hat{\varepsilon}}} + 1\right)\right) + 2,$$

and the constants and terms are defined as follows:

$$c_1 = 4c^2, \quad c_2 = \frac{\ln(2)}{c^2}, \quad c = 103, \quad \hat{\varepsilon} = \frac{\varepsilon}{\|\mathbf{w}\|}, \quad \mathcal{D}_{\hat{\varepsilon}} = \mathcal{D}(\{\mathbf{k}_i\}, g, \tilde{k} - \hat{\sigma}^2).$$

This finding indicates that the variance function of a Local Gaussian Process serves as a reliable surrogate for capturing the support (or high density regions) of a high-dimensional data distribution. Notably, the estimated support set retains computational tractability even in high-dimensional regimes. □

# B  IMPLEMENTATION DETAILS OF LGA

This section provides a detailed description of our Local Geometry Attention (LGA) implementation. While the theoretical foundation in Section 3.2 involves computationally intensive operations, we introduce an efficient approximation that maintains the core benefits of geometry-aware attention while being tractable for large-scale time series forecasting. The complete algorithm is presented in Algorithm 1.

## B.1  IMPLEMENTATION OF $f_\theta$

As described in Section 3.4, we employ a neural network $f_\theta$ to approximate the metric tensor $G(\mathbf{q})$. The network architecture is a simple feedforward design. For each head $h$, the input query vector is projected from its head dimension $D_h$ to a higher-dimensional space $D_G$, passed through a non-linearity, and projected back to $D_h$ to produce the diagonal components of the metric tensor:

$$f_{\theta_h}(\mathbf{q}) = \mathrm{Softplus}(W_2^h(\mathrm{GELU}(W_1^h(\mathbf{q})))) \tag{14}$$

---

**Algorithm 1 Local Geometry Attention (LGA)**

---

**Require:** Query $\mathbf{Q} \in \mathbb{R}^{B \times L_Q \times D}$, Key $\mathbf{K} \in \mathbb{R}^{B \times L_K \times D}$, Value $\mathbf{V} \in \mathbb{R}^{B \times L_K \times D}$
**Require:** Model dimension $D$, number of heads $H$, head dimension $D_h = D/H$, noise variance $\sigma^2$
**Ensure:** Output tensor $\in \mathbb{R}^{B \times L_Q \times D}$
 1: **function** LGA($\mathbf{Q}, \mathbf{K}, \mathbf{V}$)
 2:     **Step 1: Linear projection and reshape to multi-head format**
 3:     $\mathbf{q}_s \leftarrow W_Q(\mathbf{Q})$, reshape to $(B, L_Q, H, D_h)$
 4:     $\mathbf{k}_s \leftarrow W_K(\mathbf{K})$, reshape to $(B, L_K, H, D_h)$
 5:     $\mathbf{v}_s \leftarrow W_V(\mathbf{V})$, reshape to $(B, L_K, H, D_h)$
 6:     **Step 2: Predict metric tensor**
 7:     Apply $f_{\theta_h}$ to $\mathbf{q}_s[:,:,h,:]$ for each head $h$
 8:     $\mathbf{G} \in \mathbb{R}^{B \times L_Q \times H \times D_h}$                                           ▷ Stack predictions across heads
 9:     **if** training **then**
10:         **Step 3: Estimate true metric and update prediction networks**
11:         $\mathbf{G}_{\text{true}} \leftarrow \text{ESTIMATETRUEMETRIC}(\mathbf{q}_s.\text{detach}(), \mathbf{k}_s.\text{detach}())$
12:         Update $\{f_{\theta_h}\}_{h=1}^{H}$ using $\mathcal{L}_G = \frac{1}{H}\sum_{h=1}^{H}\|\mathbf{G}[:,:,h,:] - \mathbf{G}_{\text{true}}[:,:,h,:]\|^2$
13:     **end if**
14:     **Step 4: Compute attention scores and aggregate values**
15:     $\mathbf{S} \leftarrow \text{MAHALANOBISSCORE}(\mathbf{q}_s, \mathbf{k}_s, \mathbf{G}.\text{detach}())$             ▷ $(B, L_Q, H, L_K)$
16:     $\mathbf{A} \leftarrow \text{softmax}(\mathbf{S})$ along last dimension             ▷ $(B, L_Q, H, L_K)$
17:     $\mathbf{O} \leftarrow \mathbf{A}\mathbf{v}_s$                      ▷ Weighted sum: $(B, L_Q, H, D_h)$
18:     **Step 5: Reshape and output projection**
19:     Reshape $\mathbf{O}$ to $(B, L_Q, H \cdot D_h)$ and apply $W_O$
20:     **return** output $\in \mathbb{R}^{B \times L_Q \times D}$
21: **end function**
22: **function** MAHALANOBISSCORE($\mathbf{q}, \mathbf{k}, \mathbf{G}$)
23:     Efficiently computes $S_{ij} = -(\mathbf{k}_j - \mathbf{q}_i)^\top \text{diag}(\mathbf{G}_i)(\mathbf{k}_j - \mathbf{q}_i)$ for all pairs
24:     Cross term: $\mathbf{C} \leftarrow (\mathbf{q} \odot \mathbf{G})\mathbf{k}^\top$                ▷ $(B, L_Q, H, L_K)$
25:     Quadratic term: $\mathbf{Q} \leftarrow \mathbf{G}(\mathbf{k} \odot \mathbf{k})^\top$           ▷ $(B, L_Q, H, L_K)$
26:     $\mathbf{S} \leftarrow 2\mathbf{C} - \mathbf{Q}$
27:     **return** $\mathbf{S} \in \mathbb{R}^{B \times L_Q \times H \times L_K}$
28: **end function**
29: **function** ESTIMATETRUEMETRIC($\mathbf{q}, \mathbf{k}$)
30:     **Sample queries and keys:**
31:     Sample $N_s$ indices from batch dimension
32:     $\mathbf{q}_{\text{real}} \leftarrow \mathbf{q}[\text{sampled}] \in \mathbb{R}^{N_s \times L_Q \times H \times D_h}$
33:     $\mathbf{q}_{\text{rand}} \sim \mathcal{U}(-5,5)^{N_s \times L_Q \times H \times D_h}$
34:     $\mathbf{q}_s \leftarrow \text{concat}(\mathbf{q}_{\text{real}}, \mathbf{q}_{\text{rand}}) \in \mathbb{R}^{2N_s \times L_Q \times H \times D_h}$
35:     $\mathbf{k}_s \leftarrow \mathbf{k}[\text{sampled}] \in \mathbb{R}^{N_s \times L_K \times H \times D_h}$
36:     **Compute Gaussian kernel weights:**
37:     For each query-key pair, compute squared distance: $d_{ij}^2 = \|\mathbf{q}_i - \mathbf{k}_j\|^2$
38:     $\omega_{ij} \leftarrow \text{softmax}_j(-d_{ij}^2 \cdot \text{scale})$        ▷ Normalize over keys: $(2N_s, L_Q, L_K, H)$
39:     **Compute weighted local covariance:**
40:     For each dimension $d$: $\Sigma_{i,d} = \sum_{j=1}^{L_K} \omega_{ij} \cdot (\mathbf{q}_{i,d} - \mathbf{k}_{j,d})^2$    ▷ $(2N_s, L_Q, H, D_h)$
41:     **Compute inverse covariance:**
42:     $\mathbf{G}_{\text{true}} \leftarrow (\mathbf{\Sigma} + \sigma^2)^{-1}$        ▷ Element-wise inversion: $(2N_s, L_Q, H, D_h)$
43:     **return** $\mathbf{G}_{\text{true}}$
44: **end function**

---

where $W_1^h \in \mathbb{R}^{D_h \times D_G}$ and $W_2^h \in \mathbb{R}^{D_G \times D_h}$ are learnable weight matrices. We use separate networks for each head to capture distinct geometric patterns. The GELU activation enables the learning of complex mappings, and the final Softplus activation ensures the positive definiteness of the metric tensor by guaranteeing its diagonal elements are positive.

The hyperparameter $D_G$ controls the expressiveness of the geometric approximation. As detailed in the main paper, we typically set $D_G$ to be 8 times the head dimension $D_h$. For a model with $L$ layers and $H$ heads per layer, the total number of parameters introduced by our metric prediction networks is $2 \cdot L \cdot H \cdot D_h \cdot D_G$. This represents a modest increase in model size but significantly enhances its ability to adapt to local data geometry.

## B.2 Efficient Score Computation

The computation of the LGA score, $-(\mathbf{k} - \mathbf{q})^\top \mathbf{G}(\mathbf{q})(\mathbf{k} - \mathbf{q})$, can be optimized for efficiency. Since we approximate $\mathbf{G}(\mathbf{q})$ as a diagonal matrix, the quadratic form expands to:

$$(\mathbf{k} - \mathbf{q})^\top \mathbf{G}(\mathbf{q})(\mathbf{k} - \mathbf{q}) = \mathbf{k}^\top \mathbf{G}(\mathbf{q})\mathbf{k} - 2\mathbf{q}^\top \mathbf{G}(\mathbf{q})\mathbf{k} + \mathbf{q}^\top \mathbf{G}(\mathbf{q})\mathbf{q} \tag{15}$$

When these scores are fed into a softmax function, any term that is constant for a given query $\mathbf{q}$ across all keys $\mathbf{k}$ can be dropped, as the softmax is invariant to constant shifts. The term $\mathbf{q}^\top \mathbf{G}(\mathbf{q})\mathbf{q}$ is one such constant. Therefore, we can simplify the computation to:

$$\text{score}(\mathbf{q}, \mathbf{k}) = -\mathbf{k}^\top \mathbf{G}(\mathbf{q})\mathbf{k} + 2\mathbf{q}^\top \mathbf{G}(\mathbf{q})\mathbf{k} \tag{16}$$

This simplification, combined with the diagonal approximation of $\mathbf{G}(\mathbf{q})$, significantly reduces the computational overhead. Since the metric tensor is learned by $f_\theta$ and not explicitly estimated during inference, these optimizations make LGA both theoretically sound and computationally efficient for practical applications.

## B.3 Computational Complexity and Efficiency Analysis

To provide a hardware-agnostic comparison of computational cost, we analyzed the theoretical Floating Point Operations (FLOPs) for the proposed LGA compared to the standard Scaled Dot-Product (SDP) attention. The analysis was conducted under the experimental setting used for the ETTm1 dataset ($B = 128, N = 64, H = 16, D = 8$).

**Hardware-Agnostic FLOPs Comparison.** We provide a granular breakdown of operations to quantify the overhead of LGA.

- **SDP:** The cost is dominated by linear projections ($8BNC^2$) and the attention score computation ($4BHN^2D$).
- **LGA:** The additional cost stems from the metric prediction network ($4BNHDd_G$) and the Mahalanobis score calculation ($6BHN^2D$). During training, an auxiliary cost for metric learning is incurred ($6N_s HN^2D$), where $N_s = 256$ is a fixed sample size.

Table 6 summarizes the total GigaFLOPs (G). Training FLOPs are approximated as $3\times$ Forward FLOPs for linear layers to account for backward passes.

Table 6: Theoretical FLOPs comparison between SDP and LGA.

| Module | Inference (G) | Training (G) |
|---|---|---|
| SDP | 9.748 | 28.890 |
| **LGA (Ours)** | 12.640 | 44.862 |

**Full Transformer Block Analysis.** While LGA incurs an overhead in the attention module specifically, its impact is significantly diluted when considering the full Transformer block (Attention + Feed-Forward Network), as the FFN dominates the computational budget. Table 7 presents the relative overhead.

Table 7: Relative overhead analysis of LGA compared to SDP.

| Setting | Module Scope | SDP | LGA (Ours) | Overhead |
|---|---|---|---|---|
| Inference | Attention Only | 9.748 G | 12.640 G | +29.7% |
| | **Full Block (Attn + FFN)** | **24.78 G** | **27.67 G** | **+11.7%** |
| Training | Attention Only | 28.890 G | 44.862 G | +55.3% |
| | **Full Block (Attn + FFN)** | **73.99 G** | **89.96 G** | **+21.6%** |

As shown, the effective overhead on the full model inference is only 11.7%. This confirms that LGA offers practical efficiency, providing robust geometry-aware modeling with a manageable computational cost.

## C EXPERIMENTAL SETTINGS

### C.1 DATASETS

We assessed our approach using six widely recognized time series forecasting datasets: Weather,[3] Electricity,[4] and ETT (Zhou et al., 2021) (ETTh1, ETTh2, ETTm1, ETTm2). These datasets were selected for their diverse periodic patterns and challenging real-world prediction characteristics. Their inherent variability and natural irregularities make them particularly suitable for evaluating robustness under our spike and level shift corruptions, as these datasets already contain patterns similar to those found in noisy real-world scenarios. While the traffic dataset is commonly used in benchmarks, we excluded it due to its extremely high dimensionality (862 features). Since PatchTST employs channel independence, different dimensions share manifold representations, making the high-dimensional nature of traffic inherently challenging for unified manifold learning approaches. Complete dataset specifications are provided in Table 8. All datasets are sourced from Wu et al. (2021).

Table 8: Details of 6 real-world datasets.

| Datasets | Features | Frequency | Samples | Domain |
|---|---|---|---|---|
| Weather | 21 | 10-min | 52,696 | Weather |
| Electricity | 321 | Hourly | 17,544 | Electricity |
| ETTh1 | 7 | Hourly | 17,420 | Temperature |
| ETTh2 | 7 | Hourly | 17,420 | Temperature |
| ETTm1 | 7 | 15-min | 69,680 | Temperature |
| ETTm2 | 7 | 15-min | 69,680 | Temperature |

### C.2 HYPERPARAMETER SETTINGS

All experiments are conducted with 2 Intel(R) Xeon(R) Gold 6226R CPUs @ 2.90GHz and 4 NVIDIA RTX 4090 24GB GPUs and 1 NVIDIA H100 80GB GPU. We conducted all experiments by following the original settings of PatchTST (Nie et al., 2023), as detailed in Table 9. In addition, we fixed the random seed to 2021 (Nie et al., 2023) to enhance experimental reproducibility. For most datasets, the input length was set to 512 to thoroughly evaluate how effectively our proposed LGA method embeds periodic patterns over extended temporal horizons. However, for the ETTh1 and ETTh2 datasets, we adopted an input length of 336, as this was reported to yield stable performance in the original PatchTST paper. For a comprehensive evaluation, we examined multiple forecasting horizons (96, 192, 336, and 720 time steps). The feed-forward network architecture follows standard practice, expanding the dimension to 256 before reduction, with GELU activation functions applied between linear transformations.

---

[3] https://www.bgc-jena.mpg.de/wetter/
[4] https://archive.ics.uci.edu/ml/datasets/ElectricityLoadDiagrams20112014

The synthetically generated query set, $\mathcal{S}_{\text{gen}}$, is created based on our empirical observation that the components of the real query vectors ($\mathcal{S}_{\text{real}}$) typically ranged between approximately -5 and 5. Therefore, we generated the vectors in $\mathcal{S}_{\text{gen}}$ by sampling each component from a uniform distribution, $\mathcal{U}(-5, 5)$, to ensure the network is exposed to a diverse yet relevant region of the representation space.

Regarding baseline models, for TimeMixer (Wang et al., 2024) experiments, we applied the publicly available hyperparameter settings that were originally optimized for input length 96, as these settings demonstrated sufficiently strong performance even when applied to our longer input length of 512. Similarly, for the experiments presented in Fig. 6, which examines performance across various input lengths, we maintained these consistent hyperparameter configurations throughout all comparisons. For CATS (Kim et al., 2024), we strictly adhered to the original hyperparameter settings as published in the original paper.

For the iTransformer (Liu et al., 2024) implementation, we noted that publicly available hyperparameters were optimized solely for a 96-length input. As our primary goal was to conduct a fair comparison between the SDP attention and our proposed LGA, we adhered to these original settings for all iTransformer experiments. This configuration consists of a 2-layer architecture with an embedding dimension of 128.

Table 9: Hyperparameter settings of PatchTST with an input sequence length of 512.

| Datasets | Layers | Embedding Size | # of Heads | Batch Size | $\sigma^2$ | Input Length |
|----------|--------|----------------|------------|------------|------------|--------------|
| Weather* | 3 | 128 | 16 | 128 | $10^{-2}$ | 512 |
| Electricity* | 3 | 128 | 16 | 32 | $10^{-2}$ | 512 |
| ETTh1 | 3 | 16 | 4 | 128 | $10^{-2}$ | 336 |
| ETTh2 | 3 | 16 | 4 | 128 | $10^{-2}$ | 336 |
| ETTm1 | 3 | 128 | 16 | 128 | $10^{-2}$ | 512 |
| ETTm2 | 3 | 128 | 16 | 128 | $10^{-2}$ | 512 |

* For weather and electricity, $\sigma^2 = 1$ when H=720.

## D  FURTHER EXPERIMENTS ON LGA

To supplement the primary findings presented in the main text, we conducted further experiments to validate the key design choices of our proposed methodology and to demonstrate its practical effectiveness on real-world data.

### D.1  STABILITY ANALYSIS AND DETAILED PERFORMANCE

In the main text, we demonstrated the superior robustness of LGA based on average performance metrics. Here, we provide a comprehensive breakdown of these results to validate the training stability and reliability of our proposed method. Table 10 presents the MSE and MAE across six benchmark datasets, reporting the mean and standard deviation computed over three independent runs with different random seeds.

A critical observation from these detailed results is the stability of the learning process. As indicated by the standard deviations (denoted by the $\pm$ values), PatchLGA consistently exhibits lower variance across different initialization seeds compared to the baselines, particularly under high severity levels (Severity 4-5). While standard attention mechanisms (PatchTST) often show increased performance fluctuation when exposed to severe noise, LGA maintains a tight confidence interval. This suggests that the local geometry prior acts as an effective regularizer, guiding the model towards robust convergence regardless of random initialization. Consequently, LGA not only improves forecasting accuracy but also ensures predictable and reliable deployment in real-world scenarios where data quality is inconsistent.

Table 10: Performance comparison of PatchLGA and PatchTST across six datasets under combined, level shift, and spike corruptions. Results are averaged across forecasting horizons {96, 192, 336, 720} and reported as Mean ± Std over 3 random seeds. This detailed breakdown highlights not only the superior accuracy (lower MSE/MAE) but also the training stability (lower Std) of LGA compared to the baseline.

**ETTm1**

| | PatchLGA | | | | | | PatchTST | | | | | |
|---|---|---|---|---|---|---|---|---|---|---|---|---|
| | Combined | | Level Shift | | Spike | | Combined | | Level Shift | | Spike | |
| Severity | MSE | MAE | MSE | MAE | MSE | MAE | MSE | MAE | MSE | MAE | MSE | MAE |
| 0 | 0.3531±0.0022 | 0.3815±0.0024 | 0.3531±0.0022 | 0.3815±0.0024 | 0.3531±0.0022 | 0.3815±0.0024 | 0.3535±0.0017 | 0.3834±0.0011 | 0.3535±0.0017 | 0.3834±0.0011 | 0.3535±0.0017 | 0.3834±0.0011 |
| 1 | 0.3601±0.0019 | 0.3877±0.0020 | 0.3598±0.0020 | 0.3869±0.0020 | 0.3538±0.0022 | 0.3823±0.0024 | 0.3607±0.0015 | 0.3901±0.0010 | 0.3605±0.0016 | 0.3897±0.0009 | 0.3540±0.0017 | 0.3839±0.0011 |
| 2 | 0.3737±0.0022 | 0.3985±0.0016 | 0.3718±0.0021 | 0.3971±0.0019 | 0.3547±0.0023 | 0.3828±0.0023 | 0.3762±0.0013 | 0.4026±0.0005 | 0.3749±0.0012 | 0.4018±0.0007 | 0.3547±0.0017 | 0.3848±0.0010 |
| 3 | 0.5141±0.0050 | 0.4683±0.0010 | 0.5022±0.0041 | 0.4614±0.0012 | 0.3627±0.0018 | 0.3892±0.0019 | 0.5969±0.0198 | 0.5006±0.0071 | 0.5897±0.0200 | 0.4952±0.0071 | 0.3655±0.0013 | 0.3927±0.0009 |
| 4 | 0.6110±0.0075 | 0.5261±0.0023 | 0.5960±0.0068 | 0.5160±0.0020 | 0.3678±0.0017 | 0.3950±0.0016 | 0.6807±0.0217 | 0.5532±0.0080 | 0.6763±0.0232 | 0.5484±0.0086 | 0.3706±0.0014 | 0.3978±0.0006 |
| 5 | 0.7265±0.0098 | 0.5768±0.0029 | 0.7122±0.0089 | 0.5677±0.0030 | 0.3723±0.0013 | 0.3982±0.0014 | 0.8232±0.0270 | 0.6084±0.0084 | 0.8312±0.0297 | 0.6073±0.0091 | 0.3761±0.0016 | 0.4021±0.0008 |

**ETTm2**

| | PatchLGA | | | | | | PatchTST | | | | | |
|---|---|---|---|---|---|---|---|---|---|---|---|---|
| | Combined | | Level Shift | | Spike | | Combined | | Level Shift | | Spike | |
| Severity | MSE | MAE | MSE | MAE | MSE | MAE | MSE | MAE | MSE | MAE | MSE | MAE |
| 0 | 0.2575±0.0002 | 0.3158±0.0002 | 0.2575±0.0002 | 0.3158±0.0002 | 0.2575±0.0002 | 0.3158±0.0002 | 0.2553±0.0007 | 0.3157±0.0009 | 0.2553±0.0007 | 0.3157±0.0009 | 0.2553±0.0007 | 0.3157±0.0009 |
| 1 | 0.2643±0.0004 | 0.3253±0.0002 | 0.2636±0.0006 | 0.3242±0.0003 | 0.2583±0.0003 | 0.3170±0.0003 | 0.2633±0.0011 | 0.3266±0.0010 | 0.2625±0.0009 | 0.3255±0.0011 | 0.2562±0.0008 | 0.3169±0.0008 |
| 2 | 0.2762±0.0002 | 0.3349±0.0005 | 0.2742±0.0001 | 0.3324±0.0003 | 0.2596±0.0001 | 0.3188±0.0004 | 0.2757±0.0009 | 0.3362±0.0013 | 0.2736±0.0008 | 0.3338±0.0012 | 0.2577±0.0010 | 0.3189±0.0010 |
| 3 | 0.3034±0.0004 | 0.3547±0.0003 | 0.2993±0.0006 | 0.3506±0.0003 | 0.2616±0.0001 | 0.3214±0.0004 | 0.3068±0.0014 | 0.3584±0.0015 | 0.3029±0.0009 | 0.3546±0.0013 | 0.2601±0.0009 | 0.3216±0.0013 |
| 4 | 0.3567±0.0020 | 0.3950±0.0008 | 0.3513±0.0026 | 0.3903±0.0010 | 0.2640±0.0002 | 0.3247±0.0006 | 0.3598±0.0020 | 0.3987±0.0016 | 0.3563±0.0013 | 0.3949±0.0011 | 0.2622±0.0010 | 0.3246±0.0016 |
| 5 | 0.4233±0.0040 | 0.4286±0.0017 | 0.4155±0.0048 | 0.4225±0.0020 | 0.2672±0.0004 | 0.3279±0.0006 | 0.4294±0.0032 | 0.4324±0.0021 | 0.4246±0.0012 | 0.4275±0.0015 | 0.2659±0.0014 | 0.3281±0.0016 |

**Weather**

| | PatchLGA | | | | | | PatchTST | | | | | |
|---|---|---|---|---|---|---|---|---|---|---|---|---|
| | Combined | | Level Shift | | Spike | | Combined | | Level Shift | | Spike | |
| Severity | MSE | MAE | MSE | MAE | MSE | MAE | MSE | MAE | MSE | MAE | MSE | MAE |
| 0 | 0.2271±0.0004 | 0.2656±0.0005 | 0.2271±0.0004 | 0.2656±0.0005 | 0.2271±0.0004 | 0.2656±0.0005 | 0.2254±0.0004 | 0.2647±0.0006 | 0.2254±0.0004 | 0.2647±0.0006 | 0.2254±0.0004 | 0.2647±0.0006 |
| 1 | 0.2391±0.0012 | 0.2848±0.0009 | 0.2382±0.0012 | 0.2828±0.0011 | 0.2280±0.0004 | 0.2681±0.0005 | 0.2401±0.0009 | 0.2860±0.0018 | 0.2392±0.0012 | 0.2841±0.0017 | 0.2266±0.0001 | 0.2673±0.0005 |
| 2 | 0.2644±0.0012 | 0.3137±0.0014 | 0.2617±0.0011 | 0.3093±0.0014 | 0.2298±0.0006 | 0.2709±0.0007 | 0.2680±0.0022 | 0.3165±0.0025 | 0.2661±0.0025 | 0.3131±0.0027 | 0.2288±0.0004 | 0.2703±0.0007 |
| 3 | 0.3015±0.0031 | 0.3398±0.0021 | 0.2974±0.0033 | 0.3331±0.0021 | 0.2321±0.0006 | 0.2740±0.0009 | 0.3111±0.0049 | 0.3443±0.0042 | 0.3083±0.0055 | 0.3392±0.0045 | 0.2308±0.0001 | 0.2736±0.0007 |
| 4 | 0.3623±0.0036 | 0.3908±0.0021 | 0.3522±0.0034 | 0.3776±0.0021 | 0.2345±0.0004 | 0.2840±0.0007 | 0.3787±0.0104 | 0.3904±0.0057 | 0.3697±0.0120 | 0.3801±0.0071 | 0.2334±0.0008 | 0.2833±0.0015 |
| 5 | 0.4547±0.0038 | 0.4256±0.0026 | 0.4373±0.0034 | 0.4086±0.0026 | 0.2387±0.0007 | 0.2898±0.0009 | 0.5062±0.0174 | 0.4311±0.0075 | 0.4868±0.0191 | 0.4167±0.0091 | 0.2365±0.0009 | 0.2880±0.0015 |

**ETTh1**

| | PatchLGA | | | | | | PatchTST | | | | | |
|---|---|---|---|---|---|---|---|---|---|---|---|---|
| | Combined | | Level Shift | | Spike | | Combined | | Level Shift | | Spike | |
| Severity | MSE | MAE | MSE | MAE | MSE | MAE | MSE | MAE | MSE | MAE | MSE | MAE |
| 0 | 0.4158±0.0010 | 0.4293±0.0009 | 0.4158±0.0010 | 0.4293±0.0009 | 0.4158±0.0010 | 0.4293±0.0009 | 0.4137±0.0027 | 0.4279±0.0020 | 0.4137±0.0027 | 0.4279±0.0020 | 0.4137±0.0027 | 0.4279±0.0020 |
| 1 | 0.4168±0.0009 | 0.4307±0.0008 | 0.4166±0.0010 | 0.4305±0.0009 | 0.4161±0.0009 | 0.4296±0.0008 | 0.4145±0.0026 | 0.4294±0.0021 | 0.4145±0.0026 | 0.4290±0.0020 | 0.4138±0.0026 | 0.4281±0.0022 |
| 2 | 0.4215±0.0011 | 0.4358±0.0009 | 0.4209±0.0010 | 0.4347±0.0008 | 0.4160±0.0011 | 0.4302±0.0010 | 0.4193±0.0030 | 0.4348±0.0020 | 0.4189±0.0028 | 0.4333±0.0020 | 0.4141±0.0027 | 0.4291±0.0022 |
| 3 | 0.4732±0.0034 | 0.4667±0.0015 | 0.4682±0.0032 | 0.4622±0.0014 | 0.4202±0.0010 | 0.4336±0.0009 | 0.4740±0.0049 | 0.4668±0.0029 | 0.4690±0.0049 | 0.4621±0.0027 | 0.4187±0.0032 | 0.4326±0.0022 |
| 4 | 0.5508±0.0052 | 0.5119±0.0020 | 0.5352±0.0054 | 0.5018±0.0023 | 0.4298±0.0011 | 0.4407±0.0010 | 0.5503±0.0048 | 0.5122±0.0027 | 0.5352±0.0044 | 0.5024±0.0030 | 0.4295±0.0039 | 0.4402±0.0024 |
| 5 | 0.7185±0.0115 | 0.5826±0.0031 | 0.6878±0.0114 | 0.5658±0.0033 | 0.4424±0.0013 | 0.4499±0.0012 | 0.7239±0.0073 | 0.5857±0.0041 | 0.6982±0.0075 | 0.5705±0.0050 | 0.4438±0.0050 | 0.4503±0.0026 |

**ETTh2**

| | PatchLGA | | | | | | PatchTST | | | | | |
|---|---|---|---|---|---|---|---|---|---|---|---|---|
| | Combined | | Level Shift | | Spike | | Combined | | Level Shift | | Spike | |
| Severity | MSE | MAE | MSE | MAE | MSE | MAE | MSE | MAE | MSE | MAE | MSE | MAE |
| 0 | 0.3395±0.0023 | 0.3860±0.0022 | 0.3395±0.0023 | 0.3860±0.0022 | 0.3395±0.0023 | 0.3860±0.0022 | 0.3370±0.0074 | 0.3837±0.0036 | 0.3370±0.0074 | 0.3837±0.0036 | 0.3370±0.0074 | 0.3837±0.0036 |
| 1 | 0.3403±0.0024 | 0.3887±0.0022 | 0.3400±0.0025 | 0.3877±0.0025 | 0.3398±0.0024 | 0.3870±0.0022 | 0.3377±0.0069 | 0.3856±0.0033 | 0.3373±0.0070 | 0.3847±0.0034 | 0.3374±0.0073 | 0.3845±0.0036 |
| 2 | 0.3485±0.0038 | 0.3987±0.0037 | 0.3490±0.0034 | 0.3977±0.0032 | 0.3392±0.0028 | 0.3879±0.0029 | 0.3436±0.0058 | 0.3938±0.0026 | 0.3446±0.0066 | 0.3937±0.0029 | 0.3361±0.0065 | 0.3844±0.0033 |
| 3 | 0.3640±0.0063 | 0.4108±0.0050 | 0.3636±0.0054 | 0.4084±0.0044 | 0.3408±0.0035 | 0.3902±0.0035 | 0.3568±0.0054 | 0.4040±0.0026 | 0.3574±0.0065 | 0.4032±0.0031 | 0.3372±0.0062 | 0.3862±0.0031 |
| 4 | 0.3877±0.0055 | 0.4276±0.0039 | 0.3820±0.0051 | 0.4230±0.0035 | 0.3462±0.0040 | 0.3938±0.0034 | 0.3854±0.0047 | 0.4240±0.0026 | 0.3803±0.0059 | 0.4196±0.0031 | 0.3425±0.0065 | 0.3901±0.0032 |
| 5 | 0.4143±0.0092 | 0.4413±0.0054 | 0.4083±0.0086 | 0.4363±0.0051 | 0.3476±0.0048 | 0.3957±0.0040 | 0.4203±0.0058 | 0.4423±0.0033 | 0.4142±0.0072 | 0.4370±0.0035 | 0.3429±0.0061 | 0.3912±0.0027 |

**Electricity**

| | PatchLGA | | | | | | PatchTST | | | | | |
|---|---|---|---|---|---|---|---|---|---|---|---|---|
| | Combined | | Level Shift | | Spike | | Combined | | Level Shift | | Spike | |
| Severity | MSE | MAE | MSE | MAE | MSE | MAE | MSE | MAE | MSE | MAE | MSE | MAE |
| 0 | 0.1614±0.0007 | 0.2559±0.0006 | 0.1614±0.0007 | 0.2559±0.0006 | 0.1614±0.0007 | 0.2559±0.0006 | 0.1600±0.0003 | 0.2541±0.0004 | 0.1600±0.0003 | 0.2541±0.0004 | 0.1600±0.0003 | 0.2541±0.0004 |
| 1 | 0.1686±0.0007 | 0.2645±0.0007 | 0.1677±0.0006 | 0.2632±0.0009 | 0.1623±0.0007 | 0.2574±0.0006 | 0.1691±0.0003 | 0.2648±0.0004 | 0.1681±0.0003 | 0.2634±0.0004 | 0.1613±0.0004 | 0.2560±0.0003 |
| 2 | 0.1795±0.0007 | 0.2765±0.0007 | 0.1766±0.0005 | 0.2728±0.0009 | 0.1641±0.0009 | 0.2598±0.0005 | 0.1828±0.0010 | 0.2788±0.0008 | 0.1803±0.0012 | 0.2752±0.0010 | 0.1633±0.0004 | 0.2589±0.0003 |
| 3 | 0.1928±0.0010 | 0.2879±0.0010 | 0.1887±0.0008 | 0.2828±0.0010 | 0.1654±0.0006 | 0.2613±0.0006 | 0.1993±0.0010 | 0.2923±0.0009 | 0.1963±0.0017 | 0.2873±0.0010 | 0.1651±0.0004 | 0.2610±0.0005 |
| 4 | 0.2252±0.0015 | 0.3166±0.0014 | 0.2164±0.0013 | 0.3072±0.0016 | 0.1693±0.0007 | 0.2662±0.0005 | 0.2321±0.0010 | 0.3212±0.0004 | 0.2272±0.0025 | 0.3132±0.0004 | 0.1693±0.0003 | 0.2664±0.0005 |
| 5 | 0.2817±0.0043 | 0.3504±0.0028 | 0.2698±0.0044 | 0.3388±0.0031 | 0.1731±0.0005 | 0.2704±0.0005 | 0.2929±0.0032 | 0.3579±0.0005 | 0.2877±0.0055 | 0.3482±0.0004 | 0.1746±0.0001 | 0.2721±0.0004 |

Table 11: Detailed performance comparison of Samformer Ilbert et al. (2024) and PatchSamformer variants on the ETTm1 dataset with input length 512 under combined corruptions. PatchSamformer denotes the application of patch embedding to the Samformer architecture. While patching induces significant vulnerability to noise in the linear Samformer model (SDP), LGA successfully stabilizes the architecture, achieving the lowest error rates across varying severity levels.

| ETTm1 | H | | | 96 | | | | | 192 | | | | | 336 | | | | | 720 | | |
|---|---|---|---|---|---|---|---|---|---|---|---|---|---|---|---|---|---|---|---|---|---|
| Mod. | Atten. | Metric | 1 | 2 | 3 | 4 | 5 | 1 | 2 | 3 | 4 | 5 | 1 | 2 | 3 | 4 | 5 | 1 | 2 | 3 | 4 | 5 |
| Samformer | LGA | MSE | 0.319 | 0.336 | 0.510 | 0.659 | 0.802 | 0.354 | 0.369 | 0.522 | 0.655 | 0.767 | 0.378 | 0.393 | 0.533 | 0.653 | 0.755 | 0.429 | 0.442 | 0.564 | 0.675 | 0.755 |
| | | MAE | 0.363 | 0.378 | 0.456 | 0.532 | 0.593 | 0.382 | 0.394 | 0.469 | 0.539 | 0.591 | 0.393 | 0.405 | 0.477 | 0.543 | 0.592 | 0.422 | 0.432 | 0.499 | 0.560 | 0.600 |
| | SDP | MSE | 0.327 | 0.348 | 0.534 | 0.680 | 0.860 | 0.357 | 0.373 | 0.527 | 0.660 | 0.796 | 0.383 | 0.399 | 0.540 | 0.648 | 0.776 | 0.433 | 0.448 | 0.589 | 0.700 | 0.804 |
| | | MAE | 0.366 | 0.382 | 0.470 | 0.546 | 0.619 | 0.383 | 0.396 | 0.474 | 0.544 | 0.602 | 0.397 | 0.410 | 0.482 | 0.543 | 0.598 | 0.426 | 0.437 | 0.510 | 0.569 | 0.617 |
| Patch Samformer | LGA | MSE | 0.332 | 0.350 | 0.459 | 0.577 | 0.715 | 0.363 | 0.376 | 0.469 | 0.586 | 0.673 | 0.380 | 0.392 | 0.489 | 0.576 | 0.656 | 0.433 | 0.443 | 0.533 | 0.613 | 0.685 |
| | | MAE | 0.367 | 0.381 | 0.437 | 0.500 | 0.561 | 0.383 | 0.394 | 0.448 | 0.513 | 0.557 | 0.396 | 0.405 | 0.455 | 0.505 | 0.547 | 0.424 | 0.431 | 0.482 | 0.529 | 0.566 |
| | SDP | MSE | 0.373 | 0.410 | 0.852 | 1.171 | 1.465 | 0.371 | 0.401 | 0.779 | 0.978 | 1.240 | 0.403 | 0.432 | 0.931 | 1.154 | 1.444 | 0.438 | 0.459 | 0.786 | 0.952 | 1.100 |
| | | MAE | 0.407 | 0.430 | 0.599 | 0.708 | 0.783 | 0.408 | 0.427 | 0.580 | 0.656 | 0.723 | 0.423 | 0.441 | 0.628 | 0.711 | 0.776 | 0.436 | 0.450 | 0.586 | 0.654 | 0.688 |

## D.2 ADDITIONAL EXPERIMENTS ON OTHER MODELS

To demonstrate the broad applicability of our method beyond standard Transformer architectures, we extended our evaluation to Samformer (Ilbert et al., 2024), a state-of-the-art model distinguished by its linear complexity and channel-wise attention mechanism. In this experiment, we analyzed four model variants on the ETTm1 dataset: the original Samformer, Samformer with LGA, and their respective "Patch" versions where the input time series is processed via patch embedding. The detailed results across all forecasting horizons and corruption severity levels are summarized in Table 11.

The experimental results highlight a critical interaction between the model's structural bias and the input representation. A notable finding is the severe vulnerability introduced by patching in linear architectures. As observed in the "PatchSamformer + SDP" results, applying patch embedding to Samformer leads to catastrophic performance degradation under realistic corruptions. For instance, at the shortest horizon ($H = 96$) with Severity 5, the MSE soars to 1.465. We attribute this instability to the architectural design of Samformer; unlike conventional Transformers, it does not incorporate a non-linear Feed-Forward Network (FFN) after the attention block. Consequently, the high-dimensional linear projections inherent to patching appear to amplify input noise without the filtering or buffering effects typically provided by non-linear layers, making the model hypersensitive to corruptions.

However, replacing the standard attention with LGA in this vulnerable architecture completely reverses the degradation. The "PatchSamformer + LGA" variant not only recovers from the failure observed in the SDP counterpart but also achieves the highest robustness among all compared models. For example, at the longest forecasting horizon ($H = 720$) under maximum corruption (Severity 5), LGA reduces the MSE from 1.100 (SDP) to 0.685, significantly outperforming even the original Samformer baseline (0.804). This empirical evidence suggests that the local geometry prior of LGA acts as a potent regularizer, effectively stabilizing the attention mechanism and preventing overfitting to noise, even in high-dimensional feature spaces that lack non-linear protection. These findings confirm that LGA is a versatile and critical component for enhancing robustness across diverse architectural paradigms.

### D.3 EVALUATION ON SYNTHETIC BENCHMARKS

To rigorously validate the robustness of PatchLGA under theoretically controlled conditions, we conducted additional experiments utilizing the synthetic benchmark framework proposed by Janßen et al. Janßen et al. (2025). Unlike TSRBench, which injects corruptions into real-world data, this framework generates purely synthetic multivariate time series with parameterizable frequency and noise characteristics.

#### D.3.1 EXPERIMENTAL SETUP

Following the protocol in Janßen et al. (2025), we generated synthetic datasets across 7 distinct frequency bands, ranging from 'Very Low' to 'Very High'. For each frequency band, we constructed a multivariate dataset by combining three signal types (Sine, Smooth Square, and Smooth Sawtooth) with five distinct noise types (White, Brownian, Impulse, Trend-dependent, and Seasonal-dependent) under four SNR levels.

Consistent with our main experiments, the models were trained with an input sequence length of 512. We evaluated the forecasting performance across four prediction horizons $H \in \{96, 192, 336, 720\}$. This setup allows us to assess the model's capability to recover intrinsic signal geometry from noisy inputs across varying temporal scales.

#### D.3.2 RESULTS

Table 12 presents the detailed performance comparison between PatchLGA and the PatchTST baseline.

Table 12: MSE comparison of PatchLGA and PatchTST on the synthetic benchmark by Janßen et al. (2025). Models were trained with an input length of 512 and evaluated across four prediction horizons $H \in \{96, 192, 336, 720\}$. PatchLGA demonstrates superior accuracy (lower MSE) compared to the baseline across most frequency bands.

| Freq. | | Very High | | High | | Mid-High | | Mid | | Low-Mid | | Low | | Very Low | |
|---|---|---|---|---|---|---|---|---|---|---|---|---|---|---|---|
| Attn Type | | SDP | LGA | SDP | LGA | SDP | LGA | SDP | LGA | SDP | LGA | SDP | LGA | SDP | LGA |
| Horizon | 96 | 0.070 | **0.070** | 0.106 | **0.102** | 0.114 | **0.111** | 0.098 | **0.098** | 0.091 | **0.090** | 0.082 | **0.080** | 0.143 | **0.138** |
| | 192 | 0.070 | **0.070** | 0.105 | **0.101** | 0.114 | **0.111** | 0.098 | **0.098** | 0.091 | **0.090** | 0.082 | **0.080** | 0.143 | **0.138** |
| | 336 | 0.068 | **0.068** | 0.101 | **0.097** | 0.111 | **0.108** | 0.089 | **0.089** | 0.083 | **0.083** | 0.085 | **0.083** | 0.140 | **0.136** |
| | 720 | 0.062 | **0.062** | 0.090 | **0.086** | 0.108 | **0.105** | 0.072 | **0.072** | 0.066 | **0.066** | 0.097 | **0.095** | 0.128 | **0.124** |

The results indicate that PatchLGA consistently outperforms or matches PatchTST across different frequency characteristics and forecasting horizons. Notably, (Janßen et al., 2025) highlight that models typically degrade in the 'Very Low' frequency band because the lookback window may not capture complete periodic cycles. In this challenging regime, PatchLGA achieves a distinct improvement (e.g., reducing MSE from 0.143 to 0.138 at $H = 96$), suggesting that LGA's manifold learning capability effectively captures intrinsic geometric structures even when temporal periodicity is locally ambiguous. Furthermore, LGA maintains superiority in 'High' frequency bands, demonstrating its versatility in handling rapid fluctuations contaminated by complex noise.

Combining these findings with our TSRBench results, we conclude that LGA offers robust performance in both real-world data with injected anomalies (TSRBench) and theoretically controlled synthetic environments (Janßen et al. (2025)), validating its generalizability.

### D.4 IMPACT OF QUERY SET SELECTION ON PERFORMANCE

We conduct an ablation study to validate our choice of using a combined query set ($\mathcal{S}_{both}$) for training the metric prediction network, as detailed in Section 3.4. As shown in Table 13, using the combined set of real and synthetic queries consistently outperforms using either set alone, especially under severe corruption. This result confirms that a broader query coverage is crucial for robustly learning the local geometry of the data manifold.

Table 13: Ablation study on the query set selection for ETTm1, showing average MSE and MAE across all forecasting horizons. Lower values are better, with the best results highlighted in **bold**.

| Severity | 1 | | | 2 | | | 3 | | | 4 | | | 5 | | |
|---|---|---|---|---|---|---|---|---|---|---|---|---|---|---|---|
| Query Set | $\mathcal{S}_{both}$ | $\mathcal{S}_{gen}$ | $\mathcal{S}_{real}$ | $\mathcal{S}_{both}$ | $\mathcal{S}_{gen}$ | $\mathcal{S}_{real}$ | $\mathcal{S}_{both}$ | $\mathcal{S}_{gen}$ | $\mathcal{S}_{real}$ | $\mathcal{S}_{both}$ | $\mathcal{S}_{gen}$ | $\mathcal{S}_{real}$ | $\mathcal{S}_{both}$ | $\mathcal{S}_{gen}$ | $\mathcal{S}_{real}$ |
| MSE | 0.359 | **0.358** | 0.360 | 0.372 | **0.371** | 0.374 | **0.519** | 0.524 | 0.525 | **0.617** | 0.630 | 0.626 | **0.734** | 0.752 | 0.744 |
| MAE | **0.385** | 0.387 | 0.386 | **0.397** | 0.398 | 0.398 | **0.468** | 0.473 | 0.471 | **0.526** | 0.534 | 0.531 | **0.577** | 0.586 | 0.582 |

### D.5 VALIDATION ON REAL-WORLD ANOMALOUS DATA AND THE NECESSITY OF A PRINCIPLED BENCHMARK

To complement the results from our TSRBench, we conducted an additional experiment on a dataset containing genuine, unscripted anomalies. This validation aims to demonstrate that the robustness of LGA extends beyond controlled, synthetic corruptions to the unpredictable nature of real-world operational data.

For this purpose, we used the AWSCloudWatch dataset from the widely-recognized Numenta Anomaly Benchmark (NAB) (Lavin & Ahmad, 2015). This univariate time series represents AWS EC2 server CPU utilization and includes labeled periods of naturally occurring anomalous behavior. For the experiment, we used the same hyperparameter settings as the ETTm1 experiments, with an input sequence length of 1024 and a forecasting horizon of 96.

Table 14: Performance on the AWS CloudWatch dataset. The lower MAE indicates superior performance on real-world anomalous data.

| Model | MAE |
|---|---|
| PatchTST | 7.97 |
| **PatchLGA (Ours)** | **6.22** |

The results in Table 14 show that PatchLGA significantly outperforms the baseline, achieving a 22% reduction in MAE. This provides strong empirical evidence that local geometry-aware approach of LGA is highly effective in mitigating the impact of genuine, real-world anomalies.

However, it is important to highlight that conducting such validation for *forecasting robustness* is often infeasible with most publicly available datasets. A fair and rigorous evaluation requires several critical conditions to be met:

- **Availability of precise anomaly labels:** To verify that a model is robustly forecasting the "normal" underlying pattern, the evaluation metric (e.g., MSE, MAE) must be calculated on a ground truth that excludes the anomalous periods. Most forecasting benchmarks do not provide such granular labels.
- **No time lag in anomaly labels:** Even when labels are available, a time lag between the actual anomaly occurrence and its timestamp in the data makes it impossible to accurately identify which input segments are corrupted and which future segments should be excluded from evaluation. This limitation renders most anomaly detection datasets unsuitable for this specific validation purpose.

These strict requirements underscore the challenge of using "in-the-wild" datasets for reproducible robustness research. In such scenarios, our TSRBench becomes particularly useful as it enables a controlled and systematic evaluation of robustness. By providing clean training data and systematically corrupted test sets with known corruption boundaries, TSRBench allows for a comprehensive robustness assessment that would be difficult, if not impossible, to achieve with purely real-world datasets where the nature, extent, and precise timing of corruptions are unknown and uncontrolled.

# E    ADDITIONAL EXPERIMENTAL RESULTS AND ANALYSIS

In Section 5, we provided averaged MSE results across forecasting horizons to demonstrate the robustness of PatchLGA compared to baseline models. This section presents the complete experimental results across all datasets, forecasting horizons, and corruption scenarios. These detailed results not only validate our main findings but also provide deeper insights into how different architectures respond to various levels and types of corruptions under specific forecasting conditions.

## E.1    DETAILED PERFORMANCE ANALYSIS OF LONG-TERM FORECASTING UNDER REALISTIC CORRUPTIONS

We provide the full results of Section 5.1 for each dataset (ETTm1, ETTm2, Weather, ETTh1, ETTh2, and Electricity) across all forecasting horizons in Table 15, and 16. These detailed results expand on the average performance presented in Table 10.

The complete dataset results demonstrate several important patterns. While PatchLGA generally outperforms baselines across most corruption types, the performance differences vary by corruption type. For level shifts, PatchLGA consistently shows superior robustness across most datasets and forecasting horizons. For spike corruptions, all models demonstrate relatively less performance degradation compared to other corruption types, with both PatchLGA and PatchTST maintaining reasonable robustness. However, the most significant finding appears in combined corruption scenarios (both spikes and level shifts occurring simultaneously). In these realistic cases, PatchTST and TimeMixer often exhibit performance degradation considerably exceeding what would be expected from the individual corruption types alone, suggesting a compounding effect. PatchLGA, in contrast, maintains more consistent performance even under these challenging combined corruptions.

## E.2    EXTENDED EVALUATION OF ATTENTION MECHANISM ROBUSTNESS

In this section, we provide a more detailed analysis of the effectiveness of LGA across different attention mechanisms. Table 17 presents the complete performance results across all forecasting horizons and severity levels for three representative attention mechanisms: self-temporal attention (PatchTST), cross-temporal attention (CATS), and self-channel attention (iTransformer).

The detailed results in Table 17 confirm and extend our primary findings. A consistent trend across all architectures is that the performance gains from LGA become increasingly pronounced as corruption severity increases. For lower severity levels (1-2), the improvements are often modest, but as corruptions intensify (levels 3-5), the ability of LGA to adapt to the local data geometry provides a clear and substantial advantage. The analysis reveals distinct patterns based on the attention type:

- **Self-Temporal Attention (PatchTST):** This mechanism consistently demonstrates the greatest performance improvements when enhanced with LGA. The benefits are particularly significant for longer forecasting horizons under severe corruption. For example, at severity

Table 15: Full experimental result on ETTm1, ETTm2, weather with 512 input lengths.

| ETTm1 | | Combined | | | | | | Level Shift | | | | | | Spike | | | | | |
|---|---|---|---|---|---|---|---|---|---|---|---|---|---|---|---|---|---|---|---|
| Model | | PatchLGA | | PatchTST | | TimeMixer | | PatchLGA | | PatchTST | | TimeMixer | | PatchLGA | | PatchTST | | TimeMixer | |
| H | Metric | MSE | MAE | MSE | MAE | MSE | MAE | MSE | MAE | MSE | MAE | MSE | MAE | MSE | MAE | MSE | MAE | MSE | MAE |
| 96 | 1 | 0.297 | 0.353 | 0.304 | 0.357 | 0.307 | 0.363 | 0.297 | 0.352 | 0.304 | 0.356 | 0.306 | 0.363 | 0.289 | 0.345 | 0.293 | 0.346 | 0.300 | 0.357 |
| 96 | 2 | 0.314 | 0.368 | 0.326 | 0.377 | 0.335 | 0.383 | 0.312 | 0.366 | 0.325 | 0.376 | 0.332 | 0.380 | 0.289 | 0.346 | 0.294 | 0.348 | 0.303 | 0.359 |
| 96 | 3 | 0.527 | 0.456 | 0.625 | 0.497 | 0.619 | 0.490 | 0.511 | 0.447 | 0.616 | 0.490 | 0.602 | 0.481 | 0.302 | 0.354 | 0.310 | 0.361 | 0.314 | 0.368 |
| 96 | 4 | 0.657 | 0.530 | 0.743 | 0.566 | 0.769 | 0.558 | 0.641 | 0.519 | 0.746 | 0.563 | 0.742 | 0.544 | 0.309 | 0.363 | 0.318 | 0.369 | 0.323 | 0.375 |
| 96 | 5 | 0.820 | 0.594 | 0.944 | 0.636 | 0.953 | 0.629 | 0.802 | 0.584 | 0.954 | 0.634 | 0.924 | 0.615 | 0.316 | 0.368 | 0.326 | 0.375 | 0.329 | 0.379 |
| 192 | 1 | 0.341 | 0.376 | 0.342 | 0.378 | 0.342 | 0.380 | 0.340 | 0.376 | 0.341 | 0.378 | 0.341 | 0.380 | 0.333 | 0.370 | 0.335 | 0.372 | 0.337 | 0.376 |
| 192 | 2 | 0.354 | 0.388 | 0.357 | 0.391 | 0.358 | 0.393 | 0.351 | 0.386 | 0.355 | 0.390 | 0.356 | 0.391 | 0.334 | 0.371 | 0.336 | 0.373 | 0.338 | 0.377 |
| 192 | 3 | 0.504 | 0.462 | 0.581 | 0.491 | 0.537 | 0.474 | 0.491 | 0.455 | 0.574 | 0.486 | 0.525 | 0.467 | 0.343 | 0.378 | 0.346 | 0.380 | 0.346 | 0.384 |
| 192 | 4 | 0.603 | 0.521 | 0.661 | 0.542 | 0.643 | 0.527 | 0.587 | 0.511 | 0.657 | 0.538 | 0.625 | 0.517 | 0.348 | 0.384 | 0.349 | 0.385 | 0.351 | 0.390 |
| 192 | 5 | 0.722 | 0.573 | 0.812 | 0.601 | 0.740 | 0.574 | 0.707 | 0.564 | 0.823 | 0.601 | 0.725 | 0.565 | 0.353 | 0.387 | 0.354 | 0.389 | 0.355 | 0.392 |
| 336 | 1 | 0.372 | 0.392 | 0.370 | 0.397 | 0.381 | 0.407 | 0.372 | 0.391 | 0.370 | 0.397 | 0.381 | 0.407 | 0.366 | 0.387 | 0.364 | 0.392 | 0.376 | 0.403 |
| 336 | 2 | 0.385 | 0.402 | 0.383 | 0.408 | 0.396 | 0.418 | 0.383 | 0.400 | 0.382 | 0.407 | 0.395 | 0.417 | 0.368 | 0.388 | 0.365 | 0.393 | 0.377 | 0.403 |
| 336 | 3 | 0.502 | 0.465 | 0.630 | 0.519 | 0.600 | 0.511 | 0.490 | 0.458 | 0.624 | 0.514 | 0.590 | 0.506 | 0.375 | 0.394 | 0.375 | 0.400 | 0.383 | 0.409 |
| 336 | 4 | 0.586 | 0.516 | 0.692 | 0.563 | 0.709 | 0.567 | 0.570 | 0.506 | 0.688 | 0.559 | 0.692 | 0.558 | 0.379 | 0.399 | 0.379 | 0.405 | 0.390 | 0.415 |
| 336 | 5 | 0.686 | 0.564 | 0.818 | 0.611 | 0.829 | 0.622 | 0.670 | 0.554 | 0.831 | 0.612 | 0.814 | 0.614 | 0.383 | 0.402 | 0.382 | 0.408 | 0.395 | 0.418 |
| 720 | 1 | 0.425 | 0.421 | 0.420 | 0.424 | 0.436 | 0.432 | 0.425 | 0.420 | 0.420 | 0.424 | 0.435 | 0.431 | 0.420 | 0.417 | 0.416 | 0.421 | 0.430 | 0.427 |
| 720 | 2 | 0.436 | 0.429 | 0.433 | 0.432 | 0.449 | 0.441 | 0.435 | 0.428 | 0.432 | 0.431 | 0.447 | 0.439 | 0.421 | 0.417 | 0.416 | 0.421 | 0.431 | 0.428 |
| 720 | 3 | 0.543 | 0.488 | 0.622 | 0.519 | 0.620 | 0.524 | 0.533 | 0.482 | 0.616 | 0.514 | 0.605 | 0.515 | 0.427 | 0.423 | 0.425 | 0.426 | 0.442 | 0.436 |
| 720 | 4 | 0.623 | 0.536 | 0.684 | 0.562 | 0.745 | 0.587 | 0.609 | 0.526 | 0.673 | 0.554 | 0.722 | 0.575 | 0.431 | 0.427 | 0.430 | 0.430 | 0.447 | 0.442 |
| 720 | 5 | 0.709 | 0.578 | 0.782 | 0.604 | 0.828 | 0.626 | 0.694 | 0.568 | 0.785 | 0.601 | 0.806 | 0.614 | 0.435 | 0.430 | 0.435 | 0.433 | 0.452 | 0.444 |

| ETTm2 | | Combined | | | | | | Level Shift | | | | | | Spike | | | | | |
|---|---|---|---|---|---|---|---|---|---|---|---|---|---|---|---|---|---|---|---|
| Model | | PatchLGA | | PatchTST | | TimeMixer | | PatchLGA | | PatchTST | | TimeMixer | | PatchLGA | | PatchTST | | TimeMixer | |
| H | Metric | MSE | MAE | MSE | MAE | MSE | MAE | MSE | MAE | MSE | MAE | MSE | MAE | MSE | MAE | MSE | MAE | MSE | MAE |
| 96 | 1 | 0.177 | 0.269 | 0.178 | 0.270 | 0.181 | 0.273 | 0.176 | 0.267 | 0.177 | 0.269 | 0.180 | 0.272 | 0.168 | 0.258 | 0.167 | 0.257 | 0.173 | 0.263 |
| 96 | 2 | 0.194 | 0.283 | 0.193 | 0.283 | 0.196 | 0.286 | 0.191 | 0.280 | 0.191 | 0.281 | 0.194 | 0.283 | 0.169 | 0.260 | 0.169 | 0.260 | 0.174 | 0.265 |
| 96 | 3 | 0.232 | 0.308 | 0.236 | 0.311 | 0.234 | 0.313 | 0.228 | 0.304 | 0.233 | 0.307 | 0.229 | 0.307 | 0.171 | 0.263 | 0.171 | 0.263 | 0.176 | 0.269 |
| 96 | 4 | 0.302 | 0.358 | 0.304 | 0.359 | 0.305 | 0.362 | 0.297 | 0.353 | 0.302 | 0.356 | 0.296 | 0.355 | 0.175 | 0.267 | 0.175 | 0.267 | 0.180 | 0.273 |
| 96 | 5 | 0.396 | 0.399 | 0.400 | 0.401 | 0.409 | 0.410 | 0.388 | 0.393 | 0.398 | 0.396 | 0.394 | 0.400 | 0.179 | 0.270 | 0.180 | 0.270 | 0.185 | 0.277 |
| 192 | 1 | 0.231 | 0.305 | 0.232 | 0.308 | 0.234 | 0.309 | 0.230 | 0.304 | 0.231 | 0.307 | 0.233 | 0.309 | 0.224 | 0.296 | 0.224 | 0.297 | 0.228 | 0.301 |
| 192 | 2 | 0.244 | 0.316 | 0.246 | 0.319 | 0.248 | 0.321 | 0.242 | 0.313 | 0.244 | 0.316 | 0.247 | 0.318 | 0.226 | 0.298 | 0.226 | 0.300 | 0.229 | 0.302 |
| 192 | 3 | 0.273 | 0.337 | 0.280 | 0.343 | 0.281 | 0.346 | 0.269 | 0.332 | 0.276 | 0.339 | 0.278 | 0.342 | 0.228 | 0.301 | 0.228 | 0.303 | 0.231 | 0.305 |
| 192 | 4 | 0.327 | 0.379 | 0.336 | 0.386 | 0.344 | 0.393 | 0.321 | 0.374 | 0.332 | 0.382 | 0.339 | 0.389 | 0.230 | 0.304 | 0.230 | 0.306 | 0.233 | 0.307 |
| 192 | 5 | 0.395 | 0.414 | 0.411 | 0.422 | 0.437 | 0.438 | 0.387 | 0.407 | 0.406 | 0.417 | 0.427 | 0.432 | 0.234 | 0.308 | 0.234 | 0.310 | 0.236 | 0.311 |
| 336 | 1 | 0.283 | 0.338 | 0.280 | 0.339 | 0.287 | 0.347 | 0.282 | 0.338 | 0.279 | 0.338 | 0.287 | 0.346 | 0.279 | 0.331 | 0.275 | 0.331 | 0.284 | 0.339 |
| 336 | 2 | 0.294 | 0.347 | 0.292 | 0.348 | 0.302 | 0.357 | 0.292 | 0.344 | 0.289 | 0.346 | 0.300 | 0.355 | 0.280 | 0.333 | 0.276 | 0.332 | 0.285 | 0.340 |
| 336 | 3 | 0.317 | 0.364 | 0.318 | 0.368 | 0.335 | 0.382 | 0.312 | 0.360 | 0.313 | 0.364 | 0.332 | 0.379 | 0.282 | 0.335 | 0.279 | 0.335 | 0.286 | 0.343 |
| 336 | 4 | 0.363 | 0.402 | 0.364 | 0.406 | 0.409 | 0.432 | 0.358 | 0.397 | 0.359 | 0.401 | 0.408 | 0.431 | 0.284 | 0.339 | 0.280 | 0.338 | 0.286 | 0.343 |
| 336 | 5 | 0.419 | 0.432 | 0.421 | 0.435 | 0.505 | 0.476 | 0.411 | 0.426 | 0.414 | 0.430 | 0.502 | 0.474 | 0.287 | 0.342 | 0.284 | 0.342 | 0.287 | 0.346 |
| 720 | 1 | 0.366 | 0.390 | 0.368 | 0.394 | 0.360 | 0.393 | 0.365 | 0.389 | 0.367 | 0.393 | 0.359 | 0.392 | 0.362 | 0.384 | 0.362 | 0.386 | 0.356 | 0.387 |
| 720 | 2 | 0.374 | 0.396 | 0.376 | 0.400 | 0.370 | 0.400 | 0.372 | 0.394 | 0.374 | 0.398 | 0.368 | 0.398 | 0.363 | 0.386 | 0.364 | 0.388 | 0.357 | 0.388 |
| 720 | 3 | 0.392 | 0.411 | 0.399 | 0.418 | 0.391 | 0.415 | 0.388 | 0.407 | 0.394 | 0.414 | 0.387 | 0.412 | 0.365 | 0.388 | 0.366 | 0.391 | 0.360 | 0.391 |
| 720 | 4 | 0.431 | 0.441 | 0.441 | 0.450 | 0.435 | 0.447 | 0.425 | 0.436 | 0.435 | 0.445 | 0.432 | 0.444 | 0.367 | 0.391 | 0.368 | 0.394 | 0.359 | 0.392 |
| 720 | 5 | 0.475 | 0.466 | 0.491 | 0.477 | 0.485 | 0.475 | 0.466 | 0.460 | 0.482 | 0.471 | 0.480 | 0.470 | 0.370 | 0.394 | 0.372 | 0.397 | 0.361 | 0.395 |

| Weather | | Combined | | | | | | Level Shift | | | | | | Spike | | | | | |
|---|---|---|---|---|---|---|---|---|---|---|---|---|---|---|---|---|---|---|---|
| Model | | PatchLGA | | PatchTST | | TimeMixer | | PatchLGA | | PatchTST | | TimeMixer | | PatchLGA | | PatchTST | | TimeMixer | |
| H | Metric | MSE | MAE | MSE | MAE | MSE | MAE | MSE | MAE | MSE | MAE | MSE | MAE | MSE | MAE | MSE | MAE | MSE | MAE |
| 96 | 1 | 0.165 | 0.224 | 0.167 | 0.224 | 0.161 | 0.217 | 0.164 | 0.221 | 0.165 | 0.222 | 0.160 | 0.215 | 0.150 | 0.204 | 0.150 | 0.202 | 0.149 | 0.201 |
| 96 | 2 | 0.197 | 0.256 | 0.202 | 0.259 | 0.189 | 0.244 | 0.195 | 0.251 | 0.199 | 0.254 | 0.187 | 0.240 | 0.153 | 0.208 | 0.153 | 0.206 | 0.150 | 0.203 |
| 96 | 3 | 0.244 | 0.285 | 0.256 | 0.288 | 0.240 | 0.272 | 0.241 | 0.278 | 0.252 | 0.281 | 0.235 | 0.267 | 0.155 | 0.211 | 0.155 | 0.210 | 0.152 | 0.206 |
| 96 | 4 | 0.309 | 0.343 | 0.323 | 0.338 | 0.349 | 0.336 | 0.306 | 0.333 | 0.310 | 0.324 | 0.333 | 0.323 | 0.161 | 0.224 | 0.160 | 0.221 | 0.155 | 0.214 |
| 96 | 5 | 0.407 | 0.378 | 0.453 | 0.378 | 0.509 | 0.381 | 0.401 | 0.366 | 0.429 | 0.359 | 0.483 | 0.365 | 0.166 | 0.230 | 0.164 | 0.227 | 0.158 | 0.219 |
| 192 | 1 | 0.209 | 0.264 | 0.209 | 0.263 | 0.206 | 0.262 | 0.208 | 0.262 | 0.208 | 0.261 | 0.205 | 0.260 | 0.196 | 0.245 | 0.198 | 0.247 | 0.192 | 0.244 |
| 192 | 2 | 0.239 | 0.296 | 0.234 | 0.292 | 0.239 | 0.293 | 0.236 | 0.291 | 0.232 | 0.288 | 0.236 | 0.289 | 0.197 | 0.248 | 0.200 | 0.249 | 0.194 | 0.246 |
| 192 | 3 | 0.278 | 0.321 | 0.279 | 0.321 | 0.294 | 0.325 | 0.273 | 0.314 | 0.276 | 0.316 | 0.288 | 0.319 | 0.200 | 0.252 | 0.201 | 0.252 | 0.196 | 0.249 |
| 192 | 4 | 0.340 | 0.374 | 0.358 | 0.370 | 0.400 | 0.392 | 0.330 | 0.360 | 0.346 | 0.359 | 0.383 | 0.378 | 0.204 | 0.263 | 0.203 | 0.261 | 0.199 | 0.258 |
| 192 | 5 | 0.417 | 0.403 | 0.512 | 0.418 | 0.544 | 0.437 | 0.402 | 0.385 | 0.487 | 0.403 | 0.516 | 0.419 | 0.209 | 0.269 | 0.206 | 0.266 | 0.202 | 0.263 |
| 336 | 1 | 0.257 | 0.301 | 0.259 | 0.303 | 0.260 | 0.306 | 0.256 | 0.299 | 0.258 | 0.301 | 0.259 | 0.304 | 0.248 | 0.285 | 0.246 | 0.284 | 0.247 | 0.288 |
| 336 | 2 | 0.277 | 0.326 | 0.287 | 0.334 | 0.292 | 0.339 | 0.275 | 0.323 | 0.285 | 0.331 | 0.289 | 0.335 | 0.249 | 0.288 | 0.248 | 0.287 | 0.250 | 0.290 |
| 336 | 3 | 0.307 | 0.350 | 0.323 | 0.359 | 0.339 | 0.369 | 0.303 | 0.344 | 0.320 | 0.354 | 0.333 | 0.363 | 0.251 | 0.290 | 0.251 | 0.291 | 0.250 | 0.292 |
| 336 | 4 | 0.358 | 0.396 | 0.375 | 0.398 | 0.443 | 0.437 | 0.347 | 0.383 | 0.367 | 0.388 | 0.421 | 0.420 | 0.253 | 0.299 | 0.252 | 0.298 | 0.253 | 0.303 |
| 336 | 5 | 0.448 | 0.431 | 0.486 | 0.434 | 0.565 | 0.479 | 0.428 | 0.414 | 0.471 | 0.421 | 0.531 | 0.459 | 0.257 | 0.304 | 0.255 | 0.303 | 0.258 | 0.309 |
| 720 | 1 | 0.322 | 0.347 | 0.321 | 0.346 | 0.343 | 0.366 | 0.322 | 0.345 | 0.320 | 0.345 | 0.342 | 0.363 | 0.316 | 0.336 | 0.312 | 0.334 | 0.325 | 0.342 |
| 720 | 2 | 0.342 | 0.372 | 0.339 | 0.370 | 0.382 | 0.403 | 0.339 | 0.367 | 0.337 | 0.367 | 0.378 | 0.398 | 0.318 | 0.338 | 0.314 | 0.336 | 0.326 | 0.345 |
| 720 | 3 | 0.376 | 0.397 | 0.365 | 0.390 | 0.431 | 0.434 | 0.372 | 0.390 | 0.361 | 0.385 | 0.424 | 0.426 | 0.320 | 0.341 | 0.316 | 0.338 | 0.328 | 0.347 |
| 720 | 4 | 0.435 | 0.442 | 0.419 | 0.430 | 0.572 | 0.518 | 0.421 | 0.426 | 0.408 | 0.417 | 0.551 | 0.501 | 0.319 | 0.348 | 0.315 | 0.346 | 0.332 | 0.360 |
| 720 | 5 | 0.545 | 0.479 | 0.512 | 0.462 | 0.685 | 0.558 | 0.516 | 0.458 | 0.489 | 0.444 | 0.655 | 0.538 | 0.321 | 0.353 | 0.317 | 0.349 | 0.335 | 0.365 |

Table 16: Full experimental result on ETTh1, ETTh2, electricity with 512 input lengths.

**ETTh1**

| H | Metric | Combined PatchLGA MSE | MAE | PatchTST MSE | MAE | TimeMixer MSE | MAE | Level Shift PatchLGA MSE | MAE | PatchTST MSE | MAE | TimeMixer MSE | MAE | Spike PatchLGA MSE | MAE | PatchTST MSE | MAE | TimeMixer MSE | MAE |
|---|---|---|---|---|---|---|---|---|---|---|---|---|---|---|---|---|---|---|---|
| 96 | 1 | 0.373 | 0.399 | 0.370 | 0.397 | 0.376 | 0.402 | 0.372 | 0.399 | 0.370 | 0.396 | 0.376 | 0.402 | 0.371 | 0.397 | 0.368 | 0.394 | 0.374 | 0.400 |
| 96 | 2 | 0.380 | 0.406 | 0.377 | 0.404 | 0.383 | 0.410 | 0.379 | 0.405 | 0.376 | 0.402 | 0.383 | 0.408 | 0.371 | 0.398 | 0.368 | 0.395 | 0.374 | 0.401 |
| 96 | 3 | 0.440 | 0.441 | 0.444 | 0.441 | 0.443 | 0.445 | 0.434 | 0.436 | 0.438 | 0.436 | 0.438 | 0.440 | 0.376 | 0.402 | 0.375 | 0.400 | 0.378 | 0.405 |
| 96 | 4 | 0.499 | 0.478 | 0.505 | 0.480 | 0.512 | 0.486 | 0.484 | 0.468 | 0.492 | 0.472 | 0.496 | 0.476 | 0.383 | 0.408 | 0.385 | 0.407 | 0.387 | 0.411 |
| 96 | 5 | 0.669 | 0.547 | 0.700 | 0.557 | 0.687 | 0.559 | 0.636 | 0.530 | 0.676 | 0.544 | 0.652 | 0.540 | 0.399 | 0.417 | 0.407 | 0.418 | 0.401 | 0.421 |
| 192 | 1 | 0.417 | 0.425 | 0.414 | 0.425 | 0.434 | 0.440 | 0.417 | 0.424 | 0.414 | 0.424 | 0.434 | 0.440 | 0.415 | 0.423 | 0.413 | 0.423 | 0.432 | 0.438 |
| 192 | 2 | 0.422 | 0.430 | 0.421 | 0.431 | 0.438 | 0.444 | 0.422 | 0.429 | 0.420 | 0.430 | 0.438 | 0.444 | 0.415 | 0.423 | 0.414 | 0.424 | 0.432 | 0.439 |
| 192 | 3 | 0.467 | 0.459 | 0.474 | 0.465 | 0.480 | 0.470 | 0.463 | 0.455 | 0.469 | 0.460 | 0.476 | 0.466 | 0.418 | 0.426 | 0.419 | 0.428 | 0.434 | 0.441 |
| 192 | 4 | 0.533 | 0.500 | 0.536 | 0.504 | 0.538 | 0.506 | 0.521 | 0.491 | 0.526 | 0.498 | 0.527 | 0.500 | 0.425 | 0.432 | 0.427 | 0.434 | 0.441 | 0.445 |
| 192 | 5 | 0.693 | 0.568 | 0.720 | 0.586 | 0.678 | 0.570 | 0.668 | 0.554 | 0.707 | 0.577 | 0.653 | 0.556 | 0.436 | 0.440 | 0.440 | 0.444 | 0.450 | 0.452 |
| 336 | 1 | 0.427 | 0.432 | 0.434 | 0.437 | 0.464 | 0.463 | 0.427 | 0.432 | 0.434 | 0.437 | 0.464 | 0.463 | 0.427 | 0.432 | 0.433 | 0.437 | 0.463 | 0.462 |
| 336 | 2 | 0.431 | 0.436 | 0.437 | 0.442 | 0.469 | 0.467 | 0.430 | 0.435 | 0.437 | 0.440 | 0.467 | 0.465 | 0.427 | 0.432 | 0.434 | 0.438 | 0.464 | 0.463 |
| 336 | 3 | 0.476 | 0.465 | 0.480 | 0.469 | 0.542 | 0.507 | 0.471 | 0.460 | 0.474 | 0.464 | 0.529 | 0.499 | 0.432 | 0.436 | 0.438 | 0.441 | 0.474 | 0.469 |
| 336 | 4 | 0.554 | 0.513 | 0.558 | 0.516 | 0.619 | 0.553 | 0.540 | 0.503 | 0.543 | 0.507 | 0.591 | 0.537 | 0.440 | 0.442 | 0.447 | 0.448 | 0.489 | 0.479 |
| 336 | 5 | 0.717 | 0.584 | 0.712 | 0.586 | 0.796 | 0.631 | 0.689 | 0.568 | 0.685 | 0.570 | 0.745 | 0.605 | 0.450 | 0.451 | 0.458 | 0.456 | 0.508 | 0.492 |
| 720 | 1 | 0.446 | 0.463 | 0.445 | 0.463 | 0.475 | 0.480 | 0.446 | 0.463 | 0.446 | 0.463 | 0.475 | 0.480 | 0.447 | 0.463 | 0.446 | 0.463 | 0.475 | 0.480 |
| 720 | 2 | 0.448 | 0.467 | 0.449 | 0.467 | 0.476 | 0.483 | 0.448 | 0.466 | 0.448 | 0.466 | 0.476 | 0.482 | 0.446 | 0.463 | 0.446 | 0.464 | 0.475 | 0.481 |
| 720 | 3 | 0.500 | 0.496 | 0.499 | 0.497 | 0.527 | 0.510 | 0.495 | 0.493 | 0.494 | 0.493 | 0.521 | 0.506 | 0.450 | 0.466 | 0.450 | 0.467 | 0.480 | 0.483 |
| 720 | 4 | 0.601 | 0.549 | 0.600 | 0.551 | 0.620 | 0.558 | 0.579 | 0.537 | 0.578 | 0.538 | 0.597 | 0.546 | 0.466 | 0.476 | 0.467 | 0.477 | 0.497 | 0.493 |
| 720 | 5 | 0.768 | 0.621 | 0.765 | 0.622 | 0.770 | 0.623 | 0.730 | 0.601 | 0.727 | 0.603 | 0.728 | 0.603 | 0.479 | 0.486 | 0.481 | 0.488 | 0.512 | 0.503 |

**ETTh2**

| H | Metric | Combined PatchLGA MSE | MAE | PatchTST MSE | MAE | TimeMixer MSE | MAE | Level Shift PatchLGA MSE | MAE | PatchTST MSE | MAE | TimeMixer MSE | MAE | Spike PatchLGA MSE | MAE | PatchTST MSE | MAE | TimeMixer MSE | MAE |
|---|---|---|---|---|---|---|---|---|---|---|---|---|---|---|---|---|---|---|---|
| 96 | 1 | 0.284 | 0.344 | 0.277 | 0.338 | 0.292 | 0.348 | 0.283 | 0.343 | 0.276 | 0.337 | 0.291 | 0.348 | 0.282 | 0.341 | 0.275 | 0.336 | 0.289 | 0.347 |
| 96 | 2 | 0.291 | 0.353 | 0.287 | 0.350 | 0.304 | 0.361 | 0.291 | 0.352 | 0.286 | 0.348 | 0.302 | 0.360 | 0.282 | 0.342 | 0.275 | 0.337 | 0.290 | 0.348 |
| 96 | 3 | 0.305 | 0.363 | 0.306 | 0.362 | 0.326 | 0.376 | 0.304 | 0.361 | 0.303 | 0.360 | 0.322 | 0.373 | 0.282 | 0.343 | 0.277 | 0.339 | 0.293 | 0.350 |
| 96 | 4 | 0.334 | 0.385 | 0.343 | 0.389 | 0.358 | 0.401 | 0.329 | 0.380 | 0.336 | 0.384 | 0.352 | 0.396 | 0.287 | 0.347 | 0.282 | 0.343 | 0.294 | 0.353 |
| 96 | 5 | 0.367 | 0.401 | 0.386 | 0.411 | 0.402 | 0.423 | 0.362 | 0.396 | 0.378 | 0.404 | 0.394 | 0.417 | 0.287 | 0.348 | 0.282 | 0.344 | 0.295 | 0.354 |
| 192 | 1 | 0.347 | 0.388 | 0.358 | 0.394 | 0.351 | 0.387 | 0.347 | 0.388 | 0.358 | 0.393 | 0.351 | 0.387 | 0.346 | 0.387 | 0.359 | 0.393 | 0.351 | 0.386 |
| 192 | 2 | 0.354 | 0.397 | 0.362 | 0.401 | 0.361 | 0.398 | 0.354 | 0.397 | 0.364 | 0.401 | 0.360 | 0.397 | 0.346 | 0.387 | 0.357 | 0.392 | 0.352 | 0.387 |
| 192 | 3 | 0.364 | 0.406 | 0.373 | 0.410 | 0.381 | 0.411 | 0.365 | 0.405 | 0.375 | 0.410 | 0.377 | 0.409 | 0.347 | 0.388 | 0.358 | 0.394 | 0.354 | 0.389 |
| 192 | 4 | 0.387 | 0.423 | 0.405 | 0.430 | 0.412 | 0.433 | 0.384 | 0.421 | 0.402 | 0.427 | 0.408 | 0.430 | 0.350 | 0.391 | 0.362 | 0.397 | 0.354 | 0.390 |
| 192 | 5 | 0.406 | 0.433 | 0.437 | 0.446 | 0.451 | 0.453 | 0.403 | 0.430 | 0.432 | 0.442 | 0.445 | 0.449 | 0.350 | 0.392 | 0.362 | 0.398 | 0.355 | 0.391 |
| 336 | 1 | 0.334 | 0.389 | 0.359 | 0.400 | 0.359 | 0.406 | 0.334 | 0.388 | 0.359 | 0.400 | 0.359 | 0.405 | 0.334 | 0.387 | 0.360 | 0.400 | 0.358 | 0.405 |
| 336 | 2 | 0.341 | 0.399 | 0.358 | 0.404 | 0.369 | 0.418 | 0.343 | 0.399 | 0.363 | 0.406 | 0.370 | 0.417 | 0.332 | 0.387 | 0.355 | 0.398 | 0.358 | 0.405 |
| 336 | 3 | 0.350 | 0.408 | 0.365 | 0.412 | 0.390 | 0.433 | 0.352 | 0.407 | 0.372 | 0.414 | 0.388 | 0.431 | 0.333 | 0.389 | 0.355 | 0.400 | 0.359 | 0.408 |
| 336 | 4 | 0.373 | 0.423 | 0.387 | 0.426 | 0.428 | 0.456 | 0.369 | 0.420 | 0.387 | 0.424 | 0.420 | 0.450 | 0.339 | 0.393 | 0.364 | 0.405 | 0.365 | 0.413 |
| 336 | 5 | 0.387 | 0.430 | 0.414 | 0.440 | 0.470 | 0.477 | 0.384 | 0.427 | 0.415 | 0.439 | 0.458 | 0.469 | 0.340 | 0.394 | 0.363 | 0.405 | 0.368 | 0.416 |
| 720 | 1 | 0.387 | 0.429 | 0.379 | 0.423 | 0.405 | 0.436 | 0.386 | 0.428 | 0.379 | 0.422 | 0.405 | 0.436 | 0.388 | 0.429 | 0.380 | 0.423 | 0.406 | 0.437 |
| 720 | 2 | 0.396 | 0.439 | 0.388 | 0.432 | 0.416 | 0.447 | 0.397 | 0.438 | 0.389 | 0.432 | 0.417 | 0.446 | 0.387 | 0.429 | 0.378 | 0.423 | 0.405 | 0.437 |
| 720 | 3 | 0.412 | 0.452 | 0.405 | 0.444 | 0.438 | 0.461 | 0.412 | 0.449 | 0.406 | 0.443 | 0.437 | 0.459 | 0.388 | 0.432 | 0.379 | 0.424 | 0.406 | 0.440 |
| 720 | 4 | 0.433 | 0.468 | 0.428 | 0.462 | 0.466 | 0.480 | 0.424 | 0.460 | 0.422 | 0.457 | 0.458 | 0.475 | 0.395 | 0.436 | 0.385 | 0.429 | 0.412 | 0.443 |
| 720 | 5 | 0.457 | 0.481 | 0.470 | 0.485 | 0.514 | 0.505 | 0.447 | 0.473 | 0.462 | 0.479 | 0.503 | 0.498 | 0.396 | 0.438 | 0.386 | 0.430 | 0.414 | 0.445 |

**Electricity**

| H | Metric | Combined PatchLGA MSE | MAE | PatchTST MSE | MAE | TimeMixer MSE | MAE | Level Shift PatchLGA MSE | MAE | PatchTST MSE | MAE | TimeMixer MSE | MAE | Spike PatchLGA MSE | MAE | PatchTST MSE | MAE | TimeMixer MSE | MAE |
|---|---|---|---|---|---|---|---|---|---|---|---|---|---|---|---|---|---|---|---|
| 96 | 1 | 0.138 | 0.234 | 0.138 | 0.234 | 0.141 | 0.238 | 0.137 | 0.233 | 0.137 | 0.233 | 0.139 | 0.237 | 0.130 | 0.226 | 0.130 | 0.225 | 0.133 | 0.230 |
| 96 | 2 | 0.149 | 0.247 | 0.151 | 0.249 | 0.153 | 0.251 | 0.146 | 0.243 | 0.148 | 0.245 | 0.150 | 0.247 | 0.132 | 0.229 | 0.132 | 0.228 | 0.135 | 0.233 |
| 96 | 3 | 0.163 | 0.260 | 0.166 | 0.262 | 0.170 | 0.265 | 0.159 | 0.254 | 0.163 | 0.257 | 0.165 | 0.259 | 0.134 | 0.230 | 0.134 | 0.230 | 0.136 | 0.234 |
| 96 | 4 | 0.199 | 0.292 | 0.201 | 0.294 | 0.210 | 0.297 | 0.191 | 0.282 | 0.194 | 0.285 | 0.201 | 0.287 | 0.138 | 0.236 | 0.138 | 0.235 | 0.140 | 0.240 |
| 96 | 5 | 0.261 | 0.328 | 0.266 | 0.333 | 0.282 | 0.336 | 0.251 | 0.316 | 0.258 | 0.322 | 0.268 | 0.323 | 0.142 | 0.241 | 0.142 | 0.240 | 0.144 | 0.244 |
| 192 | 1 | 0.154 | 0.251 | 0.156 | 0.253 | 0.160 | 0.255 | 0.153 | 0.249 | 0.155 | 0.252 | 0.159 | 0.253 | 0.148 | 0.244 | 0.149 | 0.244 | 0.154 | 0.247 |
| 192 | 2 | 0.165 | 0.263 | 0.168 | 0.266 | 0.172 | 0.267 | 0.162 | 0.259 | 0.166 | 0.263 | 0.169 | 0.264 | 0.150 | 0.246 | 0.151 | 0.247 | 0.156 | 0.250 |
| 192 | 3 | 0.178 | 0.274 | 0.181 | 0.278 | 0.188 | 0.281 | 0.174 | 0.269 | 0.177 | 0.273 | 0.184 | 0.276 | 0.151 | 0.248 | 0.152 | 0.249 | 0.157 | 0.251 |
| 192 | 4 | 0.211 | 0.303 | 0.209 | 0.306 | 0.227 | 0.312 | 0.201 | 0.294 | 0.202 | 0.297 | 0.219 | 0.303 | 0.155 | 0.253 | 0.156 | 0.254 | 0.160 | 0.256 |
| 192 | 5 | 0.267 | 0.337 | 0.256 | 0.338 | 0.301 | 0.352 | 0.255 | 0.325 | 0.246 | 0.326 | 0.289 | 0.341 | 0.159 | 0.257 | 0.161 | 0.259 | 0.164 | 0.260 |
| 336 | 1 | 0.171 | 0.269 | 0.172 | 0.269 | 0.179 | 0.274 | 0.171 | 0.267 | 0.171 | 0.268 | 0.178 | 0.273 | 0.165 | 0.262 | 0.164 | 0.261 | 0.173 | 0.267 |
| 336 | 2 | 0.182 | 0.281 | 0.185 | 0.283 | 0.192 | 0.287 | 0.179 | 0.277 | 0.183 | 0.280 | 0.189 | 0.284 | 0.167 | 0.264 | 0.166 | 0.263 | 0.175 | 0.270 |
| 336 | 3 | 0.196 | 0.292 | 0.202 | 0.297 | 0.210 | 0.301 | 0.191 | 0.287 | 0.199 | 0.293 | 0.206 | 0.296 | 0.168 | 0.266 | 0.167 | 0.265 | 0.176 | 0.272 |
| 336 | 4 | 0.229 | 0.321 | 0.231 | 0.324 | 0.246 | 0.331 | 0.220 | 0.311 | 0.225 | 0.317 | 0.238 | 0.323 | 0.172 | 0.271 | 0.172 | 0.270 | 0.180 | 0.276 |
| 336 | 5 | 0.288 | 0.356 | 0.287 | 0.361 | 0.326 | 0.372 | 0.276 | 0.344 | 0.278 | 0.351 | 0.313 | 0.361 | 0.176 | 0.275 | 0.176 | 0.276 | 0.184 | 0.280 |
| 720 | 1 | 0.208 | 0.301 | 0.209 | 0.303 | 0.210 | 0.303 | 0.207 | 0.300 | 0.208 | 0.301 | 0.209 | 0.302 | 0.203 | 0.295 | 0.200 | 0.294 | 0.205 | 0.297 |
| 720 | 2 | 0.219 | 0.313 | 0.232 | 0.320 | 0.220 | 0.314 | 0.217 | 0.309 | 0.230 | 0.316 | 0.218 | 0.311 | 0.204 | 0.298 | 0.203 | 0.297 | 0.206 | 0.299 |
| 720 | 3 | 0.235 | 0.325 | 0.253 | 0.336 | 0.234 | 0.325 | 0.231 | 0.320 | 0.254 | 0.331 | 0.231 | 0.321 | 0.206 | 0.299 | 0.206 | 0.300 | 0.207 | 0.301 |
| 720 | 4 | 0.264 | 0.350 | 0.285 | 0.362 | 0.265 | 0.351 | 0.255 | 0.342 | 0.282 | 0.356 | 0.258 | 0.343 | 0.209 | 0.303 | 0.212 | 0.306 | 0.210 | 0.305 |
| 720 | 5 | 0.320 | 0.385 | 0.352 | 0.402 | 0.338 | 0.386 | 0.308 | 0.374 | 0.350 | 0.394 | 0.327 | 0.376 | 0.213 | 0.307 | 0.220 | 0.313 | 0.214 | 0.309 |

level 5 with a 336-step forecasting horizon, LGA reduces the MSE by 16.1% compared to SDP attention (from 0.818 to 0.686).

- **Cross-Temporal Attention (CATS):** LGA also enhances the robustness of cross-attention, although the improvements are less uniform than in the self-attention case. For instance, at a 720-step horizon with severity level 5, LGA improves the MSE by 21.5% (from 1.255 to 0.985). This aligns with our main analysis that operating on linearly embedded noisy inputs can sometimes limit the consistency of performance gains.

- **Self-Channel Attention (iTransformer):** For this mechanism, LGA offers modest but stable improvements, particularly under high-severity corruption. The linear embedding applied to the entire time series, as noted in our main discussion, disrupts some temporal local geometry, making the gains less pronounced than with PatchTST. For example, at the 720-step horizon under severity 5, LGA still provides a 4.0% reduction in MSE (from 1.439 to 1.382).

These comprehensive results further validate our conclusion that while LGA is a broadly applicable technique that enhances robustness across all tested attention mechanisms, its integration with self-temporal attention provides the most consistent and substantial improvements for time series forecasting under realistic corruptions.

Table 17: Detailed performance comparison of different attention mechanisms on the ETTm1 dataset with input length 512 under combined corruptions.

| ETTm1 | | H | 96 | | | | | 192 | | | | | 336 | | | | | 720 | | | | |
|---|---|---|---|---|---|---|---|---|---|---|---|---|---|---|---|---|---|---|---|---|---|---|
| Mod. | Atten. | Metric | 1 | 2 | 3 | 4 | 5 | 1 | 2 | 3 | 4 | 5 | 1 | 2 | 3 | 4 | 5 | 1 | 2 | 3 | 4 | 5 |
| PatchTST | LGA | MSE | 0.297 | 0.314 | 0.527 | 0.657 | 0.820 | 0.341 | 0.354 | 0.504 | 0.603 | 0.722 | 0.372 | 0.385 | 0.502 | 0.586 | 0.686 | 0.425 | 0.436 | 0.543 | 0.623 | 0.709 |
| | | MAE | 0.353 | 0.368 | 0.456 | 0.530 | 0.594 | 0.376 | 0.388 | 0.462 | 0.521 | 0.573 | 0.392 | 0.402 | 0.465 | 0.516 | 0.564 | 0.421 | 0.429 | 0.488 | 0.536 | 0.578 |
| | SDP | MSE | 0.304 | 0.326 | 0.625 | 0.743 | 0.944 | 0.342 | 0.357 | 0.581 | 0.661 | 0.812 | 0.370 | 0.383 | 0.630 | 0.692 | 0.818 | 0.420 | 0.433 | 0.622 | 0.684 | 0.782 |
| | | MAE | 0.357 | 0.377 | 0.497 | 0.566 | 0.636 | 0.378 | 0.391 | 0.491 | 0.542 | 0.601 | 0.397 | 0.408 | 0.519 | 0.563 | 0.611 | 0.424 | 0.432 | 0.519 | 0.562 | 0.604 |
| CATS | LGA | MSE | 0.290 | 0.313 | 0.624 | 0.972 | 1.138 | 0.325 | 0.340 | 0.635 | 0.960 | 1.055 | 0.357 | 0.370 | 0.614 | 0.856 | 0.971 | 0.410 | 0.423 | 0.774 | 1.100 | 0.985 |
| | | MAE | 0.341 | 0.358 | 0.454 | 0.563 | 0.616 | 0.365 | 0.377 | 0.478 | 0.582 | 0.624 | 0.390 | 0.400 | 0.492 | 0.577 | 0.624 | 0.424 | 0.432 | 0.560 | 0.665 | 0.647 |
| | SDP | MSE | 0.289 | 0.313 | 0.687 | 0.862 | 1.026 | 0.327 | 0.347 | 0.778 | 0.949 | 1.146 | 0.355 | 0.368 | 0.711 | 0.889 | 0.979 | 0.410 | 0.426 | 0.940 | 1.242 | 1.255 |
| | | MAE | 0.344 | 0.362 | 0.487 | 0.559 | 0.614 | 0.368 | 0.383 | 0.521 | 0.587 | 0.645 | 0.389 | 0.400 | 0.526 | 0.591 | 0.630 | 0.422 | 0.433 | 0.620 | 0.724 | 0.722 |
| iTrans. | LGA | MSE | 0.357 | 0.391 | 0.718 | 0.946 | 1.222 | 0.408 | 0.440 | 0.800 | 1.051 | 1.336 | 0.455 | 0.493 | 0.864 | 1.105 | 1.367 | 0.506 | 0.541 | 0.894 | 1.126 | 1.382 |
| | | MAE | 0.389 | 0.412 | 0.508 | 0.598 | 0.683 | 0.413 | 0.434 | 0.533 | 0.626 | 0.707 | 0.442 | 0.464 | 0.562 | 0.654 | 0.732 | 0.471 | 0.490 | 0.586 | 0.672 | 0.746 |
| | SDP | MSE | 0.358 | 0.390 | 0.690 | 0.899 | 1.166 | 0.394 | 0.425 | 0.751 | 0.965 | 1.229 | 0.453 | 0.488 | 0.853 | 1.095 | 1.401 | 0.511 | 0.558 | 0.910 | 1.149 | 1.439 |
| | | MAE | 0.387 | 0.409 | 0.498 | 0.585 | 0.667 | 0.403 | 0.424 | 0.516 | 0.603 | 0.683 | 0.436 | 0.457 | 0.557 | 0.656 | 0.740 | 0.472 | 0.494 | 0.591 | 0.680 | 0.761 |

## E.3 Comprehensive Evaluation of Input Length Impact on Forecasting Robustness

We examined how varying input sequence length affects forecasting performance under corrupted conditions. Here, we present a more detailed analysis with complete results across all input lengths, forecasting horizons, and severity levels for the ETTm1 dataset with combined corruptions. Table 18 provides comprehensive performance metrics for PatchLGA, PatchTST, and TimeMixer with input lengths ranging from 192 to 1024 timesteps. These detailed results allow us to examine the complex relationship between input context, forecasting horizon, and model architecture under various corruption intensities.

At short input lengths (e.g., 192), TimeMixer is competitive with or occasionally outperforms the transformer models at low corruption severities (level 1), particularly for the shortest forecasting horizon (H=96). However, as input length increases, the transformer models, especially PatchLGA, demonstrate progressive and significant performance improvements, while the performance of TimeMixer tends to stagnate or even deteriorate at longer input lengths (e.g., 720 and 1024). With an input length of 512, PatchLGA consistently outperforms both alternatives across all forecasting horizons and severity levels, establishing it as the most robust model at this context size. For the longest forecasting horizon (720), the impact of input length becomes even more critical. With a 192-timestep input at severity level 5, PatchLGA achieves an MSE of 1.140. When the input length is increased to 512 timesteps, the MSE improves to 0.709, a substantial 37.8% reduction. This demonstrates that long-range dependencies, effectively captured by LGA, become increasingly important for distant forecasting, especially under severe corruptions.

These findings have important implications for deploying forecasting models in real-world scenarios. While linear models may be adequate for short-term forecasting with limited historical data and

minimal corruption, transformer models with LGA provide substantial benefits when longer historical context is available, particularly under challenging corruption conditions.

Table 18: Detailed performance comparison across different input sequence lengths (192, 336, 512, 720, 1024) on the ETTm1 dataset with combined corruptions.

| ETTm1 | | $H$ | 96 | | | | | 192 | | | | | 336 | | | | | 720 | | | | |
|---|---|---|---|---|---|---|---|---|---|---|---|---|---|---|---|---|---|---|---|---|---|---|
| Input | Model | Metric | 1 | 2 | 3 | 4 | 5 | 1 | 2 | 3 | 4 | 5 | 1 | 2 | 3 | 4 | 5 | 1 | 2 | 3 | 4 | 5 |
| 192 | PatchLGA | MSE | 0.315 | 0.339 | 0.579 | 0.757 | 0.924 | 0.349 | 0.369 | 0.621 | 0.811 | 0.971 | 0.375 | 0.392 | 0.719 | 0.939 | 1.065 | 0.436 | 0.460 | 0.767 | 0.982 | 1.140 |
| | | MAE | 0.357 | 0.376 | 0.462 | 0.541 | 0.601 | 0.380 | 0.396 | 0.486 | 0.567 | 0.626 | 0.398 | 0.413 | 0.516 | 0.606 | 0.658 | 0.439 | 0.455 | 0.550 | 0.631 | 0.695 |
| | PatchTST | MSE | 0.313 | 0.342 | 0.713 | 0.889 | 1.134 | 0.350 | 0.379 | 0.818 | 1.020 | 1.273 | 0.383 | 0.408 | 0.852 | 1.061 | 1.286 | 0.440 | 0.464 | 0.878 | 1.090 | 1.336 |
| | | MAE | 0.355 | 0.380 | 0.496 | 0.576 | 0.652 | 0.380 | 0.402 | 0.536 | 0.623 | 0.695 | 0.402 | 0.420 | 0.557 | 0.650 | 0.717 | 0.436 | 0.452 | 0.583 | 0.673 | 0.737 |
| | TimeMixer | MSE | 0.312 | 0.342 | 0.667 | 0.857 | 1.072 | 0.351 | 0.379 | 0.785 | 1.037 | 1.238 | 0.376 | 0.401 | 0.733 | 0.881 | 1.107 | 0.439 | 0.463 | 0.748 | 0.886 | 1.085 |
| | | MAE | 0.358 | 0.380 | 0.483 | 0.564 | 0.635 | 0.386 | 0.407 | 0.526 | 0.619 | 0.692 | 0.400 | 0.417 | 0.521 | 0.591 | 0.662 | 0.439 | 0.453 | 0.546 | 0.609 | 0.678 |
| 336 | PatchLGA | MSE | 0.299 | 0.318 | 0.589 | 0.778 | 0.939 | 0.336 | 0.352 | 0.596 | 0.775 | 0.863 | 0.371 | 0.387 | 0.659 | 0.846 | 0.912 | 0.424 | 0.439 | 0.656 | 0.800 | 0.883 |
| | | MAE | 0.351 | 0.369 | 0.466 | 0.553 | 0.620 | 0.375 | 0.390 | 0.489 | 0.570 | 0.613 | 0.397 | 0.411 | 0.519 | 0.603 | 0.638 | 0.428 | 0.439 | 0.532 | 0.598 | 0.633 |
| | PatchTST | MSE | 0.309 | 0.338 | 0.689 | 0.852 | 1.115 | 0.343 | 0.367 | 0.762 | 0.925 | 1.165 | 0.378 | 0.402 | 0.752 | 0.889 | 1.138 | 0.430 | 0.459 | 0.951 | 1.092 | 1.329 |
| | | MAE | 0.358 | 0.382 | 0.512 | 0.591 | 0.675 | 0.380 | 0.399 | 0.536 | 0.613 | 0.685 | 0.404 | 0.422 | 0.555 | 0.624 | 0.700 | 0.434 | 0.452 | 0.621 | 0.689 | 0.748 |
| | TimeMixer | MSE | 0.307 | 0.337 | 0.681 | 0.870 | 1.076 | 0.343 | 0.365 | 0.660 | 0.804 | 0.977 | 0.373 | 0.392 | 0.658 | 0.822 | 0.975 | 0.428 | 0.441 | 0.624 | 0.744 | 0.840 |
| | | MAE | 0.357 | 0.380 | 0.498 | 0.582 | 0.658 | 0.378 | 0.395 | 0.499 | 0.562 | 0.628 | 0.400 | 0.415 | 0.521 | 0.596 | 0.657 | 0.423 | 0.433 | 0.512 | 0.571 | 0.615 |
| 512 | PatchLGA | MSE | 0.297 | 0.314 | 0.527 | 0.657 | 0.820 | 0.341 | 0.354 | 0.504 | 0.603 | 0.722 | 0.372 | 0.385 | 0.502 | 0.586 | 0.686 | 0.425 | 0.436 | 0.543 | 0.623 | 0.709 |
| | | MAE | 0.353 | 0.368 | 0.456 | 0.530 | 0.594 | 0.376 | 0.388 | 0.462 | 0.521 | 0.573 | 0.392 | 0.402 | 0.465 | 0.516 | 0.564 | 0.421 | 0.429 | 0.488 | 0.536 | 0.578 |
| | PatchTST | MSE | 0.304 | 0.326 | 0.625 | 0.743 | 0.944 | 0.342 | 0.357 | 0.581 | 0.661 | 0.812 | 0.370 | 0.383 | 0.630 | 0.692 | 0.818 | 0.420 | 0.433 | 0.622 | 0.684 | 0.782 |
| | | MAE | 0.357 | 0.377 | 0.497 | 0.566 | 0.636 | 0.378 | 0.391 | 0.491 | 0.542 | 0.601 | 0.397 | 0.408 | 0.519 | 0.563 | 0.611 | 0.424 | 0.432 | 0.519 | 0.562 | 0.604 |
| | TimeMixer | MSE | 0.307 | 0.335 | 0.619 | 0.769 | 0.953 | 0.342 | 0.358 | 0.537 | 0.643 | 0.740 | 0.381 | 0.396 | 0.600 | 0.709 | 0.829 | 0.436 | 0.449 | 0.620 | 0.745 | 0.828 |
| | | MAE | 0.363 | 0.383 | 0.490 | 0.558 | 0.629 | 0.380 | 0.393 | 0.474 | 0.527 | 0.574 | 0.407 | 0.418 | 0.511 | 0.567 | 0.622 | 0.432 | 0.441 | 0.524 | 0.587 | 0.626 |
| 720 | PatchLGA | MSE | 0.305 | 0.323 | 0.495 | 0.601 | 0.731 | 0.336 | 0.350 | 0.489 | 0.583 | 0.682 | 0.376 | 0.388 | 0.491 | 0.571 | 0.661 | 0.426 | 0.435 | 0.526 | 0.603 | 0.681 |
| | | MAE | 0.358 | 0.373 | 0.450 | 0.514 | 0.570 | 0.377 | 0.388 | 0.461 | 0.518 | 0.563 | 0.395 | 0.404 | 0.464 | 0.516 | 0.560 | 0.422 | 0.429 | 0.485 | 0.533 | 0.573 |
| | PatchTST | MSE | 0.304 | 0.324 | 0.519 | 0.615 | 0.761 | 0.340 | 0.355 | 0.525 | 0.603 | 0.706 | 0.365 | 0.380 | 0.587 | 0.657 | 0.745 | 0.419 | 0.431 | 0.562 | 0.627 | 0.711 |
| | | MAE | 0.358 | 0.373 | 0.463 | 0.521 | 0.581 | 0.379 | 0.390 | 0.472 | 0.523 | 0.569 | 0.397 | 0.406 | 0.503 | 0.550 | 0.587 | 0.423 | 0.430 | 0.500 | 0.544 | 0.583 |
| | TimeMixer | MSE | 0.332 | 0.358 | 0.705 | 0.940 | 1.079 | 0.344 | 0.364 | 0.590 | 0.707 | 0.831 | 0.371 | 0.384 | 0.540 | 0.639 | 0.741 | 0.456 | 0.468 | 0.636 | 0.745 | 0.853 |
| | | MAE | 0.380 | 0.399 | 0.549 | 0.650 | 0.697 | 0.384 | 0.398 | 0.490 | 0.546 | 0.603 | 0.396 | 0.406 | 0.483 | 0.536 | 0.586 | 0.449 | 0.456 | 0.536 | 0.590 | 0.641 |
| 1024 | PatchLGA | MSE | 0.309 | 0.326 | 0.471 | 0.569 | 0.688 | 0.355 | 0.368 | 0.487 | 0.578 | 0.681 | 0.370 | 0.381 | 0.488 | 0.571 | 0.652 | 0.415 | 0.423 | 0.513 | 0.592 | 0.666 |
| | | MAE | 0.360 | 0.373 | 0.448 | 0.510 | 0.560 | 0.385 | 0.396 | 0.463 | 0.520 | 0.568 | 0.394 | 0.403 | 0.465 | 0.518 | 0.557 | 0.419 | 0.426 | 0.482 | 0.532 | 0.569 |
| | PatchTST | MSE | 0.303 | 0.325 | 0.604 | 0.696 | 0.871 | 0.340 | 0.355 | 0.539 | 0.613 | 0.718 | 0.367 | 0.381 | 0.560 | 0.629 | 0.716 | 0.410 | 0.419 | 0.552 | 0.620 | 0.699 |
| | | MAE | 0.359 | 0.377 | 0.503 | 0.560 | 0.620 | 0.381 | 0.392 | 0.489 | 0.541 | 0.582 | 0.398 | 0.406 | 0.497 | 0.544 | 0.579 | 0.420 | 0.426 | 0.501 | 0.546 | 0.579 |
| | TimeMixer | MSE | 0.322 | 0.347 | 0.594 | 0.703 | 0.853 | 0.359 | 0.375 | 0.573 | 0.664 | 0.778 | 0.399 | 0.417 | 0.633 | 0.753 | 0.884 | 0.454 | 0.465 | 0.645 | 0.794 | 0.878 |
| | | MAE | 0.371 | 0.388 | 0.485 | 0.543 | 0.606 | 0.392 | 0.404 | 0.489 | 0.540 | 0.595 | 0.416 | 0.428 | 0.527 | 0.588 | 0.644 | 0.448 | 0.455 | 0.544 | 0.618 | 0.651 |

### E.3.1 COMPREHENSIVE COMPARISON WITH ALTERNATIVE ROBUST ATTENTION METHODS

Table 19 presents the detailed comparison between LGA and other robust attention mechanisms across all forecasting horizons and severity levels on the ETTm1 dataset with combined corruptions. The results reveal that LGA outperforms the alternative approaches in the vast majority of settings, with its advantage being particularly notable at higher corruption severities. For short-term forecasting (96 horizon) at severity level 5, LGA achieves an MSE of 0.820. This represents a substantial 28.6% improvement over MoM and a 4.2% improvement over Elliptical attention. This advantage is pronounced for medium-term horizons. For instance, on the 336 horizon at severity level 5, the MSE of LGA (0.686) is 37.1% lower than MoM and 11.6% lower than Elliptical attention . For long-term forecasting (720 horizon), while the competition is closer, LGA still demonstrates clear benefits. At severity level 3, for example, MSE of LGA (0.543) is significantly better than SDP and Elliptical attention, although slightly higher than MoM in this specific case.

These comprehensive results confirm our findings that while robust attention mechanisms like MoM and Elliptical attention succeed in vision and language tasks, they do not transfer as effectively to time series forecasting. LGA, specifically tailored for capturing local geometry of temporal structures, yields superior robustness while maintaining computational efficiency comparable to standard attention.

Table 19: Detailed performance comparison of different robust attention methods on the ETTm1 dataset with input length 512 under combined corruptions.

| ETTm1 | | $H$ | 96 | | | | | 192 | | | | | 336 | | | | | 720 | | | | |
|---|---|---|---|---|---|---|---|---|---|---|---|---|---|---|---|---|---|---|---|---|---|---|
| Mod. | Atten. | Metric | 1 | 2 | 3 | 4 | 5 | 1 | 2 | 3 | 4 | 5 | 1 | 2 | 3 | 4 | 5 | 1 | 2 | 3 | 4 | 5 |
| PatchTST | LGA | MSE | 0.297 | 0.314 | 0.527 | 0.657 | 0.820 | 0.341 | 0.354 | 0.504 | 0.603 | 0.722 | 0.372 | 0.385 | 0.502 | 0.586 | 0.686 | 0.425 | 0.436 | 0.543 | 0.623 | 0.709 |
| | | MAE | 0.353 | 0.368 | 0.456 | 0.530 | 0.594 | 0.376 | 0.388 | 0.462 | 0.521 | 0.573 | 0.392 | 0.402 | 0.465 | 0.516 | 0.564 | 0.421 | 0.429 | 0.488 | 0.536 | 0.578 |
| | SDP | MSE | 0.304 | 0.326 | 0.625 | 0.743 | 0.944 | 0.342 | 0.357 | 0.581 | 0.661 | 0.812 | 0.370 | 0.383 | 0.630 | 0.692 | 0.818 | 0.420 | 0.433 | 0.622 | 0.684 | 0.782 |
| | | MAE | 0.357 | 0.377 | 0.497 | 0.566 | 0.636 | 0.378 | 0.391 | 0.491 | 0.542 | 0.601 | 0.397 | 0.408 | 0.519 | 0.563 | 0.611 | 0.424 | 0.432 | 0.519 | 0.562 | 0.604 |
| | Ellip. | MSE | 0.301 | 0.319 | 0.587 | 0.678 | 0.856 | 0.342 | 0.354 | 0.596 | 0.626 | 0.800 | 0.368 | 0.380 | 0.659 | 0.682 | 0.776 | 0.429 | 0.444 | 1.044 | 1.033 | 1.090 |
| | | MAE | 0.356 | 0.372 | 0.480 | 0.537 | 0.601 | 0.379 | 0.390 | 0.493 | 0.525 | 0.589 | 0.395 | 0.404 | 0.522 | 0.552 | 0.589 | 0.434 | 0.443 | 0.650 | 0.676 | 0.692 |
| | MoM | MSE | 0.325 | 0.359 | 0.700 | 0.941 | 1.149 | 0.354 | 0.378 | 0.725 | 0.961 | 1.130 | 0.387 | 0.407 | 0.722 | 0.961 | 1.091 | 0.435 | 0.443 | 0.535 | 0.620 | 0.696 |
| | | MAE | 0.375 | 0.399 | 0.530 | 0.640 | 0.714 | 0.392 | 0.409 | 0.542 | 0.645 | 0.706 | 0.409 | 0.423 | 0.548 | 0.652 | 0.702 | 0.424 | 0.430 | 0.485 | 0.538 | 0.571 |

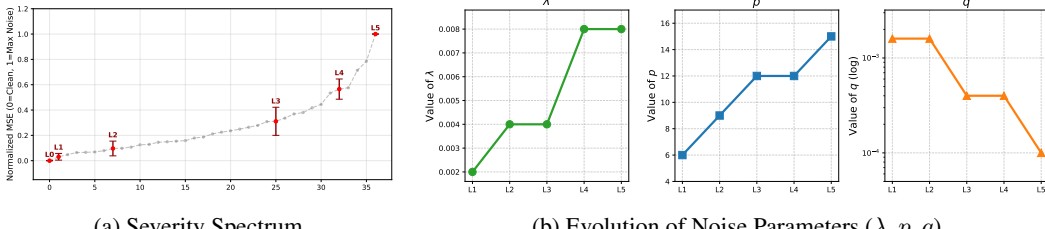

(a) Severity Spectrum        (b) Evolution of Noise Parameters ($\lambda, p, q$)

Figure 8: **Selection and Analysis of Severity Levels.** (a) The severity spectrum obtained by sorting the normalized MSE of all 37 parameter configurations. From this full spectrum, we selected five representative levels (L1–L5, marked in red) that provide a comprehensive coverage of the difficulty range with distinct performance steps. (b) The evolution of noise parameters corresponding to the selected levels. The injection frequency ($\lambda$) and segment length ($p$) strictly increase, while the transition probability ($q$, log-scale) decreases. This confirms that the increasing severity is driven by a systematic intensification of the noise mechanism rather than random permutations.

## F NOISE GENERATION AND PARAMETER SELECTION

This section provides detailed information on our implementation of corruption functions and the experimental process to select appropriate noise parameters. While the main paper presented the theoretical foundation, here we explain the practical implementation details and the empirical validation of parameter settings.

### F.1 IMPLEMENTATION OF CORRUPTION FUNCTIONS

We implement realistic corruptions for time series data through a systematic process that combines Poisson event generation with extreme value theory for threshold estimation. Algorithm 2 presents the complete procedure for injecting both level shift and spike corruptions into time series data. We fix the random seed to 2025 across all experiments to ensure reproducible results.

The GETTHRESHOLDS function in the Algorithm 2 requires detailed explanation, as it implements the DSPOT (Drift SPOT) algorithm (Siffer et al., 2017) to estimate extreme value thresholds. DSPOT extends the SPOT algorithm by explicitly accounting for distributional drift in time series data. Given a significance level $q$, the algorithm computes time-varying thresholds $z_q(t) = \inf\{z : P(X(t) > z) < q\}$ that represent the critical values for identifying extreme events at each time point.

The DSPOT algorithm operates by:

1. Initializing with a burn-in period to establish baseline statistics
2. Iteratively updating Generalized Pareto Distribution (GPD) parameters as new data arrives
3. Computing upper and lower thresholds based on the estimated tail distributions
4. Adapting to potential distributional changes through drift detection mechanisms

For our implementation, we maintain consistent DSPOT hyperparameters across all datasets to ensure fair comparison and reproducibility. The algorithm returns both upper and lower thresholds that are used to determine realistic corruption magnitudes for level shifts and spikes, ensuring that the generated corruptions reflect statistically extreme but realistic deviations from normal behavior.

### F.2 PARAMETER SELECTION PROCESS

Finding appropriate noise parameters for realistic corruptions requires both theoretical justification and empirical validation. Although the theoretical foundation of our noise model is established in Section 4, experimental verification is necessary to determine which parameter settings effectively induce progressively increasing severity across diverse data distributions.

To this end, we conducted a comprehensive grid search over 36 different parameter combinations (varying $\lambda$, $p$, and $q$) alongside a clean baseline. We evaluated the original PatchTST on six benchmark datasets (ETTh1, ETTh2, ETTm1, ETTm2, Electricity, and Weather) to measure the performance

---

**Algorithm 2 TSRBench Corruption Injection**

---

**Require:** Time series $X \in \mathbb{R}^T$, corruption rate $\lambda$, duration parameter $p$, amplitude parameter $q$
**Ensure:** Corrupted time series: $X^{\text{shift}}$, $X^{\text{spike}}$, $X^{\text{combined}}$
1: **Initialize:** $T \leftarrow \text{length}(X)$, $\varepsilon^{\text{shift}}, \varepsilon^{\text{spike}} \leftarrow \mathbf{0}^T$
    **Step 1. Generate corruption events and compute thresholds**
2:   $T' \leftarrow 2T - 1$                               ▷ Extended time interval
3:   $N \sim \text{Poisson}(\lambda \cdot T')$                       ▷ Number of events
4:   $\mathcal{T} \leftarrow \{\tau_1, \tau_2, \ldots, \tau_N\}$ where $\tau_i \sim \mathcal{U}[0, T']$
5:   $\mathcal{T} \leftarrow \{\tau \in \mathcal{T} : \tau \geq T\} - T$          ▷ Keep second half, shift to $[0, T]$
6:   $\text{upper\_threshold}, \text{lower\_threshold} \leftarrow \text{GETTHRESHOLDS}(X, q)$
    **Step 2-1. Level shift corruption injection**
7: **for** each $\tau \in \mathcal{T}$ **do**
8:     $p' \leftarrow \frac{1}{p-1}$                           ▷ Geometric parameter
9:     $d \sim \text{Geometric}(p') + 1$                   ▷ Shift duration
10:    **if** ISBIDIRECTIONAL **then**           ▷ Bidirectional SPOT variants
11:       $h_\tau^{\text{shift}} \leftarrow \begin{cases} \min(\text{upper\_threshold}[\tau : \tau + d] - X[\tau : \tau + d]) & \text{if RANDOM}() < 0.5 \\ \max(\text{lower\_threshold}[\tau : \tau + d] - X[\tau : \tau + d]) & \text{otherwise} \end{cases}$
12:    **else**
13:       $h_\tau^{\text{shift}} \leftarrow \min(\text{upper\_threshold}[\tau : \tau + d] - X[\tau : \tau + d])$
14:    **end if**
15:    $\varepsilon^{\text{shift}}[\tau : \min(\tau + d, T)] \leftarrow h_\tau^{\text{shift}}$
16: **end for**
    **Step 2-2. Spike corruption injection**
17: **for** each $\tau \in \mathcal{T}$ **do**
18:    $d_1 \sim \text{Geometric}(\frac{2}{p})$, $d_2 \sim \text{Geometric}(\frac{2}{p})$        ▷ Rising/decay durations
19:    **if** ISBIDIRECTIONAL **then**           ▷ Bidirectional SPOT variants
20:       $h_\tau^{\text{spike}} \leftarrow \begin{cases} \text{upper\_threshold}[\tau + d_1] - X[\tau + d_1] & \text{if RANDOM}() < 0.5 \\ \text{lower\_threshold}[\tau + d_1] - X[\tau + d_1] & \text{otherwise} \end{cases}$
21:    **else**
22:       $h_\tau^{\text{spike}} \leftarrow \text{upper\_threshold}[\tau + d_1] - X[\tau + d_1]$
23:    **end if**
24:    $\beta \leftarrow 10^{-4}$                              ▷ Sharpness parameter
25:    $\varepsilon \leftarrow h_\tau^{\text{spike}} \cdot \begin{cases} \exp\left(-\frac{\ln(\beta)}{d_1} \cdot (s - \tau - d_1)\right) & \text{if } s < \tau + d_1 \\ \exp\left(\frac{\ln(\beta)}{d_2} \cdot (s - \tau - d_1)\right) & \text{otherwise} \end{cases}$
26:    $\varepsilon^{\text{spike}}[\tau : \min(\tau + d_1 + d_2, T)] \leftarrow \varepsilon$
27: **end for**
    **Step 3. Generate corrupted time series**
28: $X^{\text{shift}} \leftarrow X + \varepsilon^{\text{shift}}$
29: $X^{\text{spike}} \leftarrow X + \varepsilon^{\text{spike}}$
30: $\varepsilon^{\text{combined}}(t) \leftarrow \begin{cases} \varepsilon^{\text{shift}}(t) & \text{if } |\varepsilon^{\text{shift}}(t)| > |\varepsilon^{\text{spike}}(t)| \\ \varepsilon^{\text{spike}}(t) & \text{otherwise} \end{cases} \quad \forall t \in [1, T]$
31: $X^{\text{combined}} \leftarrow X + \varepsilon^{\text{combined}}$
32: **return** $X^{\text{shift}}, X^{\text{spike}}, X^{\text{combined}}$

---

impact of each configuration. Figure 8(a) illustrates the "Severity Spectrum," where we sorted all configurations based on their average impact on model performance (Normalized MSE) across all datasets. This aggregated spectrum reveals the full range of difficulty levels our noise model can generate, independent of specific dataset characteristics.

From this continuous spectrum, we identified five representative severity levels (L1–L5) based on the following criteria:

- **Comprehensive Coverage:** The selected levels are distributed across the spectrum to cover distinct difficulty tiers, ensuring that the benchmark evaluates robustness under diverse conditions ranging from mild to extreme.
- **Physically Interpretable Evolution:** As shown in Figure 8(b), we ensured that the chosen levels correspond to a consistent physical intensification of the noise. Specifically, the injection frequency ($\lambda$) strictly increases from 0.002 to 0.008, and the transition probability ($q$) decreases logarithmically. This monotonicity guarantees that the degradation in model performance is a direct result of the progressively challenging noise mechanics.

Table 1 summarizes the final parameter sets derived from this process, which are used for all subsequent robustness evaluations. Also, we provide the results of these experiments on the ETTm1 dataset in Table 20, 21, and 22.

Table 20: Performance comparison between PatchLGA and PatchTST on the ETTm1 dataset with both level shift and spike corruptions, using input length 512. PatchLGA replaces standard self-attention with LGA, consistently showing improved robustness across all scenarios.

| Combined | | | PatchLGA | | | | | | PatchTST | | | | | |
|---|---|---|---|---|---|---|---|---|---|---|---|---|---|---|
| $\lambda$ | | | 0.002 | | 0.004 | | 0.008 | | 0.002 | | 0.004 | | 0.008 | |
| $H$ | $p$ | $q$ | MSE | MAE | MSE | MAE | MSE | MAE | MSE | MAE | MSE | MAE | MSE | MAE |
| 96 | 6 | $1.6 \cdot 10^{-4}$ | 0.297 | 0.353 | 0.305 | 0.361 | 0.326 | 0.379 | 0.304 | 0.357 | 0.316 | 0.370 | 0.336 | 0.387 |
| | | $4 \cdot 10^{-5}$ | 0.330 | 0.372 | 0.403 | 0.408 | 0.466 | 0.451 | 0.370 | 0.396 | 0.474 | 0.446 | 0.534 | 0.488 |
| | | $1 \cdot 10^{-5}$ | 0.346 | 0.381 | 0.441 | 0.425 | 0.502 | 0.469 | 0.399 | 0.408 | 0.540 | 0.470 | 0.590 | 0.510 |
| | 9 | $1.6 \cdot 10^{-4}$ | 0.302 | 0.356 | 0.314 | 0.368 | 0.344 | 0.393 | 0.309 | 0.361 | 0.326 | 0.377 | 0.353 | 0.401 |
| | | $4 \cdot 10^{-5}$ | 0.357 | 0.385 | 0.468 | 0.435 | 0.566 | 0.494 | 0.409 | 0.413 | 0.553 | 0.475 | 0.650 | 0.533 |
| | | $1 \cdot 10^{-5}$ | 0.383 | 0.397 | 0.529 | 0.458 | 0.620 | 0.519 | 0.452 | 0.429 | 0.647 | 0.505 | 0.727 | 0.562 |
| | 12 | $1.6 \cdot 10^{-4}$ | 0.305 | 0.359 | 0.320 | 0.373 | 0.357 | 0.404 | 0.312 | 0.364 | 0.331 | 0.381 | 0.366 | 0.410 |
| | | $4 \cdot 10^{-5}$ | 0.383 | 0.397 | 0.527 | 0.456 | 0.657 | 0.530 | 0.435 | 0.424 | 0.625 | 0.497 | 0.743 | 0.566 |
| | | $1 \cdot 10^{-5}$ | 0.420 | 0.412 | 0.611 | 0.487 | 0.732 | 0.561 | 0.490 | 0.444 | 0.741 | 0.533 | 0.844 | 0.602 |
| | 15 | $1.6 \cdot 10^{-4}$ | 0.307 | 0.361 | 0.324 | 0.377 | 0.366 | 0.413 | 0.314 | 0.365 | 0.333 | 0.384 | 0.374 | 0.418 |
| | | $4 \cdot 10^{-5}$ | 0.407 | 0.406 | 0.577 | 0.473 | 0.730 | 0.558 | 0.463 | 0.434 | 0.674 | 0.514 | 0.829 | 0.596 |
| | | $1 \cdot 10^{-5}$ | 0.455 | 0.425 | 0.681 | 0.510 | 0.820 | 0.594 | 0.530 | 0.457 | 0.809 | 0.554 | 0.944 | 0.636 |
| 192 | 6 | $1.6 \cdot 10^{-4}$ | 0.341 | 0.376 | 0.347 | 0.383 | 0.360 | 0.395 | 0.342 | 0.378 | 0.350 | 0.387 | 0.363 | 0.398 |
| | | $4 \cdot 10^{-5}$ | 0.365 | 0.393 | 0.416 | 0.422 | 0.459 | 0.452 | 0.381 | 0.403 | 0.459 | 0.443 | 0.489 | 0.468 |
| | | $1 \cdot 10^{-5}$ | 0.376 | 0.399 | 0.443 | 0.434 | 0.485 | 0.466 | 0.401 | 0.412 | 0.513 | 0.463 | 0.526 | 0.485 |
| | 9 | $1.6 \cdot 10^{-4}$ | 0.345 | 0.380 | 0.354 | 0.388 | 0.374 | 0.406 | 0.346 | 0.382 | 0.357 | 0.391 | 0.376 | 0.409 |
| | | $4 \cdot 10^{-5}$ | 0.384 | 0.403 | 0.462 | 0.443 | 0.531 | 0.488 | 0.410 | 0.417 | 0.522 | 0.469 | 0.579 | 0.508 |
| | | $1 \cdot 10^{-5}$ | 0.403 | 0.412 | 0.507 | 0.461 | 0.570 | 0.507 | 0.444 | 0.431 | 0.602 | 0.496 | 0.636 | 0.532 |
| | 12 | $1.6 \cdot 10^{-4}$ | 0.347 | 0.382 | 0.358 | 0.392 | 0.385 | 0.415 | 0.349 | 0.384 | 0.361 | 0.395 | 0.387 | 0.418 |
| | | $4 \cdot 10^{-5}$ | 0.403 | 0.412 | 0.504 | 0.462 | 0.603 | 0.521 | 0.434 | 0.427 | 0.581 | 0.491 | 0.661 | 0.542 |
| | | $1 \cdot 10^{-5}$ | 0.431 | 0.425 | 0.567 | 0.487 | 0.653 | 0.544 | 0.483 | 0.446 | 0.684 | 0.524 | 0.733 | 0.570 |
| | 15 | $1.6 \cdot 10^{-4}$ | 0.350 | 0.383 | 0.361 | 0.395 | 0.391 | 0.421 | 0.352 | 0.385 | 0.364 | 0.398 | 0.394 | 0.424 |
| | | $4 \cdot 10^{-5}$ | 0.420 | 0.420 | 0.544 | 0.479 | 0.663 | 0.547 | 0.456 | 0.437 | 0.627 | 0.508 | 0.730 | 0.570 |
| | | $1 \cdot 10^{-5}$ | 0.457 | 0.436 | 0.621 | 0.508 | 0.722 | 0.573 | 0.516 | 0.458 | 0.746 | 0.545 | 0.812 | 0.601 |
| 336 | 6 | $1.6 \cdot 10^{-4}$ | 0.372 | 0.392 | 0.380 | 0.398 | 0.389 | 0.408 | 0.370 | 0.397 | 0.378 | 0.404 | 0.390 | 0.414 |
| | | $4 \cdot 10^{-5}$ | 0.391 | 0.405 | 0.434 | 0.431 | 0.468 | 0.457 | 0.418 | 0.426 | 0.499 | 0.468 | 0.534 | 0.495 |
| | | $1 \cdot 10^{-5}$ | 0.399 | 0.410 | 0.455 | 0.441 | 0.489 | 0.469 | 0.434 | 0.433 | 0.554 | 0.487 | 0.555 | 0.505 |
| | 9 | $1.6 \cdot 10^{-4}$ | 0.376 | 0.394 | 0.385 | 0.402 | 0.401 | 0.417 | 0.374 | 0.400 | 0.383 | 0.408 | 0.402 | 0.424 |
| | | $4 \cdot 10^{-5}$ | 0.405 | 0.413 | 0.468 | 0.448 | 0.526 | 0.487 | 0.453 | 0.442 | 0.569 | 0.496 | 0.620 | 0.534 |
| | | $1 \cdot 10^{-5}$ | 0.419 | 0.421 | 0.502 | 0.463 | 0.558 | 0.505 | 0.483 | 0.454 | 0.648 | 0.521 | 0.662 | 0.551 |
| | 12 | $1.6 \cdot 10^{-4}$ | 0.378 | 0.396 | 0.388 | 0.405 | 0.411 | 0.426 | 0.377 | 0.402 | 0.387 | 0.411 | 0.412 | 0.432 |
| | | $4 \cdot 10^{-5}$ | 0.419 | 0.421 | 0.502 | 0.465 | 0.586 | 0.516 | 0.481 | 0.454 | 0.630 | 0.519 | 0.692 | 0.563 |
| | | $1 \cdot 10^{-5}$ | 0.438 | 0.431 | 0.548 | 0.485 | 0.627 | 0.537 | 0.522 | 0.471 | 0.727 | 0.548 | 0.749 | 0.585 |
| | 15 | $1.6 \cdot 10^{-4}$ | 0.380 | 0.398 | 0.391 | 0.407 | 0.418 | 0.431 | 0.380 | 0.404 | 0.390 | 0.413 | 0.418 | 0.437 |
| | | $4 \cdot 10^{-5}$ | 0.431 | 0.427 | 0.533 | 0.478 | 0.638 | 0.540 | 0.502 | 0.463 | 0.669 | 0.533 | 0.749 | 0.586 |
| | | $1 \cdot 10^{-5}$ | 0.456 | 0.439 | 0.590 | 0.503 | 0.686 | 0.564 | 0.553 | 0.483 | 0.777 | 0.565 | 0.818 | 0.611 |
| 720 | 6 | $1.6 \cdot 10^{-4}$ | 0.425 | 0.421 | 0.432 | 0.426 | 0.440 | 0.434 | 0.420 | 0.424 | 0.429 | 0.429 | 0.437 | 0.437 |
| | | $4 \cdot 10^{-5}$ | 0.442 | 0.433 | 0.482 | 0.457 | 0.513 | 0.481 | 0.447 | 0.440 | 0.504 | 0.471 | 0.536 | 0.495 |
| | | $1 \cdot 10^{-5}$ | 0.449 | 0.438 | 0.502 | 0.467 | 0.530 | 0.492 | 0.457 | 0.446 | 0.538 | 0.484 | 0.546 | 0.501 |
| | 9 | $1.6 \cdot 10^{-4}$ | 0.428 | 0.423 | 0.436 | 0.429 | 0.451 | 0.442 | 0.423 | 0.426 | 0.433 | 0.432 | 0.448 | 0.445 |
| | | $4 \cdot 10^{-5}$ | 0.454 | 0.440 | 0.513 | 0.474 | 0.568 | 0.509 | 0.469 | 0.451 | 0.559 | 0.495 | 0.612 | 0.530 |
| | | $1 \cdot 10^{-5}$ | 0.466 | 0.447 | 0.543 | 0.487 | 0.593 | 0.523 | 0.491 | 0.461 | 0.615 | 0.514 | 0.630 | 0.540 |
| | 12 | $1.6 \cdot 10^{-4}$ | 0.430 | 0.424 | 0.438 | 0.431 | 0.460 | 0.449 | 0.426 | 0.428 | 0.437 | 0.435 | 0.458 | 0.453 |
| | | $4 \cdot 10^{-5}$ | 0.466 | 0.447 | 0.543 | 0.488 | 0.623 | 0.536 | 0.492 | 0.462 | 0.622 | 0.519 | 0.684 | 0.562 |
| | | $1 \cdot 10^{-5}$ | 0.482 | 0.456 | 0.584 | 0.506 | 0.656 | 0.553 | 0.522 | 0.475 | 0.699 | 0.544 | 0.714 | 0.575 |
| | 15 | $1.6 \cdot 10^{-4}$ | 0.432 | 0.426 | 0.440 | 0.433 | 0.466 | 0.454 | 0.429 | 0.430 | 0.439 | 0.437 | 0.464 | 0.458 |
| | | $4 \cdot 10^{-5}$ | 0.477 | 0.452 | 0.570 | 0.500 | 0.670 | 0.558 | 0.511 | 0.470 | 0.661 | 0.534 | 0.744 | 0.586 |
| | | $1 \cdot 10^{-5}$ | 0.497 | 0.463 | 0.620 | 0.522 | 0.709 | 0.578 | 0.549 | 0.485 | 0.753 | 0.564 | 0.782 | 0.604 |

Table 21: Performance comparison between PatchLGA and PatchTST on the ETTm1 dataset with level shift corruptions, using input length 512. PatchLGA replaces standard self-attention with LGA, consistently showing improved robustness across all scenarios.

| Level Shift | | | PatchLGA | | | | | | PatchTST | | | | | |
|---|---|---|---|---|---|---|---|---|---|---|---|---|---|---|
| | | $\lambda$ | 0.002 | | 0.004 | | 0.008 | | 0.002 | | 0.004 | | 0.008 | |
| $H$ | $p$ | $q$ | MSE | MAE | MSE | MAE | MSE | MAE | MSE | MAE | MSE | MAE | MSE | MAE |
| 96 | 6 | $1.6 \cdot 10^{-4}$ | 0.297 | 0.352 | 0.303 | 0.360 | 0.325 | 0.378 | 0.304 | 0.356 | 0.316 | 0.369 | 0.336 | 0.387 |
| | | $4 \cdot 10^{-5}$ | 0.326 | 0.370 | 0.394 | 0.403 | 0.454 | 0.443 | 0.370 | 0.394 | 0.472 | 0.443 | 0.538 | 0.488 |
| | | $1 \cdot 10^{-5}$ | 0.341 | 0.377 | 0.430 | 0.418 | 0.491 | 0.463 | 0.399 | 0.407 | 0.544 | 0.470 | 0.597 | 0.512 |
| | 9 | $1.6 \cdot 10^{-4}$ | 0.300 | 0.355 | 0.312 | 0.366 | 0.342 | 0.391 | 0.308 | 0.360 | 0.325 | 0.376 | 0.353 | 0.400 |
| | | $4 \cdot 10^{-5}$ | 0.353 | 0.382 | 0.455 | 0.427 | 0.552 | 0.484 | 0.407 | 0.411 | 0.547 | 0.470 | 0.653 | 0.531 |
| | | $1 \cdot 10^{-5}$ | 0.377 | 0.393 | 0.512 | 0.450 | 0.607 | 0.511 | 0.449 | 0.426 | 0.644 | 0.502 | 0.737 | 0.563 |
| | 12 | $1.6 \cdot 10^{-4}$ | 0.303 | 0.357 | 0.317 | 0.370 | 0.354 | 0.401 | 0.311 | 0.363 | 0.329 | 0.379 | 0.365 | 0.409 |
| | | $4 \cdot 10^{-5}$ | 0.378 | 0.393 | 0.511 | 0.447 | 0.641 | 0.519 | 0.432 | 0.421 | 0.616 | 0.490 | 0.746 | 0.563 |
| | | $1 \cdot 10^{-5}$ | 0.413 | 0.407 | 0.592 | 0.477 | 0.716 | 0.552 | 0.485 | 0.440 | 0.737 | 0.528 | 0.853 | 0.601 |
| | 15 | $1.6 \cdot 10^{-4}$ | 0.305 | 0.359 | 0.320 | 0.373 | 0.363 | 0.409 | 0.313 | 0.364 | 0.331 | 0.381 | 0.374 | 0.416 |
| | | $4 \cdot 10^{-5}$ | 0.401 | 0.402 | 0.560 | 0.463 | 0.707 | 0.545 | 0.459 | 0.430 | 0.664 | 0.504 | 0.830 | 0.592 |
| | | $1 \cdot 10^{-5}$ | 0.446 | 0.419 | 0.661 | 0.499 | 0.802 | 0.584 | 0.523 | 0.451 | 0.807 | 0.547 | 0.954 | 0.634 |
| 192 | 6 | $1.6 \cdot 10^{-4}$ | 0.340 | 0.376 | 0.345 | 0.382 | 0.358 | 0.393 | 0.341 | 0.378 | 0.349 | 0.386 | 0.363 | 0.398 |
| | | $4 \cdot 10^{-5}$ | 0.361 | 0.390 | 0.408 | 0.417 | 0.448 | 0.445 | 0.380 | 0.401 | 0.457 | 0.440 | 0.487 | 0.466 |
| | | $1 \cdot 10^{-5}$ | 0.372 | 0.395 | 0.435 | 0.428 | 0.475 | 0.460 | 0.400 | 0.411 | 0.516 | 0.463 | 0.531 | 0.486 |
| | 9 | $1.6 \cdot 10^{-4}$ | 0.344 | 0.379 | 0.351 | 0.386 | 0.371 | 0.403 | 0.345 | 0.381 | 0.355 | 0.390 | 0.376 | 0.408 |
| | | $4 \cdot 10^{-5}$ | 0.380 | 0.399 | 0.452 | 0.437 | 0.518 | 0.480 | 0.409 | 0.415 | 0.517 | 0.465 | 0.578 | 0.506 |
| | | $1 \cdot 10^{-5}$ | 0.398 | 0.408 | 0.495 | 0.454 | 0.559 | 0.500 | 0.442 | 0.428 | 0.602 | 0.494 | 0.645 | 0.533 |
| | 12 | $1.6 \cdot 10^{-4}$ | 0.346 | 0.380 | 0.356 | 0.390 | 0.382 | 0.412 | 0.348 | 0.383 | 0.359 | 0.394 | 0.386 | 0.417 |
| | | $4 \cdot 10^{-5}$ | 0.398 | 0.408 | 0.491 | 0.455 | 0.587 | 0.511 | 0.432 | 0.425 | 0.574 | 0.486 | 0.657 | 0.538 |
| | | $1 \cdot 10^{-5}$ | 0.424 | 0.420 | 0.552 | 0.479 | 0.639 | 0.535 | 0.480 | 0.442 | 0.684 | 0.522 | 0.742 | 0.570 |
| | 15 | $1.6 \cdot 10^{-4}$ | 0.349 | 0.382 | 0.358 | 0.393 | 0.389 | 0.418 | 0.351 | 0.384 | 0.362 | 0.396 | 0.394 | 0.423 |
| | | $4 \cdot 10^{-5}$ | 0.415 | 0.415 | 0.530 | 0.470 | 0.643 | 0.536 | 0.454 | 0.434 | 0.618 | 0.501 | 0.725 | 0.565 |
| | | $1 \cdot 10^{-5}$ | 0.449 | 0.430 | 0.605 | 0.499 | 0.707 | 0.564 | 0.512 | 0.454 | 0.746 | 0.541 | 0.823 | 0.601 |
| 336 | 6 | $1.6 \cdot 10^{-4}$ | 0.372 | 0.391 | 0.378 | 0.397 | 0.387 | 0.406 | 0.370 | 0.397 | 0.377 | 0.404 | 0.390 | 0.414 |
| | | $4 \cdot 10^{-5}$ | 0.388 | 0.403 | 0.427 | 0.426 | 0.457 | 0.449 | 0.416 | 0.424 | 0.499 | 0.466 | 0.532 | 0.493 |
| | | $1 \cdot 10^{-5}$ | 0.395 | 0.407 | 0.446 | 0.436 | 0.479 | 0.462 | 0.432 | 0.431 | 0.561 | 0.488 | 0.560 | 0.505 |
| | 9 | $1.6 \cdot 10^{-4}$ | 0.375 | 0.394 | 0.383 | 0.400 | 0.399 | 0.415 | 0.374 | 0.400 | 0.382 | 0.407 | 0.402 | 0.424 |
| | | $4 \cdot 10^{-5}$ | 0.402 | 0.410 | 0.459 | 0.443 | 0.513 | 0.478 | 0.452 | 0.440 | 0.566 | 0.492 | 0.618 | 0.531 |
| | | $1 \cdot 10^{-5}$ | 0.414 | 0.417 | 0.490 | 0.457 | 0.546 | 0.496 | 0.481 | 0.452 | 0.652 | 0.521 | 0.673 | 0.552 |
| | 12 | $1.6 \cdot 10^{-4}$ | 0.378 | 0.395 | 0.386 | 0.403 | 0.409 | 0.423 | 0.377 | 0.402 | 0.386 | 0.410 | 0.412 | 0.431 |
| | | $4 \cdot 10^{-5}$ | 0.415 | 0.417 | 0.490 | 0.458 | 0.570 | 0.506 | 0.478 | 0.452 | 0.624 | 0.514 | 0.688 | 0.559 |
| | | $1 \cdot 10^{-5}$ | 0.432 | 0.426 | 0.533 | 0.477 | 0.613 | 0.528 | 0.519 | 0.468 | 0.730 | 0.547 | 0.761 | 0.586 |
| | 15 | $1.6 \cdot 10^{-4}$ | 0.380 | 0.396 | 0.388 | 0.405 | 0.415 | 0.429 | 0.380 | 0.404 | 0.388 | 0.412 | 0.419 | 0.437 |
| | | $4 \cdot 10^{-5}$ | 0.426 | 0.423 | 0.519 | 0.471 | 0.619 | 0.529 | 0.500 | 0.461 | 0.660 | 0.526 | 0.745 | 0.582 |
| | | $1 \cdot 10^{-5}$ | 0.448 | 0.434 | 0.573 | 0.494 | 0.670 | 0.554 | 0.550 | 0.480 | 0.778 | 0.563 | 0.831 | 0.612 |
| 720 | 6 | $1.6 \cdot 10^{-4}$ | 0.425 | 0.420 | 0.431 | 0.425 | 0.438 | 0.432 | 0.420 | 0.424 | 0.428 | 0.428 | 0.435 | 0.436 |
| | | $4 \cdot 10^{-5}$ | 0.439 | 0.430 | 0.476 | 0.453 | 0.503 | 0.473 | 0.444 | 0.438 | 0.500 | 0.467 | 0.527 | 0.489 |
| | | $1 \cdot 10^{-5}$ | 0.445 | 0.434 | 0.494 | 0.462 | 0.520 | 0.484 | 0.453 | 0.443 | 0.537 | 0.481 | 0.541 | 0.496 |
| | 9 | $1.6 \cdot 10^{-4}$ | 0.427 | 0.422 | 0.435 | 0.428 | 0.449 | 0.440 | 0.423 | 0.426 | 0.432 | 0.431 | 0.447 | 0.444 |
| | | $4 \cdot 10^{-5}$ | 0.451 | 0.437 | 0.506 | 0.468 | 0.556 | 0.501 | 0.466 | 0.449 | 0.555 | 0.491 | 0.603 | 0.523 |
| | | $1 \cdot 10^{-5}$ | 0.461 | 0.443 | 0.533 | 0.481 | 0.582 | 0.515 | 0.487 | 0.458 | 0.614 | 0.511 | 0.630 | 0.536 |
| | 12 | $1.6 \cdot 10^{-4}$ | 0.430 | 0.423 | 0.437 | 0.430 | 0.458 | 0.447 | 0.426 | 0.428 | 0.436 | 0.434 | 0.457 | 0.452 |
| | | $4 \cdot 10^{-5}$ | 0.462 | 0.443 | 0.533 | 0.482 | 0.609 | 0.526 | 0.488 | 0.459 | 0.616 | 0.514 | 0.673 | 0.554 |
| | | $1 \cdot 10^{-5}$ | 0.476 | 0.451 | 0.571 | 0.499 | 0.643 | 0.544 | 0.518 | 0.471 | 0.699 | 0.542 | 0.714 | 0.572 |
| | 15 | $1.6 \cdot 10^{-4}$ | 0.431 | 0.425 | 0.438 | 0.431 | 0.464 | 0.452 | 0.429 | 0.429 | 0.437 | 0.435 | 0.464 | 0.457 |
| | | $4 \cdot 10^{-5}$ | 0.471 | 0.448 | 0.558 | 0.493 | 0.652 | 0.547 | 0.507 | 0.467 | 0.653 | 0.528 | 0.731 | 0.579 |
| | | $1 \cdot 10^{-5}$ | 0.490 | 0.457 | 0.605 | 0.514 | 0.694 | 0.568 | 0.545 | 0.482 | 0.749 | 0.559 | 0.785 | 0.601 |

Table 22: Performance comparison between PatchLGA and PatchTST on the ETTm1 dataset with spike corruptions, using input length 512. PatchLGA replaces standard self-attention with LGA, consistently showing improved robustness across all scenarios.

| Spike | | | PatchLGA | | | | | | PatchTST | | | | | |
|---|---|---|---|---|---|---|---|---|---|---|---|---|---|---|
| | | $\lambda$ | 0.002 | | 0.004 | | 0.008 | | 0.002 | | 0.004 | | 0.008 | |
| $H$ | $p$ | $q$ | MSE | MAE | MSE | MAE | MSE | MAE | MSE | MAE | MSE | MAE | MSE | MAE |
| 96 | 6 | $1.6 \cdot 10^{-4}$ | 0.289 | 0.345 | 0.289 | 0.345 | 0.290 | 0.347 | 0.293 | 0.346 | 0.293 | 0.347 | 0.294 | 0.348 |
| | | $4 \cdot 10^{-5}$ | 0.291 | 0.347 | 0.296 | 0.350 | 0.301 | 0.356 | 0.296 | 0.349 | 0.301 | 0.354 | 0.306 | 0.358 |
| | | $1 \cdot 10^{-5}$ | 0.293 | 0.348 | 0.297 | 0.351 | 0.302 | 0.356 | 0.297 | 0.350 | 0.301 | 0.354 | 0.306 | 0.359 |
| | 9 | $1.6 \cdot 10^{-4}$ | 0.289 | 0.345 | 0.289 | 0.346 | 0.291 | 0.348 | 0.293 | 0.346 | 0.294 | 0.348 | 0.295 | 0.350 |
| | | $4 \cdot 10^{-5}$ | 0.292 | 0.348 | 0.298 | 0.352 | 0.305 | 0.359 | 0.298 | 0.351 | 0.305 | 0.357 | 0.311 | 0.363 |
| | | $1 \cdot 10^{-5}$ | 0.294 | 0.349 | 0.300 | 0.353 | 0.306 | 0.360 | 0.299 | 0.352 | 0.307 | 0.357 | 0.312 | 0.364 |
| | 12 | $1.6 \cdot 10^{-4}$ | 0.289 | 0.345 | 0.290 | 0.346 | 0.292 | 0.350 | 0.293 | 0.347 | 0.294 | 0.348 | 0.296 | 0.351 |
| | | $4 \cdot 10^{-5}$ | 0.293 | 0.349 | 0.302 | 0.354 | 0.309 | 0.363 | 0.301 | 0.353 | 0.310 | 0.361 | 0.318 | 0.369 |
| | | $1 \cdot 10^{-5}$ | 0.296 | 0.351 | 0.304 | 0.356 | 0.312 | 0.365 | 0.303 | 0.356 | 0.314 | 0.362 | 0.320 | 0.370 |
| | 15 | $1.6 \cdot 10^{-4}$ | 0.289 | 0.345 | 0.291 | 0.347 | 0.293 | 0.350 | 0.293 | 0.347 | 0.295 | 0.350 | 0.297 | 0.352 |
| | | $4 \cdot 10^{-5}$ | 0.294 | 0.349 | 0.304 | 0.357 | 0.314 | 0.366 | 0.303 | 0.355 | 0.316 | 0.365 | 0.324 | 0.374 |
| | | $1 \cdot 10^{-5}$ | 0.297 | 0.352 | 0.308 | 0.359 | 0.316 | 0.368 | 0.307 | 0.358 | 0.320 | 0.368 | 0.326 | 0.375 |
| 192 | 6 | $1.6 \cdot 10^{-4}$ | 0.333 | 0.370 | 0.334 | 0.371 | 0.335 | 0.372 | 0.335 | 0.372 | 0.335 | 0.372 | 0.336 | 0.373 |
| | | $4 \cdot 10^{-5}$ | 0.336 | 0.372 | 0.339 | 0.375 | 0.344 | 0.380 | 0.337 | 0.373 | 0.341 | 0.377 | 0.344 | 0.379 |
| | | $1 \cdot 10^{-5}$ | 0.337 | 0.373 | 0.340 | 0.376 | 0.345 | 0.380 | 0.338 | 0.374 | 0.342 | 0.377 | 0.345 | 0.380 |
| | 9 | $1.6 \cdot 10^{-4}$ | 0.333 | 0.370 | 0.334 | 0.371 | 0.335 | 0.373 | 0.335 | 0.372 | 0.336 | 0.373 | 0.336 | 0.374 |
| | | $4 \cdot 10^{-5}$ | 0.336 | 0.373 | 0.341 | 0.376 | 0.346 | 0.382 | 0.338 | 0.374 | 0.343 | 0.378 | 0.346 | 0.382 |
| | | $1 \cdot 10^{-5}$ | 0.338 | 0.374 | 0.342 | 0.377 | 0.347 | 0.383 | 0.339 | 0.375 | 0.344 | 0.379 | 0.347 | 0.383 |
| | 12 | $1.6 \cdot 10^{-4}$ | 0.333 | 0.371 | 0.335 | 0.372 | 0.335 | 0.373 | 0.335 | 0.372 | 0.336 | 0.373 | 0.337 | 0.374 |
| | | $4 \cdot 10^{-5}$ | 0.337 | 0.374 | 0.343 | 0.378 | 0.348 | 0.384 | 0.339 | 0.375 | 0.346 | 0.380 | 0.349 | 0.385 |
| | | $1 \cdot 10^{-5}$ | 0.339 | 0.375 | 0.345 | 0.379 | 0.350 | 0.385 | 0.341 | 0.377 | 0.348 | 0.382 | 0.351 | 0.386 |
| | 15 | $1.6 \cdot 10^{-4}$ | 0.333 | 0.371 | 0.335 | 0.372 | 0.335 | 0.373 | 0.335 | 0.372 | 0.337 | 0.374 | 0.337 | 0.375 |
| | | $4 \cdot 10^{-5}$ | 0.338 | 0.375 | 0.345 | 0.380 | 0.351 | 0.386 | 0.341 | 0.376 | 0.349 | 0.383 | 0.353 | 0.387 |
| | | $1 \cdot 10^{-5}$ | 0.341 | 0.376 | 0.347 | 0.382 | 0.353 | 0.387 | 0.344 | 0.379 | 0.352 | 0.385 | 0.354 | 0.389 |
| 336 | 6 | $1.6 \cdot 10^{-4}$ | 0.366 | 0.387 | 0.367 | 0.388 | 0.368 | 0.389 | 0.364 | 0.392 | 0.364 | 0.392 | 0.365 | 0.393 |
| | | $4 \cdot 10^{-5}$ | 0.369 | 0.389 | 0.372 | 0.392 | 0.376 | 0.396 | 0.366 | 0.394 | 0.370 | 0.396 | 0.373 | 0.399 |
| | | $1 \cdot 10^{-5}$ | 0.370 | 0.390 | 0.373 | 0.393 | 0.378 | 0.397 | 0.367 | 0.395 | 0.371 | 0.397 | 0.374 | 0.400 |
| | 9 | $1.6 \cdot 10^{-4}$ | 0.367 | 0.387 | 0.368 | 0.388 | 0.369 | 0.390 | 0.364 | 0.393 | 0.365 | 0.393 | 0.366 | 0.394 |
| | | $4 \cdot 10^{-5}$ | 0.370 | 0.390 | 0.373 | 0.392 | 0.378 | 0.397 | 0.367 | 0.395 | 0.372 | 0.398 | 0.376 | 0.402 |
| | | $1 \cdot 10^{-5}$ | 0.371 | 0.391 | 0.375 | 0.394 | 0.380 | 0.399 | 0.369 | 0.396 | 0.373 | 0.398 | 0.376 | 0.402 |
| | 12 | $1.6 \cdot 10^{-4}$ | 0.367 | 0.387 | 0.368 | 0.388 | 0.369 | 0.390 | 0.365 | 0.393 | 0.365 | 0.393 | 0.366 | 0.394 |
| | | $4 \cdot 10^{-5}$ | 0.371 | 0.390 | 0.375 | 0.394 | 0.379 | 0.399 | 0.368 | 0.396 | 0.375 | 0.400 | 0.379 | 0.405 |
| | | $1 \cdot 10^{-5}$ | 0.372 | 0.392 | 0.377 | 0.395 | 0.381 | 0.401 | 0.371 | 0.398 | 0.377 | 0.401 | 0.379 | 0.405 |
| | 15 | $1.6 \cdot 10^{-4}$ | 0.367 | 0.388 | 0.369 | 0.389 | 0.368 | 0.390 | 0.365 | 0.393 | 0.366 | 0.394 | 0.365 | 0.394 |
| | | $4 \cdot 10^{-5}$ | 0.372 | 0.391 | 0.377 | 0.395 | 0.381 | 0.400 | 0.370 | 0.396 | 0.378 | 0.403 | 0.382 | 0.408 |
| | | $1 \cdot 10^{-5}$ | 0.374 | 0.393 | 0.379 | 0.397 | 0.383 | 0.402 | 0.373 | 0.399 | 0.381 | 0.404 | 0.382 | 0.408 |
| 720 | 6 | $1.6 \cdot 10^{-4}$ | 0.420 | 0.417 | 0.421 | 0.417 | 0.422 | 0.418 | 0.416 | 0.421 | 0.416 | 0.421 | 0.417 | 0.421 |
| | | $4 \cdot 10^{-5}$ | 0.423 | 0.419 | 0.425 | 0.421 | 0.429 | 0.425 | 0.419 | 0.422 | 0.422 | 0.424 | 0.427 | 0.427 |
| | | $1 \cdot 10^{-5}$ | 0.424 | 0.420 | 0.426 | 0.422 | 0.431 | 0.426 | 0.420 | 0.423 | 0.424 | 0.425 | 0.429 | 0.429 |
| | 9 | $1.6 \cdot 10^{-4}$ | 0.420 | 0.417 | 0.421 | 0.417 | 0.422 | 0.419 | 0.416 | 0.421 | 0.416 | 0.421 | 0.418 | 0.422 |
| | | $4 \cdot 10^{-5}$ | 0.423 | 0.419 | 0.426 | 0.421 | 0.430 | 0.426 | 0.419 | 0.423 | 0.423 | 0.425 | 0.428 | 0.428 |
| | | $1 \cdot 10^{-5}$ | 0.425 | 0.420 | 0.428 | 0.423 | 0.432 | 0.428 | 0.421 | 0.424 | 0.425 | 0.426 | 0.431 | 0.430 |
| | 12 | $1.6 \cdot 10^{-4}$ | 0.420 | 0.417 | 0.421 | 0.417 | 0.422 | 0.419 | 0.416 | 0.421 | 0.417 | 0.421 | 0.418 | 0.422 |
| | | $4 \cdot 10^{-5}$ | 0.424 | 0.420 | 0.427 | 0.423 | 0.431 | 0.427 | 0.420 | 0.423 | 0.425 | 0.426 | 0.430 | 0.430 |
| | | $1 \cdot 10^{-5}$ | 0.426 | 0.421 | 0.429 | 0.424 | 0.434 | 0.429 | 0.422 | 0.425 | 0.427 | 0.428 | 0.433 | 0.432 |
| | 15 | $1.6 \cdot 10^{-4}$ | 0.421 | 0.417 | 0.422 | 0.418 | 0.422 | 0.419 | 0.416 | 0.421 | 0.417 | 0.421 | 0.418 | 0.422 |
| | | $4 \cdot 10^{-5}$ | 0.425 | 0.420 | 0.429 | 0.424 | 0.433 | 0.428 | 0.421 | 0.424 | 0.427 | 0.428 | 0.432 | 0.431 |
| | | $1 \cdot 10^{-5}$ | 0.427 | 0.422 | 0.432 | 0.426 | 0.435 | 0.430 | 0.423 | 0.426 | 0.430 | 0.430 | 0.435 | 0.433 |

# G    VISUALIZATION OF TSRBENCH

To provide a comprehensive visual understanding of TSRBench, we present a series of visualizations that illustrate the effects of different corruption types across various severity levels and datasets. These visualizations serve as a qualitative complement to the quantitative results presented in the main paper. Figures 9, 10, 11, 12, 13, and 14 display examples of time series data from our six benchmark datasets (ETTh1, ETTh2, ETTm1, ETTm2, Weather, and Electricity) under realistic corruptions at varying severity levels. Each figure shows level shift corruptions (left column) and spike corruptions (right column), demonstrating how these corruptions manifest differently across diverse time series data types.

As the severity level increases from 1 to 5, we can observe the progressive intensification of both corruption types. For spike corruptions, higher severity levels not only produce spikes with greater amplitudes but also increase their frequency throughout the time series. This creates challenging scenarios where models must distinguish between legitimate data points and anomalous spikes that occur more frequently and with larger magnitudes. Level shift corruptions, meanwhile, exhibit two key patterns as severity increases: first, the magnitude of the shifts becomes more pronounced, creating larger deviations from the original signal; second, the duration of these shifts becomes notably wider, meaning the corrupted signal remains in an altered state for longer periods. This temporal extension of corruption is particularly challenging for forecasting models that rely on consistent patterns.

These visualizations highlight the statistically grounded nature of our corruption generation process. Rather than arbitrary or manual corruption placement, TSRBench simulates realistic corruptions that preserve the underlying data distribution while introducing controlled perturbations. This approach allows for systematic evaluation of model robustness under conditions that closely resemble real-world scenarios where data quality cannot be guaranteed. The progressive severity scale enables researchers to assess not only whether models are robust to corruptions but also to quantify at which corruption intensity their performance begins to degrade significantly, providing valuable insights for deploying these models in practical applications.

# H    USE OF LARGE LANGUAGE MODELS

In the preparation of this paper, a Large Language Model (LLM) was utilized as a general-purpose writing-assistance tool. The role of the LLM was limited to improving the quality of the prose, including enhancing clarity, correcting grammatical errors, and refining sentence structure to ensure the manuscript was articulate and readable.

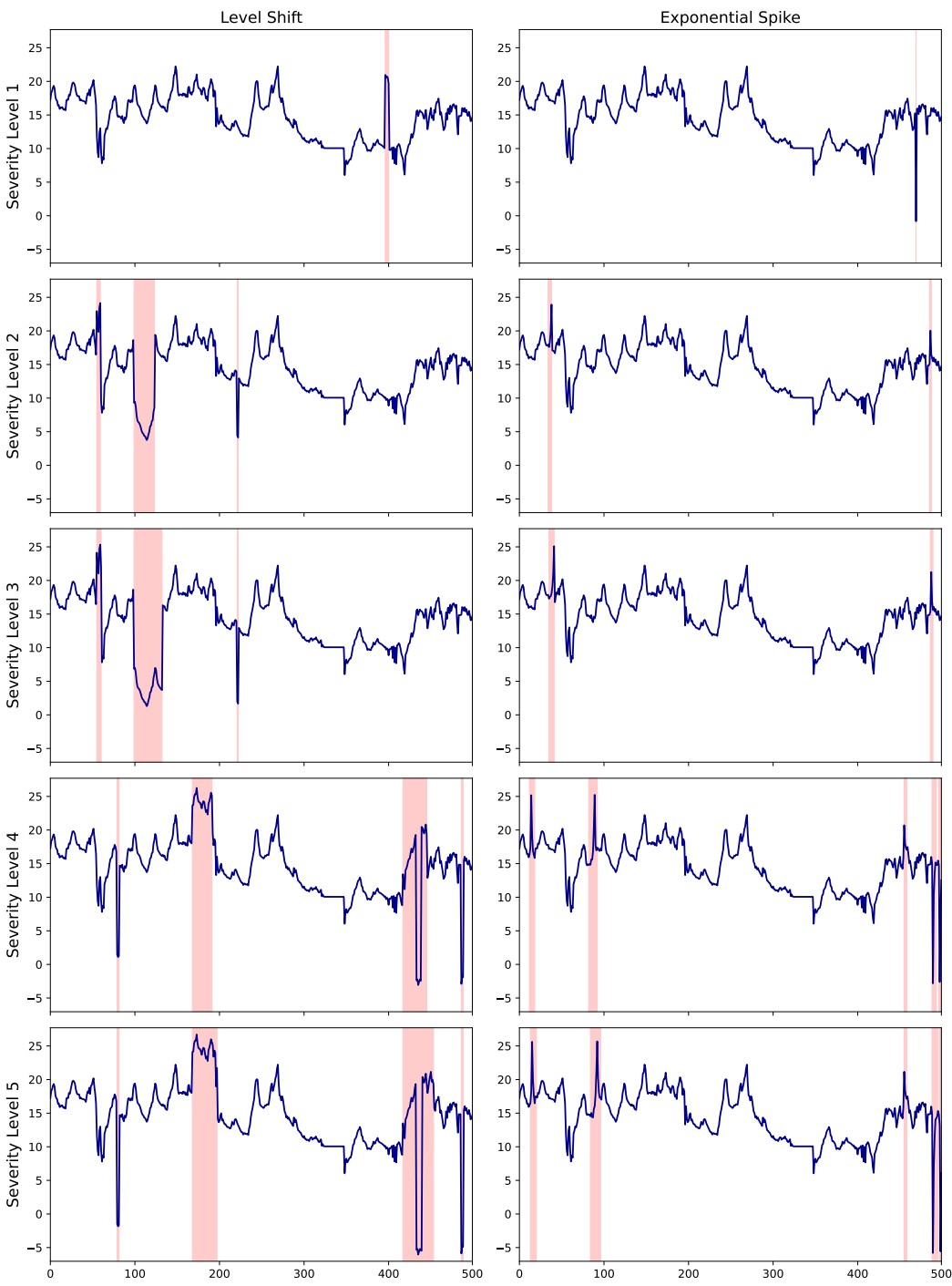

Figure 9: ETTh1 dataset with level shift corruptions (left) and spike corruptions (right) across severity levels 1-5. Each row represents a different severity level, demonstrating the progressive intensification of realistic corruptions in the time series data.

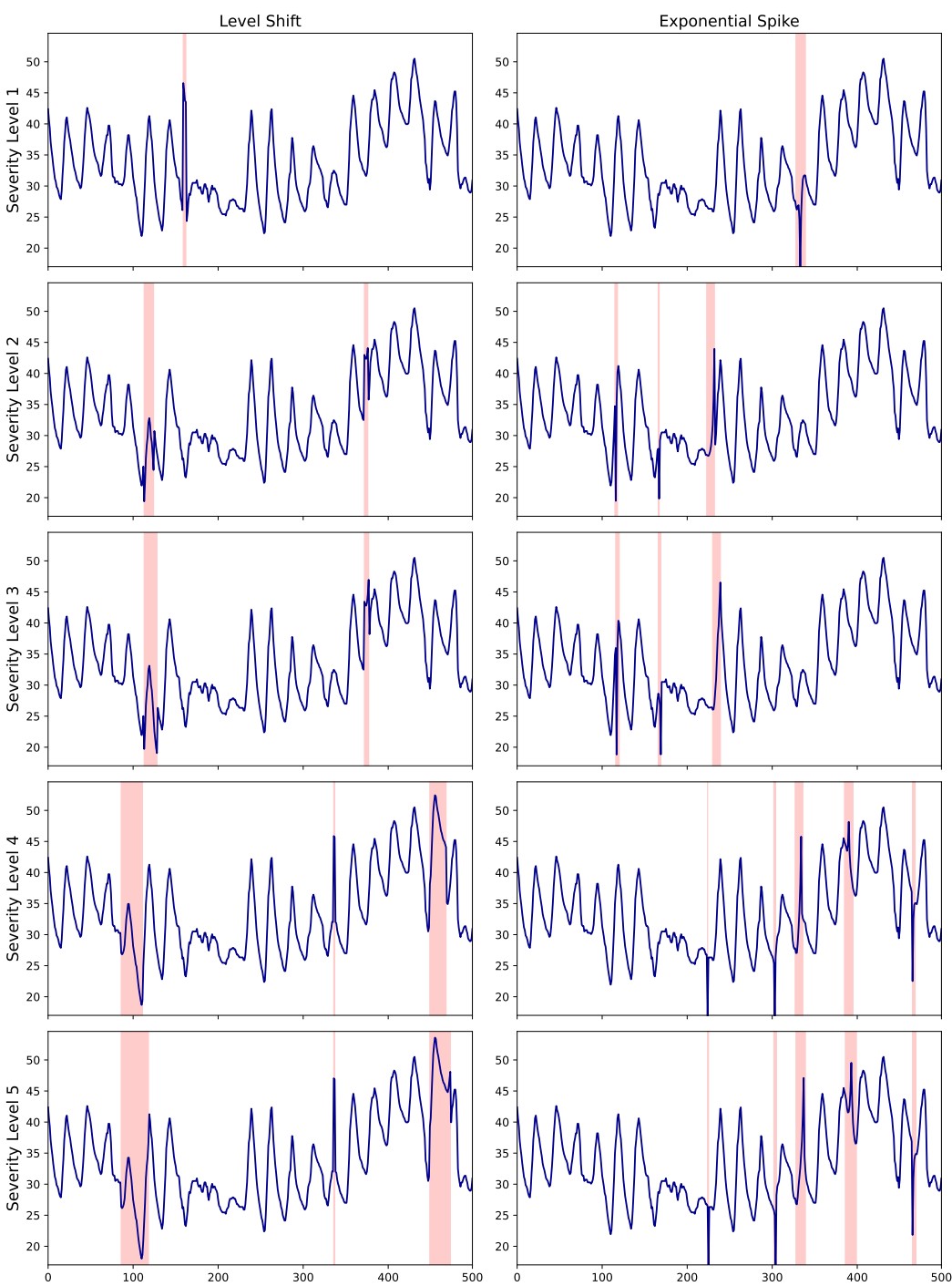

Figure 10: ETTh2 dataset with level shift corruptions (left) and spike corruptions (right) across severity levels 1-5. Each row represents a different severity level, demonstrating the progressive intensification of realistic corruptions in the time series data.

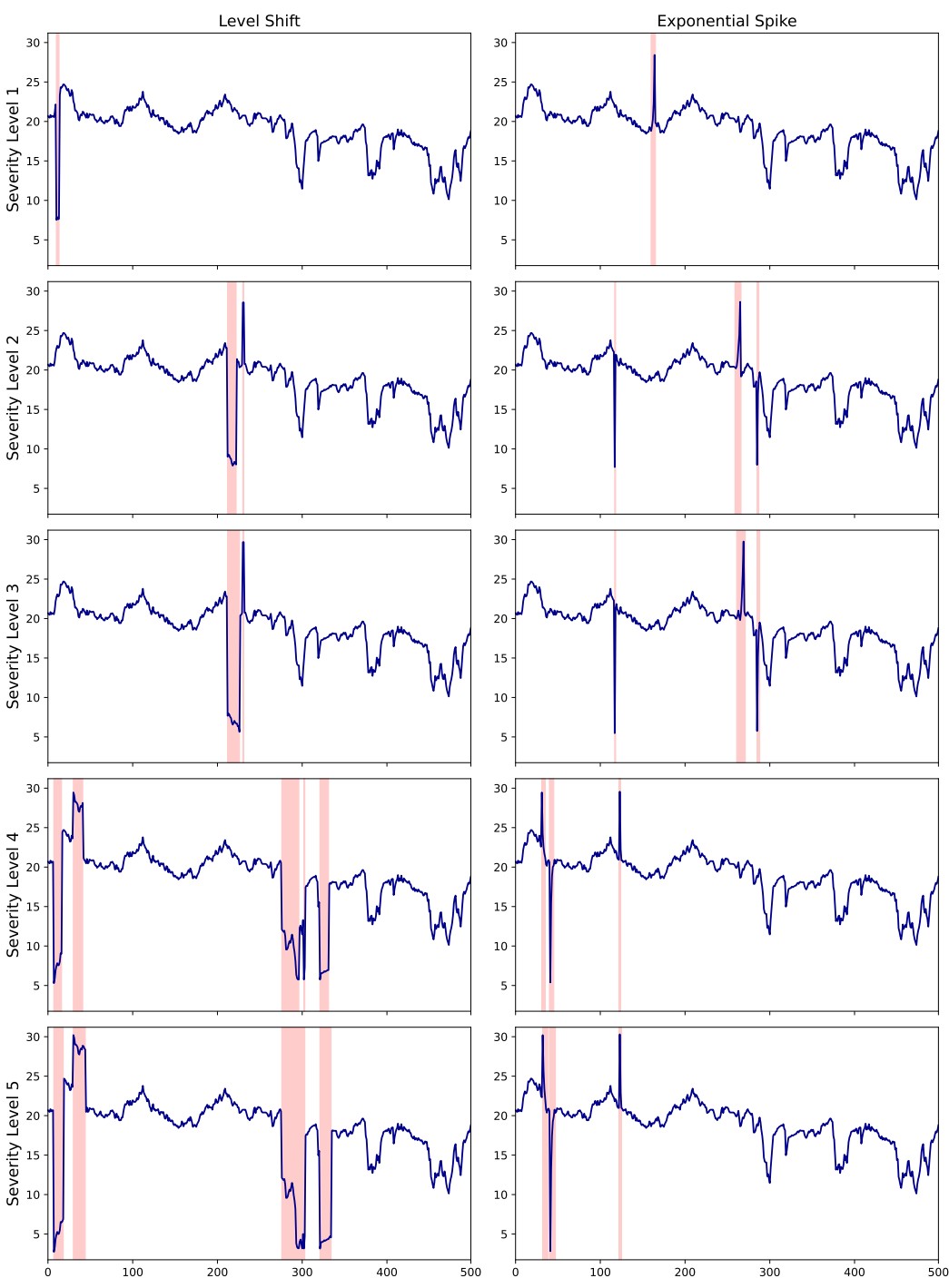

Figure 11: ETTm1 dataset with level shift corruptions (left) and spike corruptions (right) across severity levels 1-5. Each row represents a different severity level, demonstrating the progressive intensification of realistic corruptions in the time series data.

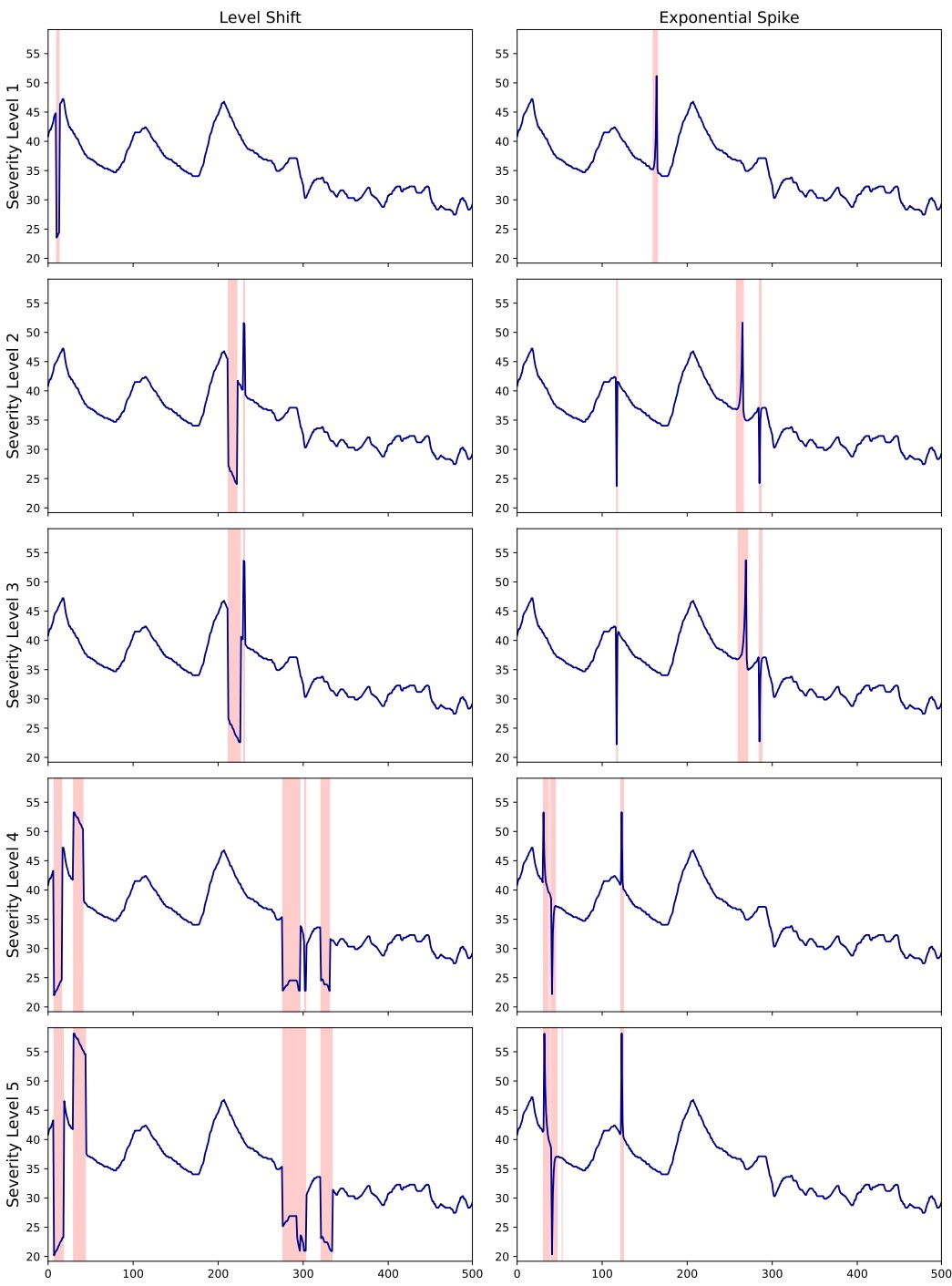

Figure 12: ETTm2 dataset with level shift corruptions (left) and spike corruptions (right) across severity levels 1-5. Each row represents a different severity level, demonstrating the progressive intensification of realistic corruptions in the time series data.

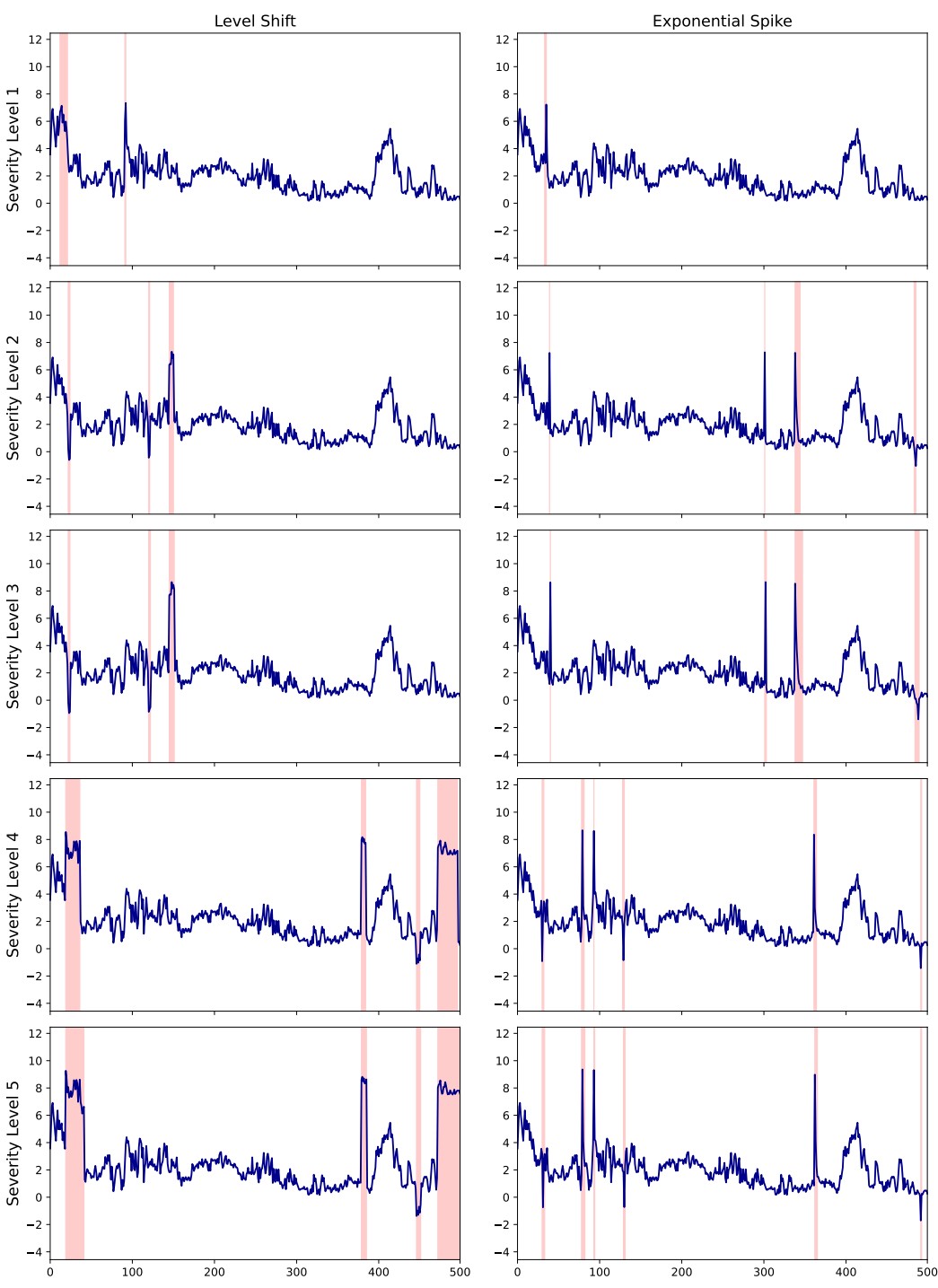

Figure 13: Weather dataset with level shift corruptions (left) and spike corruptions (right) across severity levels 1-5. Each row represents a different severity level, demonstrating the progressive intensification of realistic corruptions in the time series data.

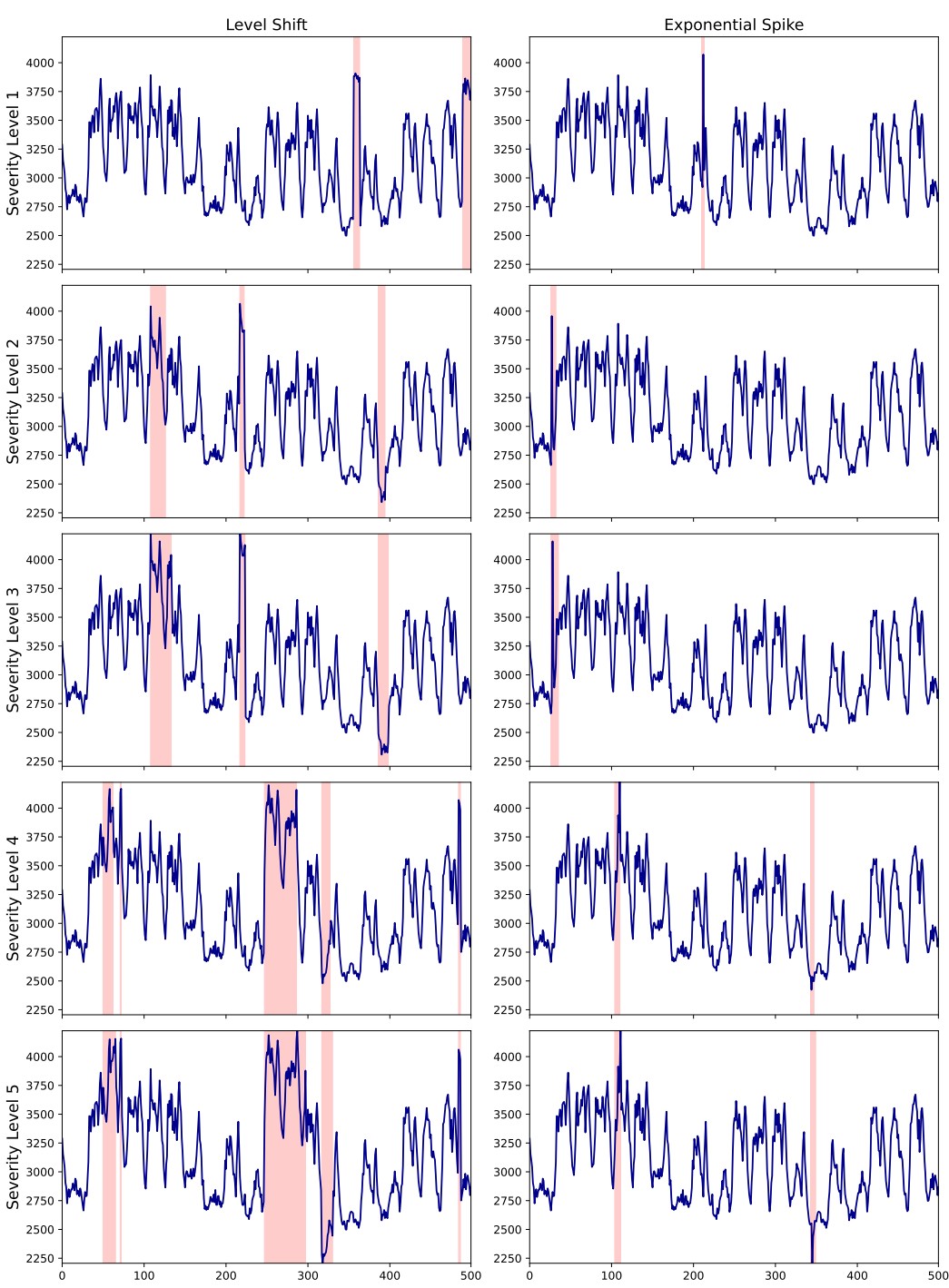

Figure 14: Electricity dataset with level shift corruptions (left) and spike corruptions (right) across severity levels 1-5. Each row represents a different severity level, demonstrating the progressive intensification of realistic corruptions in the time series data.

