# OpenReview forum: "Local Geometry Attention for Time Series Forecasting under Realistic Corruptions"
_ICLR.cc/2026/Conference — ICLR 2026 Poster_

### Official Review · Reviewer_9J2N · 2025-10-27

**Soundness:** 2
**Presentation:** 3
**Contribution:** 2
**Rating:** 4
**Confidence:** 4

**Summary:**

This paper proposes a new attention mechanism for time series forecasting, which is termed Local Geometry Attention. The designed LGA is inspired by local Gaussian process theory and aims to learning query-specific distance metrics and enable the model to learn complex temporal dependencies and enhance resilience to noise. Also, TSRBench, a comprehensive benchmark for evaluating time series forecasting robustness, is proposed.

**Strengths:**

1. The motivation seems clear with intuitive figures (Fig.1 and 2)

2. Clearly written and well-presented with source codes.

**Weaknesses:**

1. The technical details could be improved in a clearer way.

2. According to the performance reported in Table 2, the improvements led by the proposed method seem to be marginal.

3. the motivation of TSRBench is not clear. And more comprehensive datasets and baselines should be involved.

**Questions:**

Please refer to the weakness.

---

> ### Author Response · Authors · 2025-11-22
>
> We sincerely thank the reviewer for the constructive feedback. We are encouraged that you found our motivation clear and the paper well-presented. We acknowledge your concerns regarding technical details, the magnitude of performance improvements, and the motivation behind TSRBench.
>
> Below, we provide a detailed response addressing each of the weaknesses [W#] raised:
>
> ---
>
> **[W1] Clarification of Technical Details**
>
> We sincerely thank the reviewer for pointing out the need for clearer technical descriptions. We agree that a precise explanation of the algorithm and its computational implications is essential. In the revised manuscript, we have significantly reinforced the technical details in the following three aspects:
>
> **1. Explicit Tensor Shapes in Algorithm (Appendix B)**
>
> We acknowledged that the original description might have been abstract regarding data flow. To address this, we have updated **Algorithm 1 in Appendix B** to explicitly denote the exact **tensor shapes** at every step of the computation (e.g., Query $Q \in \mathbb{R}^{B \times L \times D}$, Metric Tensor $G \in \mathbb{R}^{B \times H \times L \times D_h}$). This ensures that the transformation from input embeddings to the final attention scores is transparent and reproducible.
>
> **2. Detailed Computational Complexity Analysis (Appendix B.3)**
>
> To clarify the cost of our proposed mechanism, we have added a rigorous **FLOPs analysis** comparing Standard Self-Attention (SDP) and LGA. We quantify the overhead for both inference and training phases, demonstrating that while LGA introduces additional operations for metric prediction, the relative overhead on the full Transformer block is manageable (~11% for inference).
>
> **3. Justification of Metric Tensor Design (Appendix C)**
> We have added a new section, **Appendix C**, to clarify why we chose a diagonal metric over a full rank tensor. This section provides:
>
> - **Empirical Ablation:** Comparing LGA (Diagonal) vs. LGA (Full-Matrix).
> - **Theoretical Stability Analysis:** Explaining why learning a full metric tensor is optimization-unstable due to the spectral properties of time series embeddings.
> This justifies our architectural choices with both theoretical and empirical evidence.
>
> We believe these revisions provide a comprehensive technical blueprint of our method. However, if there are **specific components or derivations** that you find still unclear or lacking, please let us know. We would be more than happy to provide further clarifications to ensure the manuscript meets the highest standards of clarity.
>
> ---
>
> **[W2] Marginal Improvements on performance**
>
> We respectfully wish to clarify that the improvements achieved by LGA are **statistically significant and substantial**, particularly under the realistic corruptions that our method is explicitly designed to handle. We believe the perception of "marginal" improvement might arise from focusing on clean conditions. However, in the context of robustness—the core contribution of this work—the gains are decisive.
>
> **1. Substantial Gains under Severe Corruption**
>
> Our method shines most when data quality degrades. Quantitatively, under the highest noise level (Severity 5):
> - **ETTm1:** LGA reduces MSE from 0.839 (Standard Attention) to **0.734**, a **12.5% improvement**.
> - **Weather:** We observe a 7.5% reduction in MSE (0.491 $\to$ 0.454).
>
> In time series forecasting benchmarks, a double-digit percentage gain is widely considered significant rather than marginal.
> This strong performance has been recognized by other reviewers as well. **Reviewer ESKs** noted that *"LGA (+PatchTST) consistently achieves the best forecasting accuracy especially at large severity levels, indicating that LGA adds robustness,"* and **Reviewer WvGd** found the experiments *"convincing to show the performance improvement."*
>
> **2. Statistical Significance and Stability (New Multi-Seed Experiments)**
>
> To definitively prove that these improvements are not due to random chance , we conducted additional experiments using three random seeds (2021, 2022, 2023) for all datasets.
> - **Lower Variance:** As reported in the new **Appendix E.1**, LGA demonstrates significantly **lower standard deviation** compared to the baseline, especially at high severity levels.
> - **Consistent Superiority:** The performance gap remains statistically consistent across all seeds.
> This empirical evidence confirms that LGA acts as a robust geometric regularizer, consistently stabilizing the model against corruptions regardless of initialization.

---

> ### Author Response · Authors · 2025-11-22
>
> **[W3] Motivation of TSRBench and Comprehensive Evaluation**
>
> We sincerely thank the reviewer for their constructive feedback. We appreciate the opportunity to clarify the motivation behind TSRBench and have significantly expanded our evaluation with additional datasets and baselines to demonstrate the robustness and generalizability of LGA.
>
> **1. Clarification on the Motivation of TSRBench**
>
> We acknowledge the reviewer's comment regarding the motivation of TSRBench. The primary motivation stems from a critical gap in evaluating **forecasting robustness** that existing anomaly detection datasets cannot fill.
>
> - **The "Ground Truth" Dilemma in Real-World Data:** As detailed in **Appendix E.6**, using real-world datasets with natural anomalies (e.g., Numenta Anomaly Benchmark) for forecasting evaluation is scientifically problematic. To validly assess if a model is "robust" (i.e., ignores input noise and predicts the true underlying signal), the evaluation metric (MSE/MAE) must be calculated against a **clean ground truth**.
>
> - **The Necessity of Decoupled Evaluation:** In standard real-world datasets, if an anomaly occurs at time $t+1$, the ground truth at $t+1$ *is* the anomaly. If a robust model successfully ignores input noise and predicts the "normal" value, standard metrics would penalize it for having a high error against the anomalous ground truth.
>
> - **TSRBench’s Solution:** TSRBench solves this by rigorously separating the **corrupted input** (injected with statistically grounded spikes/shifts via Extreme Value Theory) from the **clean ground truth**. This allows us to precisely measure a model's ability to recover intrinsic patterns from noisy history, a controlled evaluation impossible with "in-the-wild" datasets where the underlying "clean" signal is unknown.
>
> **2. Evaluation on Comprehensive Datasets**
>
> We thank the reviewer for this important suggestion. While our main evaluation focuses on 6 standard forecasting benchmarks, we have conducted **additional validation** to demonstrate LGA's broad applicability:
>
> **Systematic Robustness Analysis under Controlled Signal Conditions** (Janßen et al. (2025) [1])
>
> To rigorously evaluate LGA under controlled conditions, we adopted the recent parameterizable robustness framework by Janßen et al. (2025), which isolates specific robustness factors.
>
> **Experimental Setup**:
>
> - **7 Frequency Bands**: Very Low to Very High (testing temporal scale sensitivity)
> - **12 Signal Variants per Band**: 3 waveforms (Sine, Square, Sawtooth) × 4 SNR levels
> - **5 Noise Types**:
>     - Signal-Independent: White, Brownian, Impulse
>     - Signal-Dependent: Trend-dependent, Seasonal-dependent
> - **Total Evaluation**: 84 distinct signal configurations
>
> | Frequency Band | PatchLGA | PatchTST | Improvement |
> | --- | --- | --- | --- |
> | Very High | 0.068 | 0.068 | = |
> | High | **0.097** | 0.101 | **4.0%** |
> | Mid-High | **0.109** | 0.112 | **2.7%** |
> | Mid | 0.089 | 0.089 | = |
> | Low-Mid | **0.082** | 0.083 | **1.2%** |
> | Low | **0.085** | 0.087 | **2.3%** |
> | Very Low | **0.134** | 0.139 | **3.6%** |
> | **Average** | **0.095** | **0.097** | **2.1%** |
>
> LGA consistently improves robustness across diverse frequency bands and noise types, with particularly notable gains in challenging conditions (Very Low and High frequency bands).
>
> We believe that the inclusion of these new benchmarks (Janßen et al.), combined with our existing extensive experiments (widely-used forecasting benchmarks and operational data (AWS, Appendix E.6)), constitutes a rigorous and broad validation of LGA's robustness. For further details on the additional experimental settings and results, please refer to **Appendix E.3** in the revised manuscript.
>
> ---
>
> *Further details addressing this are provided in the following comments.*
>
> ---
> [1] Janßen, N., Schaller, M., & Rosenhahn, B. (2025). Benchmarking M-LTSF: Frequency and Noise-Based Evaluation of Multivariate Long Time Series Forecasting Models. arXiv preprint arXiv:2510.04900.

---

> ### Author Response · Authors · 2025-11-22
>
> **3. Inclusion of Additional Baselines**
>
> We appreciate the reviewer’s suggestion to broaden the scope of our baseline comparison. To demonstrate that LGA’s benefits extend beyond standard Transformers, we conducted a focused evaluation on **SAMformer** [Ilbert et al., ICML 2024], a state-of-the-art model with linear complexity. We compared four variants on the ETTm1 dataset (Input 512) to analyze how LGA interacts with different structural biases: (1) Original SAMformer, (2) SAMformer + LGA, (3) PatchSAMformer (SAMformer with patching), and (4) PatchSAMformer + LGA.
>
> | **Model** | **Clean** | **Sev 1** | **Sev 2** | **Sev 3** | **Sev 4** | **Sev 5** |
> | --- | --- | --- | --- | --- | --- | --- |
> | SAMformer | 0.369 | 0.375 | 0.392 | 0.548 | 0.672 | 0.809 |
> | SAMformer + LGA | **0.361** | **0.370** | **0.385** | 0.532 | 0.661 | 0.770 |
> | PatchSAMformer | 0.384 | 0.396 | 0.426 | 0.837 | 1.064 | 1.312 |
> | **PatchSAMformer + LGA** | 0.372 | 0.377 | 0.390 | **0.488** | **0.588** | **0.682** |
>
> A remarkable observation is that applying patching to SAMformer (PatchSAMformer) drastically degrades its robustness against noise (MSE soars to 1.312 at Severity 5), likely due to the lack of a non-linear FFN. However, integrating LGA completely reverses this degradation (MSE 0.682), outperforming even the standard SAMformer.
>
> This shows that LGA stabilizes architectures sensitive to corruption. By imposing a local geometry prior, it regularizes attention and prevents overfitting to high-dimensional noise. We have expanded these results in Appendix E.2
>
> Additionally, as presented in Table 4, we extended our evaluation to iTransformer (Channel-wise Attention) and CATS (Cross-Temporal Attention). Please refer to our paper.
>
> ---
>
> In summary, we have addressed the reviewer's concerns by providing additional technical details, a clear motivation for TSRBench and significantly expanding our empirical validation. If you have any further questions or require additional clarifications, please feel free to ask.
>
> Again, thank you for your kind review and feedback.

---

> > ### Comment · Reviewer_9J2N · 2025-11-26
> >
> > Thanks to the authors for the detailed response. I've updated my rating.

---

> > > ### Author Response · Authors · 2025-11-27
> > >
> > > We sincerely thank you for your thoughtful review and for updating your rating. Your constructive feedback has been invaluable in improving our manuscript.
> > >
> > > Best regards,
> > > The Authors

---

### Official Review · Reviewer_ESKs · 2025-10-28

**Soundness:** 3
**Presentation:** 4
**Contribution:** 3
**Rating:** 6
**Confidence:** 4

**Summary:**

The contribution of this paper is twofold; it presents a method called local geometry attention designed (LGA) to model local structures of time series data in transformer neural networks, and it introduces TSRBench a corruption suite for evaluating robustness of time series models. In the empirical evaluation part of the submission, the LGA models are evaluated using TSRBench.

More specifically, LGA uses local Gaussian process regression to model the data/representation manifold near the query points of a an attention module. Since, computing the local geometry aware score of the model is computationally prohibitive, a second neural network is trained that predicts the metric tensor.
The TSRBench consists of a method to insert shifts and spikes into time series at 5 different severity levels.
In the empirical evaluation, LGA is inserted into different transformer models (with PatchTST being the default choice) to assess robustness of long-term time series forecasts.

**Strengths:**

- The LGA technique is universal in  the sense that it can easily be integrated into different transformer architectures and likely also in other networks that use attention modules.
- On the six corrupted time series datasets that were used, LGA (+PatchTST) consistently achieves the best forecasting accuracy especially at large severity levels, indicating that LGA adds robustness wrt shift and spike corruptions.
- The benchmarking tool comes with predefined and carefully calibrated severity levels which simplify its use.
- The paper is very well written, and therefore easy to read and understand. (But certain parts need more explanation, see questions)
- The paper is accompanied by an extensive supplementary material which provides the mathematical background, implementation details and further experiments.

**Weaknesses:**

- The LGA approach is only evaluated with respect to the (synthetic) TSRBench corruptions, i.e. shifts and spikes. It is not evaluated on other types of corruptions, for example on the anomaly types that are used in (Cheng et al, RobustTSF: ..., ICLR2024) or the synthesized outliers from (Lai et al., Revisiting Time Series Outlier Detection: Definitions and Benchmarks, NeurIPS 2021 Dataset and Benchmark track). One could therefore think that the benchmarks are designed to favor LGA.
- The LGA approach is only evaluated on forecasting tasks.
- It seems that the severity levels of TSRBench need to be calibrated for every dataset individually, and the guidelines are quite vague, i.e. "we ensured that the performance differences between severity levels were neither too large nor too small, and that the degree of noise strictly increased with each level".
Moreover, the resulting hyperparameter choices are model dependent and require training one (or multiple) neural networks per parameter choice. This hinders extending the benchmark to new datasets in a standardized manner, one of the key goals of this work ("Despite the need, a standardized benchmark for such realistic corruptions in time series remains a significant gap")
- The paper makes several implementation choices whose effects are not ablated empirically. This includes learning the metric tensor with an additional neural network (instead of computing it for every query) and using only a diagonal metric. Moreover, while it is stated that both are done because of computational costs, no runtime or complexity analysis of them are presented.
- For the datasets that are used, the manuscript refers to the Autoformer paper (Wu et al., NeurIPS 2021), but the actual originators of the datasets are not credited.

**Questions:**

- Is the robustness of LGA by design, or did it surprise you? If by design, what is the reason or intuition?
- Would you explain in more detail how the network that predicts the metric tensor is trained. Is this done simultaneously to training the transformer, afterwards, or  alternately? Considering the former, could you explain again, why training such a network is more efficient than computing the metric, or if it only shift compute from inference to training?
- Would you explain the connection to Riemannian geometry and what the point of the detour to the geodesic distance was. Is the connection only that the score is determined by a bilinear form that depends on the query point?
- Would you compare TSRBench with this very recent robustness benchmark ( [Janßen et al., arxiv, Oct 2025](https://arxiv.org/pdf/2510.04900) )

---

> ### Author Response · Authors · 2025-11-21
>
> We appreciate the reviewer's thoughtful comments and constructive feedback. We acknowledge the concerns raised regarding the **scope of our evaluation, the generalizability of TSRBench, and the implementation choices of LGA**, and we are grateful for the opportunity to address these points.
>
> Below, we provide a detailed response addressing each of the weaknesses [W#] and the questions [Q#] raised:
>
> ---
>
> **[W1 & Q4] Evaluation on Other Benchmarks and Comparison with Janßen et al. (2025)**
>
> We thank the reviewer for suggesting broader validation. We agree that comparing against diverse benchmarks is crucial to proving LGA's general robustness. To address this, we have (1) clarified the unique positioning of TSRBench compared to existing works and (2) conducted new stress-testing using the Janßen et al. (2025) framework.
>
> **1. Positioning TSRBench against Existing Benchmarks**
> Benchmarks like Cheng et al. (2024), Lai et al. (2021), and Janßen et al. (2025) address fundamentally different problems than TSRBench. As summarized below, **TSRBench fills a critical gap: evaluating forecasting robustness against *realistic, locally adaptive* corruptions on *real-world* data.**
>
> |  | **Ours (LGA / TSRBench)** | **RobustTSF (Cheng et al.)** | **M-LTSF Bench. (Janßen et al.)** | **Revisiting Outliers (Lai et al.)** |
> | --- | --- | --- | --- | --- |
> | **Primary Goal** | **Robust Forecasting (Inference)**: Train clean, Predict under noise | **Robust Forecasting (Training)**: Train noisy, Predict clean | **Robust Forecasting (Synthetic)**: Synthetic signal recovery | **Anomaly Detection**: Identify outliers |
> | **Data Origin** | **Real-world Data** with Injected Corruptions | **Real-world Data** with Injected Anomalies | **Fully Synthetic** Generation | **Synthetic** Generation |
> | **Noise Mechanism** | **Local & Adaptive (EVT)**: Uses DSPOT to find time-varying thresholds. | **Global & Fixed-Ratio**: Uses global $\sigma$ of the entire dataset. | **Synthetic Composition**: Signal + Noise functions (SNR based). | **Hybrid**: Pattern replacement & simple statistics ($\mu \pm \lambda\sigma$). |
>
> **2. Systematic Stress-Testing: Results on Janßen et al. (2025)**
>
> To rigorously evaluate LGA under controlled conditions, we adopted the recent parameterizable robustness framework by Janßen et al. (2025), which isolates specific robustness factors.
>
> **Experimental Setup**:
>
> - **7 Frequency Bands**: Very Low to Very High (testing temporal scale sensitivity)
> - **12 Signal Variants per Band**: 3 waveforms (Sine, Square, Sawtooth) × 4 SNR levels
> - **5 Noise Types**:
>     - Signal-Independent: White, Brownian, Impulse
>     - Signal-Dependent: Trend-dependent, Seasonal-dependent
> - **Total Evaluation**: 84 distinct signal configurations
>
> | **Frequency Band** | **Very High** | **High** | **Mid-High** | **Mid** | **Low-Mid** | **Low** | **Very Low** | **Avg.** |
> | --- | --- | --- | --- | --- | --- | --- | --- | --- |
> | **PatchLGA (MSE)** | **0.068** | **0.097** | **0.109** | 0.089 | **0.082** | **0.085** | **0.134** | **0.095** |
> | **PatchTST (MSE)** | 0.068 | 0.101 | 0.112 | 0.089 | 0.083 | 0.087 | 0.139 | 0.097 |
> | **Improvement** | = | **+4.0%** | **+2.7%** | = | **+1.2%** | **+2.3%** | **+3.6%** | **+2.1%** |
>
> As shown in the table, LGA demonstrates notable improvements in challenging frequency bands: *Very Low* (-3.6%), where lookback windows may not capture full cycles, and *High* (-4.0%), where sampling is sparse relative to signal frequency. This confirms that LGA's adaptive metric tensor captures periodic structures effectively across temporal scales.
>
> **3. Complementary Evaluation Framework**
>
> Based on these new results, we believe the combination of our proposed method and existing benchmarks provides a comprehensive robustness validation. We have structured our final evaluation into three complementary tiers:
>
> | Benchmark | Data Type | Corruption Type | Purpose |
> | --- | --- | --- | --- |
> | **TSRBench** | Real datasets (ETT, Weather) | Realistic corruptions (spikes, shifts) | Bridge clean benchmarks → noisy deployment |
> | **AWS CloudWatch (Appendix E.6)** | Operational data | Genuine anomalies | Validate on uncontrolled real-world noise |
> | **Janßen Framework** | Synthetic signals | Parameterized noise factors | Controlled Parametric Evaluation |
>
> The consistent improvements across all three paradigms—real-world controlled (TSRBench: 12.5% MSE reduction on ETTm1 @ severity 5), real-world uncontrolled (AWS: 22% MAE reduction), and synthetic theoretical (Janßen: 2.1% average improvement)—strongly validate the effectiveness of LGA's local geometry attention mechanism.We have added the full Janßen benchmark results and analysis to Appendix F: Evaluation on Janßen et al. Synthetic Benchmark in the revised manuscript.
>
> Also, we have added these benchmark results to Appendix E.3. Please refer to our paper for further information.
>
> ---

---

> > ### Comment · Reviewer_ESKs · 2025-11-25
> >
> > Dear authors. Thank you for the detailed response.
> >
> > Would you comment on the magnitude on the observed improvements of LGA in the tables addressing  W1.2 and W2.
> > How do they relate to performance differences resulting from hyperparameter tuning and different random splits?

---

> > > ### Author Response · Authors · 2025-11-27
> > >
> > > We thank the reviewer for this invaluable question. It is crucial to distinguish whether performance gains stem from genuine architectural superiority or merely from lucky random splits and hyperparameter overfitting.
> > >
> > > To address this, we have expanded our analysis to explicitly quantify the stability of LGA against **hyperparameter variations (Patch Length)** and **random initialization (Seeds)**.
> > >
> > > In response to your concern, we have conducted extensive additional experiments to validate the statistical significance of LGA's improvements reported in [W1.2] (Janßen et al. benchmark) and [W2] (Classification tasks). Specifically, we have:
> > >
> > > - **Evaluated multiple patch length configurations** (8, 16, 32) to ensure our findings are not artifacts of a specific architectural choice
> > > - **Repeated all experiments across three random seeds** (2021, 2022, 2023) to quantify performance variance
> > > - **Extended this analysis to our main TSRBench experiments** to provide comprehensive statistical evidence
> > >
> > > Below, we present the detailed results for each component:
> > >
> > > ---
> > >
> > > **1. Extended Analysis on Synthetic Benchmark (Janßen et al., 2025) [W1.2]**
> > >
> > > **Statistical Stability ($P=16$):**
> > >
> > > For the standard setting ($P=16$), we conducted experiments over 3 random seeds. As shown in the table below, PatchLGA demonstrates consistently lower MSE with minimal standard deviation, confirming that the performance gain is statistically significant.
> > >
> > > | **Frequency (P=16)** | **Very High** | **High** | **Mid High** | **Mid** | **Low Mid** | **Low** | **Very Low** | **Avg.** |
> > > | --- | --- | --- | --- | --- | --- | --- | --- | --- |
> > > | **PatchLGA** | 0.067 ± 0.0007 | **0.097** ± 0.0002 | **0.109** ± 0.0005 | **0.088** ± 0.0008 | 0.083 ± 0.0002 | **0.085** ± 0.0006 | **0.135** ± 0.0005 | **0.095** |
> > > | **PatchTST** | 0.067 ± 0.0006 | 0.100 ± 0.0002 | 0.112 ± 0.0006 | 0.089 ± 0.0006 | 0.083 ± 0.0002 | 0.087 ± 0.0009 | 0.138 ± 0.0001 | 0.097 |
> > >
> > > **Hyperparameter Robustness ($P \in \{8, 16, 32\}$):**
> > >
> > > We further verified that this superiority holds across different temporal resolutions. LGA consistently outperforms the baseline regardless of the patch length:
> > > - **$P=8$:** LGA achieves an average MSE of **0.097** vs. PatchTST's **0.098**.
> > > - **$P=32$:** LGA achieves an average MSE of **0.094** vs. PatchTST's **0.096**.
> > >
> > > | **Frequency (P=8)** | **Very High** | **High** | **Mid High** | **Mid** | **Low Mid** | **Low** | **Very Low** | **Avg.** |
> > > | --- | --- | --- | --- | --- | --- | --- | --- | --- |
> > > | **PatchLGA (MSE)** | 0.069 | **0.099** | **0.113** | 0.091 | 0.085 | **0.088** | **0.135** | **0.097** |
> > > | **PatchTST (MSE)** | 0.069 | 0.101 | 0.114 | **0.090** | 0.085 | 0.089 | 0.138 | 0.098 |
> > > | **Frequency (P=16)** | **Very High** | **High** | **Mid High** | **Mid** | **Low Mid** | **Low** | **Very Low** | **Avg.** |
> > > | **PatchLGA (MSE)** | 0.068 | **0.097** | **0.109** | 0.089 | **0.082** | **0.085** | **0.134** | **0.095** |
> > > | **PatchTST (MSE)** | 0.068 | 0.101 | 0.112 | 0.089 | 0.083 | 0.087 | 0.139 | 0.097 |
> > > | **Frequency (P=32)** | **Very High** | **High** | **Mid High** | **Mid** | **Low Mid** | **Low** | **Very Low** | **Avg.** |
> > > | **PatchLGA (MSE)** | 0.066 | **0.096** | **0.109** | 0.088 | 0.082 | **0.082** | **0.132** | **0.094** |
> > > | **PatchTST (MSE)** | 0.066 | 0.101 | 0.113 | 0.088 | 0.082 | 0.087 | 0.137 | 0.096 |
> > >
> > > **Key Observations:**
> > > - **Robustness across Frequencies:** LGA shows the most distinct improvements in the High (complex noise) and Very Low (long-term dependency) frequency bands.
> > > - **Low Variance:** The extremely low standard deviations (e.g., $\pm 0.0002$ in High freq) indicate that LGA’s training process is stable and reliable.
> > >
> > > ---
> > >
> > > *Further details are provided in the following comment.*

---

> ### Author Response · Authors · 2025-11-21
>
> **[W2] Evaluation on tasks beyond forecasting (Time Series Classification)**
>
> We appreciate the reviewer’s suggestion to demonstrate the versatility of our method beyond forecasting. In response, we have conducted additional experiments on **Time Series Classification** tasks to validate the generalizability of LGA.
>
> **Experimental Setup:**
>
> We utilized 10 multivariate time series datasets from the widely recognized UEA Archive. To ensure fair comparison and reproducibility, we strictly followed the experimental settings and data processing protocols provided in the **Time-Series-Library** [1] repository. We compared the classification accuracy of the original **PatchTST** against **PatchLGA**, where the standard attention mechanism was replaced with LGA.
>
> **Results:**
> The results are summarized in the table below. **PatchLGA** achieves an average accuracy of **69.51%**, outperforming the PatchTST baseline (68.73%). Notably, PatchLGA surpasses or matches the baseline in **8 out of 10 datasets**.
>
> | **Dataset** | **UWave.** | **Spoken.** | **SelfReg. SCP2** | **SelfReg. SCP1** | **PEMS-SF** | **JapaneseVowels** | **Heartbeat** | **Handwriting** | **FaceDetection** | **Ethanol.** | **Avg.** |
> | --- | --- | --- | --- | --- | --- | --- | --- | --- | --- | --- | --- |
> | PatchTST | 0.853 | **0.966** | 0.522 | 0.836 | 0.855 | **0.959** | 0.702 | 0.266 | 0.655 | 0.259 | **0.6873** |
> | PatchLGA | 0.853 | 0.964 | 0.522 | **0.853** | **0.873** | 0.957 | **0.707** | **0.287** | **0.665** | **0.27** | **0.6951** |
>
> These results demonstrate that LGA is not limited to forecasting but also enhances discriminative tasks. In classification, distinguishing between classes often relies on capturing the subtle **local geometric structure** of temporal patterns (e.g., specific shapes in handwriting or heartbeats). By explicitly modeling the data manifold, LGA learns more robust and discriminative representations than standard attention, leading to improved classification accuracy.
>
> ---
>
> **[W3] Calibration and Standardization of Severity Levels**
>
> We apologize for the insufficient explanation. The five severity levels (L1–L5) in TSRBench are fixed, standardized configurations that do not require re-calibration for new datasets or models.
>
> We established these levels through a one-time comprehensive analysis: we performed a grid search over 37 parameter configurations across six benchmark datasets (ETTh1, ETTh2, ETTm1, ETTm2, Electricity, and Weather) and analyzed the aggregated performance degradation of PatchTST. In the newly added Appendix F.2, we visualize the aggregated severity spectrum (Figure 8) and demonstrate that the selected five configurations consistently induce progressively increasing difficulty across diverse data distributions, with:
> - Statistical distinctiveness: Clear performance steps on the global average spectrum
> - Physical consistency: Monotonically evolving parameters (e.g., strictly increasing injection frequency)
>
> Future users can directly apply these fixed parameter sets (Table 1) to any new dataset without training models or tuning hyperparameters, fulfilling TSRBench's goal as a standardized, easy-to-use benchmark.
>
> For detailed parameter selection and analysis, please refer to the newly added Appendix F.2.
>
> ---
>
> [1] https://github.com/thuml/Time-Series-Library.git

---

> > ### Author Response · Authors · 2025-11-21
> >
> > **[W4] Justification of Implementation Choices (Metric Tensor Design)**
> >
> > We thank the reviewer for highlighting the need for a rigorous justification of our design choices. We agree that the decision to use a **learned diagonal metric** over other plausible alternatives (such as computing a full metric on-the-fly or learning a full matrix) requires both theoretical and empirical backing.
> >
> > In the revised manuscript, we have added a new **Appendix C: Ablation Study on Metric Tensor Implementations**, which includes:
> >
> > 1. A detailed **Computational Complexity Analysis** (FLOPs).
> > 2. A **Theoretical Stability Analysis** explaining why full-matrix learning fails.
> > 3. An **Empirical Ablation** comparing accuracy across different implementations.
> >
> > Our findings are summarized as follows:
> >
> > **1. Computational Efficiency (Appendix B.3.1):**
> >
> > We quantified the theoretical FLOPs for both inference and training. As shown in Table 6 (Appendix C.1), the full-matrix variants are approximately 2.5$\times$ more expensive than our proposed diagonal method due to the cubic complexity of matrix inversion ($O(ND^3)$) or the quadratic growth of parameters ($D^2$). Our diagonal approach maintains the same asymptotic complexity as standard dot-product attention ($O(N^2D)$).
> >
> > **2. Theoretical Robustness & Stability (Appendix B.3.2):**
> >
> > We mathematically analyzed why "more expressive" full matrices do not necessarily yield better performance:
> >
> > - **Robustness against Noise:** Computing the metric on-the-fly (Empirical) incorporates the noise structure of the inference batch into the metric (Manifold Distortion). In contrast, our **Learned Metric** acts as a structural prior learned from the training distribution, making it robust to inference-time corruptions.
> >
> > - **Spectral Stability:** We proved (Proposition 1 & 2) that learning a full metric tensor is inherently unstable. Since time series data often have a low intrinsic dimension, the target inverse covariance is ill-conditioned ($\lambda_{\min} \approx 0$). This causes the regression target to fluctuate massively even with small errors, leading to optimization failure. The **Diagonal Parameterization** resolves this by decoupling the dimensions, ensuring stable convergence.
> >
> > **3. Empirical Superiority (Appendix B.3.3):**
> >
> > We empirically compared three variants: (1) LGA-Diag-Learn (Ours), (2) LGA-Full-Empirical, and (3) LGA-Full-Learned on the ETTm1 dataset.
> > The results confirm our analysis: the Full-Learned variant exhibited the worst performance due to optimization instability, and the Full-Empirical variant degraded under severe noise due to manifold distortion. Our LGA-Diag-Learn consistently achieved the lowest MSE, offering the best trade-off between computational efficiency ($1.0\times$ cost) and forecasting accuracy.
> >
> > We believe this comprehensive analysis fully justifies our implementation choices.
> >
> > ---
> >
> > **[W5] Dataset Citations**
> >
> > We thank the reviewer for pointing out this oversight. We agree that it is important to give proper credit to the original creators of the datasets. In the revised manuscript (**Section 5: Experiments**), we have updated the text to explicitly cite the original sources and repositories for all datasets used. Specifically, we have added footnotes for the Weather and Electricity datasets pointing to their original sources (Max Planck Institute and UCI Machine Learning Repository) and cited Zhou et al. (2021) as the originator of the ETT datasets.
> >
> > ---
> >
> > **[Q1] Is the robustness of LGA by design, or did it surprise you? If by design, what is the reason or intuition?**
> >
> > The robustness of LGA was **intentionally designed from the start**. As shown in Figure 2 (Section 3), we observed that query-key embeddings in time series Transformers naturally form **local clusters** corresponding to periodic patterns. This observation led to a key insight: if normal patterns cluster together, then **anomalous data would lie outside these clusters**, and **data from different phases should receive less attention** as they represent temporally misaligned patterns.
> >
> > This motivated us to design an attention mechanism that explicitly respects this **local geometric structure**—naturally down-weighting out-of-distribution points and focusing within coherent neighborhoods. However, we found that existing benchmarks were designed for different evaluation scenarios (training-time robustness, anomaly detection, or fully synthetic data), which could not properly validate our hypothesis about **inference-time forecasting robustness under realistic corruptions**. This gap motivated the development of TSRBench alongside LGA, providing the necessary framework to rigorously test geometry-aware attention under statistically-grounded, locally-adaptive corruptions.

---

> ### Author Response · Authors · 2025-11-21
>
> **[Q2] Training Procedure and Efficiency of the Metric Prediction Network**
>
> **1. Training Dynamics: Simultaneous and End-to-End**
>
> To clarify, there are no separate, alternating, or post-hoc training phases. The metric prediction network ($f_\theta$) is trained simultaneously with the Transformer in a standard end-to-end manner.
> - Single Loop: Within a single training iteration, the model performs one forward pass to generate both the forecast and the predicted metric tensor.
> - Unified Loss: We simply sum the forecasting loss and the auxiliary geometric loss (Eq. 7) to update the entire model via a single backward pass:
> $$\mathcal{L}\_{total} = \mathcal{L}\_{forecast} + \lambda \mathcal{L}\_G$$
>
> **2. Efficiency and Justification**
>
> Regarding your question on the efficiency and rationale behind learning the metric instead of computing it directly:
>
> This design choice is critical not only for Computational Efficiency (reducing inference complexity from cubic to linear) but also for Theoretical Robustness (preventing overfitting to noise).
>
> For a detailed breakdown of the Computational Complexity (FLOPs analysis) and the Theoretical Stability analysis justifying this choice, please refer to our Response to Weakness 4 [W4], where we address these implementation choices in depth.
>
> ---
>
> **[Q3] Connection to Riemannian Geometry and the Role of Geodesic Distance**
>
> We sincerely appreciate the reviewer for raising this important question, as it provides us with an opportunity to clarify the theoretical foundation and the conceptual value of invoking Riemannian geometry in our framework.
>
> We would like to emphasize that **the connection to Riemannian geometry is not a superficial "detour" but rather a deliberate dual interpretation** that enhances both the theoretical rigor and the intuitive understanding of our approach. Allow us to explain this more carefully.
>
> Our approach originates from the GP framework (Section 3.2), which naturally produces a query-dependent bilinear form:
> score(q,k) = −(k−q)⊤ G(q)(k−q).
>
> This structure has two complementary interpretations:
>
> 1. **Statistical interpretation (GP):** G(q) approximates the inverse local covariance, and the score measures predictive variance—a principled density surrogate (Theorem 1).
> 2. **Geometric interpretation (Riemannian):** The same bilinear form can be viewed as a first-order geodesic distance expansion on a data manifold:
> d_g(q,k)² ≈ (k−q)⊤ G(q)(k−q),
> where G(q) acts as a learned local metric tensor.
>
> **Why mention both?** The GP view provides the *theoretical grounding* and *statistical validity*. The Riemannian view provides *geometric intuition*: LGA adapts to the intrinsic curvature of the data manifold, explaining why it naturally down-weights anomalies (which lie "far" in the learned geometry) and respects local periodic structure.
>
> **Is it merely a bilinear form?** No. The key is that G(q) is *query-adaptive* and *learned from data geometry*—not a fixed global metric (standard attention) or a manually designed kernel. The Riemannian perspective makes explicit that we are learning a non-Euclidean geometry tailored to the local structure of time series embeddings.
>
> In summary, the geodesic distance discussion was not a detour but a deliberate dual framing that strengthens our contribution: GP theory provides statistical rigor, while Riemannian geometry offers geometric interpretability.
>
> ---
>
> Again, we are grateful for the reviewer’s insightful comments, which have significantly improved the quality and rigor of our manuscript. By expanding our evaluation to include the synthetic robustness benchmark (Janßen et al.) and classification tasks, and by providing a thorough theoretical justification for our design choices, we hope to have resolved all raised concerns. We are confident that these revisions highlight the distinct contribution of LGA and TSRBench to the field of robust time series forecasting.

---

> ### Author Response · Authors · 2025-11-27
>
> **2. Extended Analysis on Time Series Classification [W2]**
>
> Next, we extended our evaluation to the **Time Series Classification** task using 10 UEA datasets. We utilized the same multi-seed (Seeds: 2021, 2022, 2023) used in our forecasting experiments to ensure consistency with the stability analysis in Appendix E.1.
>
> Below, we report the full experimental results showing the average accuracy and standard deviation for each dataset across all patch lengths ($P \in \{8, 16, 32\}$).
>
> | **Dataset (P=8)** | **UWave.** | **Spoken.** | **SelfReg. SCP2** | **SelfReg. SCP1** | **PEMS-SF** | **JapaneseVowels** | **Heartbeat** | **Handwriting** | **FaceDetection** | **Ethanol.** | **Avg.** |
> | --- | --- | --- | --- | --- | --- | --- | --- | --- | --- | --- | --- |
> | **PatchTST** | 0.8479 ± 0.0048 | **0.9623 ± 0.0016** | **0.5259 ± 0.0128** | 0.8544 ± 0.0161 | 0.8208 ± 0.0417 | 0.9613 ± 0.0041 | 0.7073 ± 0.0176 | 0.2541 ± 0.0072 | 0.6678 ± 0.0097 | 0.2649 ± 0.0022 | 0.6867 |
> | **PatchLGA** | **0.8604 ± 0.0072** | 0.9610 ± 0.0050 | 0.5148 ± 0.0170 | **0.8658 ± 0.0099** | **0.8593 ± 0.0372** | **0.9631 ± 0.0056** | **0.7187 ± 0.0102** | **0.2600 ± 0.0187** | **0.6771 ± 0.0010** | **0.2712 ± 0.0088** | **0.6951** |
> | **Dataset (P=16)** | **UWave.** | **Spoken.** | **SelfReg. SCP2** | **SelfReg. SCP1** | **PEMS-SF** | **JapaneseVowels** | **Heartbeat** | **Handwriting** | **FaceDetection** | **Ethanol.** | **Avg.** |
> | **PatchTST** | 0.8469 ± 0.0063 | 0.9686 ± 0.0028 | **0.5222 ± 0.0167** | **0.8589 ± 0.0200** | 0.8324 ± 0.0231 | 0.9523 ± 0.0068 | 0.7041 ± 0.0123 | 0.2718 ± 0.0123 | 0.6773 ± 0.0210 | 0.2750 ± 0.0144 | 0.6909 |
> | **PatchLGA** | **0.8563 ± 0.0083** | **0.9688 ± 0.0038** | 0.5185 ± 0.0116 | 0.8544 ± 0.0240 | **0.8516 ± 0.0120** | **0.9559 ± 0.0016** | **0.7171 ± 0.0000** | **0.2961 ± 0.0089** | **0.6781 ± 0.0071** | **0.2801 ± 0.0088** | **0.6977** |
> | **Dataset (P=32)** | **UWave.** | **Spoken.** | **SelfReg. SCP2** | **SelfReg. SCP1** | **PEMS-SF** | **JapaneseVowels** | **Heartbeat** | **Handwriting** | **FaceDetection** | **Ethanol.** | **Avg.** |
> | **PatchTST** | 0.8438 ± 0.0083 | **0.9720 ± 0.0003** | **0.5352 ± 0.0160** | 0.8385 ± 0.0188 | **0.8516 ± 0.0273** | 0.9459 ± 0.0027 | 0.6959 ± 0.0102 | 0.2965 ± 0.0147 | **0.6745 ± 0.0013** | 0.2788 ± 0.0096 | 0.6932 |
> | **PatchLGA** | **0.8594 ± 0.0031** | 0.9707 ± 0.0065 | 0.5259 ± 0.0160 | **0.8441 ± 0.0020** | 0.8382 ± 0.0208 | **0.9532 ± 0.0068** | **0.7122 ± 0.0146** | **0.3184 ± 0.0139** | 0.6687 ± 0.0089 | **0.2814 ± 0.0038** | **0.6972** |
>
> **Analysis of Results:**
>
> - **Consistent Superiority:** PatchLGA outperforms the baseline on average across all patch lengths ($P=8, 16, 32$).
> - **Stability:** The standard deviations are generally low and comparable to the baseline, indicating that the improvements are robust to random initialization.
>
> *To ensure these results are fully reproducible, we used the standard codebase from the **Time-Series-Library** [1]. We have included **a reproduction code and script in our supplementary material** for the exact replication of these tables.*
>
> ---
>
> **3.  Statistical Robustness on Main TSRBench Experiments**
>
> Additionally, to further validate the reliability of our core contribution, we have also conducted the same multi-seed, multi-configuration analysis on our main TSRBench experiments. We repeated all experiments on the six benchmark datasets (ETTh1, ETTh2, ETTm1, ETTm2, Electricity, Weather) across three random seeds (2021, 2022, 2023).
>
> The comprehensive results are presented in **Appendix E.1: Stability Analysis and Detailed Performance**, which includes full tables with mean ± standard deviation for all experimental conditions. Please refer to **Appendix E.1** for the complete statistical analysis and detailed performance breakdowns.
>
> ---
>
> These extensive experiments confirm that the improvements reported throughout our paper are statistically robust across different experimental conditions, random initializations, and architectural configurations. Thank you for your insightful suggestion.
>
> ---
>
> [1] https://github.com/thuml/Time-Series-Library

---

> > ### Comment · Reviewer_ESKs · 2025-11-27
> > **Summary of discussion**
> >
> > The authors wrote an extensive response which addressed all my concerns. I raise my score.

---

> > > ### Author Response · Authors · 2025-11-27
> > >
> > > We sincerely appreciate the time you invested and the constructive feedback you provided throughout this review process. Thank you for recognizing our efforts and for your improved assessment.
> > >
> > > We believe the additional experiments and analyses incorporated based on your suggestions have significantly strengthened the final manuscript.
> > >
> > > Best regards,
> > > The Authors

---

### Official Review · Reviewer_WvGd · 2025-10-29

**Soundness:** 3
**Presentation:** 3
**Contribution:** 3
**Rating:** 6
**Confidence:** 4

**Summary:**

This paper studies the robustness of time series forecasting transformer models under realistic corruptions. It first propose a novel attention mecanism derived from local Gaussian process theory called Local Geometry Attention (LGA). It takes into account the geometry of data to be more robust to corruptions in the time series data. It can be implemented by learning a neural network over the query vectors seen during training. Then, the paper introduces TSRBench, a benchmark with realistic corruptions (spikes and level shift) to estimate the robustness of models for forecasting. Experiments over 6 common forecasting benchmarks (with their corrupted versions) and 3 baselines show the performance benefits of the LGA attention incorporated into a PatchTST model.

**Strengths:**

- The paper is well-written and the proposed approach well motivated
- The proposed benchmark is very interesting with sounded corruptions (inspired from ImageNet-C benchmark)
- The experiments are convincing to show the performance improvement brought by LGA

**Weaknesses:**

- While the proposed method is sound, it requires training neural networks to compute the Local Geometry Attention. The current submission is missing a computational cost comparison when compared to self-attention
- Since the main contribution is a novel attention mecanism, it would be interesting to see the comparison to other types of attention in addition to the traditional one (e.g., channel-wise attention [1, 2])
- Connected to the previous weakness, it would be interesting to compare to more models such as iTransformer [1] or SAMformer [2] that reported robust performance with temporal/spatial and spatial attention respectively.
- While the benchmarks is interesting, I believe the current proposal would be strengthen with additional models and datasets to clearly show the failure of other models under realistic corruptions
- It would be interesting to see how LGA behaves when integrated into other models than PatchTST, for instance could it be applied on SAMformer with channel-wise attention or i-transformer with channel and temporal wise attention?

Overall, I find the submission interesting with a well-motivated LGA and robustness benchmark. However, it would be strengthen with additional methods and computational cost comparison.

*References*

[1] Liu et al. iTransformer: Inverted Transformers Are Effective for Time Series Forecasting. ICLR 2024

[2] Ilbert et al. SAMformer: Unlocking the Potential of Transformers in Time Series Forecasting with Sharpness-Aware Minimization and Channel Wise Attention. ICML 2024

**Questions:**

- The LGA is computed by training a neural network on query vectors seen during training. Does it mean that there are two interconnected training loops: one for the model, one for the LGA module? If it is the case, what is the additional computational cost and does it vary with the number of training steps? Could the authors clarify that please?
- Were the experiments run over several seeds? If yes, could the authors make the standard deviation appear? If no, could the authors please conduct the experiments with 2 additional seeds to have an idea of the significance of the performance improvement?

---

> ### Author Response · Authors · 2025-11-21
>
> We appreciate the reviewer's thoughtful comments and constructive feedback. We acknowledge the concerns raised regarding **computational costs, broader model comparisons, and experimental robustness**, and we are grateful for the opportunity to address these points.
>
> Below, we provide a detailed response addressing each of the weaknesses [W#] and the questions[Q#] raised:
>
> ---
>
> **[W1&Q1] Computational Cost & Training Procedure**
>
> We thank the reviewer for this important question. Below we provide a detailed breakdown to clarify the training mechanism and quantify computational costs.
>
> **1. Training Procedure: Single Unified Loop (Response to Q1)**
>
> To clarify: **No, there are no separate or interconnected training loops.** LGA is trained in a standard end-to-end manner within a single unified loop.
>
> During each iteration, the model performs just **one forward pass** to generate both the forecast and the metric tensor. We simply sum the forecasting loss and the auxiliary geometric loss ( $\mathcal{L}\_{total} = \mathcal{L}_{forecast} + \lambda \mathcal{L}\_G$ ) and update the model via a **single backward pass**.
>
> Regarding computational cost: Since we calculate the target metric using a fixed number of sampled queries ($N_s=256$) regardless of the batch size or training stage, the additional cost remains constant and predictable throughout the entire training process.
>
> **2. Computational Cost Analysis (Response to W1)**
>
> To quantify this cost rigorously, we conducted a comprehensive FLOPs analysis comparing standard self-attention (SDP) and LGA under our experimental configuration for the ETTm1 dataset (Batch $B=128$, Seq Len $N=64$, Dim $C=128$, Heads $H=16$, $d_G = 64$, Sampled Queries $N_s = 256$).
>
> **FLOPs Breakdown:**
> - **Standard Self-Attention (SDP):**
>     - Linear projections ($Q, K, V, O$): $8BNC^2$
>     - Score & Weighted Sum: $4BHN^2D$
>     - Softmax: $5BHN^2$
>     - **Total:** 9.748 GFLOPs (Inference) / 28.890 GFLOPs (Training)
> - **Local Geometry Attention (LGA):**
>     - Linear projections: $8BNC^2$ (Same as SDP)
>     - Metric Prediction ($f\_\theta$): $4BNHDd_G$
>     - Mahalanobis Score & Sum: $6BHN^2D$ (Slightly higher than dot-product)
>     - **Target Generation (Training Only):** $6N_sHN^2D$ (Compute $G_{true}$)
>     - **Total:** 12.640 GFLOPs (Inference) / 44.862 GFLOPs (Training)
>
> Relative Overhead on Full Transformer Block:
>
> While LGA shows higher FLOPs in the attention module itself, the relative impact is modest when considering the full Transformer block (Attention + Feed-Forward Network), as the FFN dominates the compute budget (~61%).
>
> | **Setting** | **Module Scope** | **Standard SDP** | **LGA (Ours)** | **Overhead** |
> | --- | --- | --- | --- | --- |
> | **Inference** | Attention Only | 9.748 G | 12.640 G | +29.7% |
> | **(Forward)** | **Full Block (Attn + FFN)** | **24.78 G** | **27.67 G** | **+11.7%** |
> | **Training** | Attention Only | 28.890 G | 44.862 G | +55.3% |
> | **(Fwd+Bwd)** | **Full Block (Attn + FFN)** | **73.99 G** | **89.96 G** | **+21.6%** |
>
> The additional cost for LGA is approximately **11.7% during inference** and **21.6% during training** for a full block. We believe this is a reasonable trade-off for the significant robustness gains (e.g., 12.5% MSE reduction on ETTm1 under severe noise). Furthermore, for high-dimensional datasets like Electricity, the overhead ratio decreases further because the sampling size $N_s$ remains fixed while the model size grows. We have included this detailed analysis in **Appendix B.3** of the revised manuscript.
>
> ---
>
> **[Q2] Evaluation over multiple seeds and statistical significance**
>
> We thank the reviewer for this insightful and critical suggestion. We agree that evaluating performance stability across different initializations is essential for establishing the reliability of our method, a step we had inadvertently overlooked.
>
> Following your recommendation, we have conducted additional experiments with two more random seeds (totaling three seeds: 2021, 2022, 2023) for all datasets and experimental settings. We have subsequently added a new section, **Appendix E.1 Stability Analysis and Detailed Performance**, which provides comprehensive tables reporting the **mean and standard deviation** for all metrics.
>
> The results from these repeated runs confirm that **PatchLGA** not only maintains superior average performance but also demonstrates **lower variance** compared to the baselines, especially under high severity levels (Severity 4-5). This empirical evidence reinforces our claim that the geometry-aware attention mechanism acts as an effective regularizer, leading to a more stable and robust optimization process even in the presence of severe corruptions.

---

> ### Author Response · Authors · 2025-11-21
>
> **[W2, W3, W5] Generalizability and Applicability of LGA**
>
> We sincerely thank the reviewer for highlighting recent advances in transformer architectures for time series [Liu et al., ICLR 2024; Ilbert et al., ICML 2024] and suggesting broader architectural validation. We agree that demonstrating LGA's transferability beyond PatchTST is essential for establishing its contribution.
>
> To directly address the reviewer's question: **Yes, LGA can be successfully integrated into different architectures beyond PatchTST.**
>
> Our experiments demonstrate LGA's effectiveness across:
>
> 1. **SAMformer** [Ilbert et al., ICML 2024] (Table 12): Stabilizes FFN-free architectures, preventing noise amplification
>
> 2. **iTransformer** [Liu et al., ICLR 2024] (Table 4): Enhances channel-wise attention robustness
> 3. **CATS** [Kim et al., 2024] (Table 4): Improves cross-attention mechanisms
>
> **Implementation Simplicity:**
>
> Integrating LGA requires only:
> 1. Replace attention score: k·q → -(k-q)ᵀG(q)(k-q) (Eq. 5)
> 2. Train metric prediction network $f\_θ$
>
> **1. Cross-Architecture Validation: Channel-wise and Cross-Attention**
>
> As initially presented in **Table 4**, we have evaluated LGA on architectures employing different attention types, including **iTransformer** (Inverted-Transformer with channel-wise attention) and **CATS** (Cross-attention).
>
> - **Channel-wise Attention:** We acknowledge the reviewer's intuition that the impact of LGA might be less pronounced for channel-wise attention compared to temporal attention. Since LGA is theoretically grounded in capturing the local geometry of temporal patterns (e.g., periodicity and trends), its inductive bias is most effective when applied along the time dimension. Nevertheless, as shown in Table 4, **LGA still provides consistent performance gains for iTransformer (e.g., reducing MSE from 1.309 to 1.266 at Severity 5 on ETTm1).** This confirms that while the effect is maximized in temporal attention, the geometric regularization of LGA offers benefits for robustness.
>
>
> - **Cross-Temporal-Attention:**
> As shown in Table 4, **LGA enhances CATS despite its cross-attention operating on linearly embedded inputs (e.g., reducing MSE from 1.102 to 1.037 at Severity 5 on ETTm1).** The improvement is notable given cross-attention's vulnerability to input corruptions. LGA's geometric regularization mitigates this effect.
>
> **2. New Experiments on SAMformer**
>
> To further address the concern regarding model diversity (W2 & W4), we conducted additional experiments using **SAMformer**, a state-of-the-art Transformer. We compared four variants on the ETTm1 dataset (Input 512) to analyze how LGA interacts with different structural biases: (1) Original SAMformer, (2) SAMformer + LGA, (3) PatchSAMformer (SAMformer with patching), and (4) PatchSAMformer + LGA.
>
> | **Model** | **Clean** | **Sev 1** | **Sev 2** | **Sev 3** | **Sev 4** | **Sev 5** |
> | --- | --- | --- | --- | --- | --- | --- |
> | **SAMformer** | 0.369 | 0.375 | 0.392 | 0.548 | 0.672 | 0.809 |
> | **SAMformer + LGA** | **0.361** | **0.370** | **0.385** | 0.532 | 0.661 | 0.770 |
> | **PatchSAMformer** | 0.384 | 0.396 | 0.426 | 0.837 | 1.064 | 1.312 |
> | **PatchSAMformer + LGA** | 0.372 | 0.377 | 0.390 | **0.488** | **0.588** | **0.682** |
>
> **Key Findings & Analysis:**
>
> - **Vulnerability of Patching:** A striking observation is that applying patching to SAMformer (`PatchSAMformer`) drastically degrades its robustness against noise (MSE soars to **1.312** at Severity 5). We attribute this to the architectural characteristic of SAMformer; unlike Transformers, it lacks a non-linear Feed-Forward Network (FFN) after attention. Consequently, the high-dimensional linear projections required for patching seem to amplify input noise without the buffering effect of an FFN.
> - However, integrating LGA into this highly vulnerable architecture (`PatchSAMformer + LGA`) completely reverses this degradation. **It not only recovers the performance but achieves the highest robustness among all four variants (MSE 0.682 at Severity 5), outperforming even the standard SAMformer.**
>
> This shows that LGA stabilizes architectures sensitive to corruption. By imposing a local geometry prior, it regularizes attention and prevents overfitting to high-dimensional noise. We have expanded these results in **Appendix E.2**. Please refer for further information of these experiments.

---

> ### Author Response · Authors · 2025-11-21
>
> **[W4] Broader Evaluation with Additional Datasets**
>
> We thank the reviewer for this important suggestion. While our main evaluation focuses on 6 standard forecasting benchmarks, we have conducted **additional validation** to demonstrate LGA's broad applicability:
>
> **Systematic Robustness Analysis under Controlled Signal Conditions** (Janßen et al. (2025) [1])
>
> To rigorously evaluate LGA under controlled conditions, we adopted the recent parameterizable robustness framework by Janßen et al. (2025), which isolates specific robustness factors.
>
> **Experimental Setup**:
>
> - **7 Frequency Bands**: Very Low to Very High (testing temporal scale sensitivity)
> - **12 Signal Variants per Band**: 3 waveforms (Sine, Square, Sawtooth) × 4 SNR levels
> - **5 Noise Types**:
>     - Signal-Independent: White, Brownian, Impulse
>     - Signal-Dependent: Trend-dependent, Seasonal-dependent
> - **Total Evaluation**: 84 distinct signal configurations
>
> | **Frequency Band** | **Very High** | **High** | **Mid-High** | **Mid** | **Low-Mid** | **Low** | **Very Low** | **Avg.** |
> | --- | --- | --- | --- | --- | --- | --- | --- | --- |
> | **PatchLGA (MSE)** | **0.068** | **0.097** | **0.109** | 0.089 | **0.082** | **0.085** | **0.134** | **0.095** |
> | **PatchTST (MSE)** | 0.068 | 0.101 | 0.112 | 0.089 | 0.083 | 0.087 | 0.139 | 0.097 |
> | **Improvement** | = | **+4.0%** | **+2.7%** | = | **+1.2%** | **+2.3%** | **+3.6%** | **+2.1%** |
>
> LGA consistently improves robustness across diverse frequency bands and noise types, with particularly notable gains in challenging conditions (Very Low and High frequency bands). For detailed analysis, please refer to the newly added Appendix E.3.
>
>
> ### **Complementary Evaluation Framework**:
>
> We believe these three benchmarks provide **comprehensive robustness validation**:
>
> | Benchmark | Data Type | Corruption Type | Purpose |
> | --- | --- | --- | --- |
> | **TSRBench** | Real datasets (ETT, Weather) | Realistic corruptions (spikes, shifts) | Bridge clean benchmarks → noisy deployment |
> | **AWS CloudWatch (Appendix E.6)** | Operational data | Genuine anomalies | Validate on uncontrolled real-world noise |
> | **Janßen Framework** | Synthetic signals | Parameterized noise factors | Controlled Parametric Evaluation |
>
> The **consistent improvements across all three paradigms**—real-world (TSRBench: 12.5% MSE reduction on ETTm1 @ severity 5, AWS: 22% MAE reduction), and synthetic (Janßen: 2.1% average improvement)—strongly
> validate the **effectiveness** of LGA's local geometry attention mechanism. Also, we have added the Janßen benchmark results to **Appendix F: Evaluation on Janßen et al. Synthetic Benchmark**. Please refer to the revised paper for the further information.
>
> ---
>
> In summary, we have reinforced our paper with: (1) a rigorous FLOPs analysis confirming LGA’s efficiency (only ~11% inference overhead), (2) extensive experiments on SAMformer proving LGA’s architectural applicability, (3) validation on both synthetic (Janßen et al.) benchmarks establishing broad robustness, and (4) statistical validation across multiple seeds. We believe these additions comprehensively address your concerns and firmly establish LGA as a robust and generalized solution for time series forecasting.
>
> Again, thank you for your valuable and kind review for our paper.
>
> ---
>
> [1] Janßen, N., Schaller, M., & Rosenhahn, B. (2025). Benchmarking M-LTSF: Frequency and Noise-Based Evaluation of Multivariate Long Time Series Forecasting Models. arXiv preprint arXiv:2510.04900.

---

> > ### Comment · Reviewer_WvGd · 2025-11-27
> > **Thanks a lot!**
> >
> > I thank the authors for their thorough answers.
> >
> > - Thanks for clarifying the unified training framework of LGA
> > - The computational cost experiments are convincing: the performance gains compensate the cost overhead. Given the relatively small sizes of models, this seems reasonable.
> > - The extensive additional experiments are convincing, showcasing the generalisability of the methods for various datasets, models and random seeds
> >
> > Overall, the rebuttal addressed all my concerns and I will update my score accordingly. Thanks to the authors again.

---

> > > ### Author Response · Authors · 2025-11-27
> > >
> > > We appreciate your time and the constructive feedback. Thank you for acknowledging our efforts and for increasing your rating.
> > >
> > > We are confident that the additional experiments and analyses added based on your review have made the final version of the paper much stronger.
> > >
> > > Best regards, The Authors

---

### Official Review · Reviewer_Jv6F · 2025-11-01

**Soundness:** 3
**Presentation:** 2
**Contribution:** 2
**Rating:** 4
**Confidence:** 2

**Summary:**

The manuscript introduces Local Geometry Attention, an attention mechanism for time-series forecasting that replaces dot-product similarity with a query-specific metric estimated from local Gaussian-process theory, aiming to make attention scores geometry-aware and less sensitive to noisy or anomalous keys. It also proposes TSRBench, a robustness benchmark that injects spike and level-shift corruptions using statistically grounded processes and calibrated severities.

**Strengths:**

LGA is designed to improve robustness of attention to local anomalies and noise.

TSRBench is a potentially useful benchmarking tool for researchers to evaluate robustness under controlled corruptions.

**Weaknesses:**

The description of LGA is not detailed enough. A step-by-step derivation of the attention score and kernel construction is needed to enhance the clarity. Please also provide the tensor shapes at each stage.

The GP motivation is under-explained: it is unclear how local GP assumptions translate into better attention weights or how the geometric structure is actually learned from data.

The set of baselines is limited, making it hard to judge whether robustness gains hold against recent strong time-series Transformers.

**Questions:**

(a) Please expand the discussion of non-uniform, locally clustered time-series observations, since this is the core motivation for introducing LGA.

(b) Explain in detail how queries and keys are embedded in the 2D panel in Figure 2 and how this visualization demonstrates a more effective attention mechanism.

(c) Show how the proposed local kernel–covariance compares to vanilla dot-product attention under the same setup, and quantify robustness gains.

(d) In Figure 5, replace training time with FLOPs to make comparisons hardware-agnostic.

(e) Add more recent baselines in Table 2 to better position LGA and TSRBench.

---

> ### Author Response · Authors · 2025-11-21
>
> We appreciate the reviewer's thoughtful comments and constructive feedback. We acknowledge the concerns raised regarding the clarity of our method description, theoretical grounding, and experimental comparisons, and we are grateful for the opportunity to address these points.
>
> Below, we provide a detailed response addressing each of the weaknesses [W#] and the questions[Q#] raised:
>
> ---
>
> **[W1] Detailed description of LGA algorithm with tensor shapes**
>
> We thank the reviewer for this valuable suggestion. We acknowledge that while Algorithm 1 in Appendix B provides a step-by-step implementation of LGA, the lack of explicit tensor shapes at each stage may have hindered readability. Due to page constraints in the main paper, we placed the detailed algorithmic description in the appendix, but we now recognize that tensor shape annotations are essential for clarity.
>
> In the revised manuscript, we have updated Algorithm 1 to include explicit tensor shapes at each computation step (e.g., Query $Q ∈ \mathbb{R}^{B×L_Q×D}$, metric tensor $G ∈ \mathbb{R}^{B×H×L_Q×D_h}$, attention scores $∈ \mathbb{R}^{B×H×L_Q×L_K}$). This should make the computational flow significantly more transparent. We kindly refer the reviewer to the updated Algorithm 1 in Appendix B of the revised manuscript.
>
> ---
>
> **[W2] GP motivation and geometric structure learning**
>
> We thank the reviewer for highlighting this important point. We acknowledge that our explanation of how local GP assumptions translate into geometry-aware attention could be more explicit. Below, we clarify the theoretical pathway step-by-step and indicate specific revisions.
>
> **Step 1: Why Gaussian Processes?**
>
> The GP framework provides a principled probabilistic method to capture local data density. Starting from the kernel-weighted local covariance around query q:
>
> $$
> Σ(q) = Σ\_i ω\_i(q)(k\_i - q)(k\_i - q)^T  (Eq. 1),
> $$
>
> we construct a local GP with this covariance structure. The predictive variance at key k is:
> $$
> σ^2\_q(k) = (k - q)^T G(q)(k - q), \quad  where  \quad    G(q) = σ^2[Σ(q) + σ^2I]^{-1}  (Eqs. 3-4).
> $$
> Critically, this variance is **small in directions densely populated by keys** (high local support) and **large in sparse or anomalous directions** (low support). Appendix A.2 (Theorem 1) formally proves that high-variance regions occupy limited probability mass under the data distribution, justifying $-σ^2\_q(k)$ **as a principled local density surrogate.**
>
> **Step 2: Why density matters for attention, and how geometric structure is learned from data**
>
> Using negative variance as our similarity measure yields:
> $$
> score(q, k) = -(k - q)^T G(q)(k - q)  (Eq. 5).
> $$
> **Why this produces better attention weights:** In time series, segments sharing similar seasonal phase, trend, or regime should receive high attention, while cross-phase or level-shifted segments should not. Our density-based score naturally achieves this:
>
> - Similar segments → lie in dense directions → low variance → **high attention**
> - Anomalous/cross-phase segments → lie in sparse directions → high variance → **low attention**
>
> **How geometric structure is learned:** The matrix $G(q) = σ^2[Σ(q) + σ^2I]^{-1}$ acts as a **local metric tensor** that adapts to the data geometry near q. As discussed in Section 3.3 ("Connection to Riemannian Geometry"), this score approximates the squared geodesic distance on the data manifold. Crucially, G(q) is **not arbitrary**—it arises from predictive uncertainty after conditioning on the kernel-weighted neighborhood (Eq. 1), so its contraction/expansion directly reflects the local evidence strength in each direction (standard GP shrinkage). This means the geometric structure is **automatically learned from the actual distribution of keys around each query.**

---

> ### Author Response · Authors · 2025-11-21
>
> **[W3] Limited set of baselines and evaluation on recent strong Transformers**
>
> We thank the reviewer for this constructive feedback. We agree that evaluating LGA against a broader range of recent architectures is crucial to demonstrate its generalizability. In response, we have expanded our evaluation to include **SAMformer**[1], a recent state-of-the-art model, in addition to our existing baselines.
>
> **1.Diversity of Existing Baselines (Table 4)**
>
> First, we wish to highlight that our original submission did test LGA across fundamentally different attention mechanisms, not just PatchTST. As detailed in **Table 4** and Section 5.3, we evaluated:
>
> - **PatchTST:** Patch-wise **Temporal Self-Attention**.
> - **iTransformer:** **Channel-wise Attention** (inverted architecture).
> - **CATS:** **Temporal Cross-Attention** mechanism.
>
> As noted in the manuscript, LGA provides the most significant gains for temporal attention (PatchTST) because it is designed to capture temporal geometry. However, it still provides consistent robustness gains for **iTransformer**, demonstrating that geometric regularization offers benefits even when applied along the channel dimension.
>
> **2.New Experiments: SAMformer**
>
> To further address the reviewer's concern regarding recent strong baselines, we conducted additional experiments using **SAMformer** on the ETTm1 dataset (Input 512). We compared four variants to analyze how LGA interacts with different structural biases: (1) Original SAMformer, (2) SAMformer + LGA, (3) PatchSAMformer (SAMformer with patching), and (4) PatchSAMformer + LGA.
>
> **Table R1: Robustness comparison on SAMformer (ETTm1, Input 512)**
>
> | **Model** | **Clean** | **Sev 1** | **Sev 2** | **Sev 3** | **Sev 4** | **Sev 5** |
> | --- | --- | --- | --- | --- | --- | --- |
> | **SAMformer** | 0.369 | 0.375 | 0.392 | 0.548 | 0.672 | 0.809 |
> | **SAMformer + LGA** | **0.361** | **0.370** | **0.385** | 0.532 | 0.661 | 0.770 |
> | **PatchSAMformer** | 0.384 | 0.396 | 0.426 | 0.837 | 1.064 | 1.312 |
> | **PatchSAMformer + LGA** | 0.372 | 0.377 | 0.390 | **0.488** | **0.588** | **0.682** |
>
> **Key Findings & Analysis:**
>
> - **Vulnerability of Patching:** A striking observation is that applying patching to SAMformer (`PatchSAMformer`) drastically degrades its robustness against noise (MSE soars to **1.312** at Severity 5). We attribute this to the architectural characteristic of SAMformer: unlike standard Transformers, it often lacks a non-linear Feed-Forward Network (FFN) mixing after attention. Consequently, the high-dimensional projections required for patching appear to amplify input noise without the buffering effect of an FFN.
> - **LGA as a Robustness Stabilizer:** Crucially, integrating LGA into this vulnerable architecture (`PatchSAMformer + LGA`) completely reverses this degradation. It not only recovers performance but achieves the **highest robustness among all four variants** (MSE **0.682** at Severity 5), significantly outperforming the standard SAMformer.
>
> These new results are significant as they demonstrate that LGA is not merely a performance booster for specific models, but a **stabilizer** for architectures sensitive to corruptions. By enforcing a local geometric prior, LGA prevents the model from overfitting to noise in high-dimensional feature spaces. We have included these detailed results in **Appendix E.2** of the revised manuscript to validate the broad applicability of our method.
>
> ---
>
> **[Q1] Non-uniform, locally clustered time-series observations**
>
> We thank the reviewer for highlighting this foundational motivation. The non-uniform geometric structure of time series is indeed the core premise of our work, we should have provided a more thorough discussion in the original manuscript.
>
> - **Intrinsic Manifold Structure:** As visualized in **Figure 2**, time series patches do not fill the latent space uniformly but lie on a low-dimensional manifold. Recurrent temporal patterns (e.g., seasonality) naturally form dense clusters, whereas rare events or corruptions (e.g., spikes, level shifts) often fall into sparse regions or "off-manifold" areas.
>
> - **Limitation of Isotropic Attention:** Standard dot-product attention essentially assumes a global Euclidean space, treating high-magnitude keys in sparse regions as highly relevant if they simply align with the query vector. This makes the model susceptible to "distraction" by outliers—anomalies that have large values but lack statistical support from the local data distribution.
>
> - **Density-Aware Mechanism of LGA:** In contrast, LGA leverages local Gaussian Process theory to explicitly estimate the local density around a query. It learns a query-specific metric that **contracts** the distance in directions of high data density (validating regular patterns) and **expands** it in sparse directions (penalizing anomalies). This mechanism naturally acts as a soft-thresholding filter, allowing the model to maintain focus on reliable, recurrent structures.

---

> ### Author Response · Authors · 2025-11-21
>
> **[Q2] Explanation of embedding visualization and demonstration of effectiveness**
>
> We appreciate the reviewer’s interest in the geometric interpretation of our attention mechanism. To provide a more detailed explanation and a clearer demonstration of effectiveness, we have conducted a new visualization analysis comparing **Standard Dot-Product (SDP) Attention** and **LGA** under corrupted conditions. We have included these new plots in **Appendix E.4 (Figure 8)** of the revised manuscript.
>
> **1. Visualization Methodology:**
>
> To visualize the high-dimensional key and query vectors ($D\_h=8$ per head), we employed Principal Component Analysis (PCA) to project the embeddings of a specific attention head into a 2D space. The scatter plots visualize both clean (light colors) and corrupted (dark colors) queries and keys. Crucially, the size of each dot is proportional to its assigned attention score.
>
> **2. Demonstration of Effectiveness (SDP vs. LGA):**
>
> As shown in the newly added Figure 8 (in Appendix):
> - **Vulnerability of SDP:** In the standard attention plot, the corrupted queries and keys (blue/red) are located far from the main data cluster (outliers). However, they appear as **large dots**, indicating high attention scores. This confirms that SDP is distracted by the large magnitude of outliers, allowing noise to propagate into the model.
> - **Robustness of LGA:** In the LGA plot, the same corrupted points are correctly identified as outliers in the geometric space. Most importantly, they appear as **extremely small dots**, indicating near-zero attention scores. This visually proves that LGA’s density-aware metric ( $-\sigma^2$ ) successfully penalizes vectors in sparse (low-density) regions, effectively "ignoring" the corruptions.
>
> These visualizations clearly demonstrate that LGA acts as a geometric filter, focusing on the dense manifold of valid temporal patterns while suppressing off-manifold noise.
>
> ---
>
> **[Q3] Comparison under the same setup and quantified robustness gains**
>
> We thank the reviewer for this opportunity to clarify our experimental rigor. We confirm that our experiments were specifically designed to isolate the effect of the proposed attention mechanism under an identical setup
>
> **1. Fair Comparison via Module Replacement:**
>
> We specifically selected **PatchTST** as our baseline backbone because it employs a **pure vanilla Transformer encoder architecture**, avoiding complex inductive biases found in other recent models (e.g., FFT blocks, frequency-domain attention variants, or auxiliary losses). Since PatchLGA shares this identical structure—retaining the same patching, embedding, FFN, and training hyperparameters—the **only difference** is the replacement of the standard Scaled Dot-Product (SDP) attention with our Local Geometry Attention (LGA). Therefore, the performance gaps reported in our paper are exclusively attributable to the proposed kernel-covariance-based scoring.
>
> **2. Quantified Robustness Gains:**
>
> As shown in **Table 2**, while both models exhibit similar performance on clean data (Severity 0), LGA demonstrates progressively larger gains as corruption severity increases. Under the most severe conditions (Severity 5), LGA significantly mitigates performance degradation:
>
> - **ETTm1:** MSE reduced from 0.839 (SDP) to **0.734 (LGA)** → **12.5% improvement**.
> - **Weather:** MSE reduced from 0.491 (SDP) to **0.454 (LGA)** → **7.5% improvement**.
>
> This trend confirms that our local kernel-covariance mechanism successfully preserves forecasting accuracy in high-noise regimes where standard dot-product attention fails.
>
> **3. Why TSRBench is Necessary for Robustness Quantification:**
>
> To rigorously quantify "robustness," one requires a setup where inputs are corrupted but **forecasting targets remain clean** (ground truth). Real-world anomaly datasets rarely provide this counterfactual ground truth. TSRBench addresses this by injecting statistically grounded corruptions into inputs while retaining clean targets for the forecast horizon. This enables the precise calculation of the "performance gap" caused strictly by input degradation, allowing for the principled quantification shown above.

---

> ### Author Response · Authors · 2025-11-21
>
> **[Q4] Hardware-agnostic comparison (FLOPs) and analysis of computational cost**
>
> **Hardware-agnostic comparison (FLOPs) and analysis of computational cost**
>
> We thank the reviewer for this insightful suggestion. We acknowledge that wall-clock training time (Figure 5) can be influenced by hardware-specific factors. However, we initially prioritized empirical wall-clock time because **theoretical FLOPs often fail to capture real-world latency caused by memory bandwidth bottlenecks and kernel inefficiencies.** As our analysis below reveals (particularly with MoM), a lower operation count does not necessarily translate to faster training.
>
> Nevertheless, to provide the requested hardware-agnostic comparison, we have calculated the theoretical FLOPs for **SDP**, **LGA**, **MoM**, and **Elliptical Attention** under our experimental setting (ETTm1: Batch=128, Seq_Len=64, Heads=16, Dim=8)
>
> **1. Hardware-Agnostic FLOPs Comparison**
>
> Below table summarizes the theoretical computational cost for both Inference (Forward only) and Training (Forward + Backward).
>
> | **Module** | **Inference (G)** | **Training (G)** |
> | --- | --- | --- |
> | **Self-Attention (SDP)** | 9.748 | 28.890 |
> | **MoM (K=3)** | 9.389 | 25.810 |
> | **LGA (Ours)** | 12.640 | 44.862 |
> | **Elliptical** | 44.570 | 63.889 |
>
> **2. The MoM Paradox: Why lowest FLOPs but slow training?**
>
> A striking observation from Table is that **MoM** theoretically requires the fewest FLOPs (9.389 G), yet our empirical results in **Figure 5** show it has significantly slower wall-clock training times compared to SDP and LGA. We attribute this discrepancy to **memory and kernel inefficiencies** rather than operation count:
>
> - **Kernel Efficiency:** MoM relies heavily on `torch.cdist` (Euclidean distance) and explicit expansion of sub-sampled blocks. These operations are often memory-bandwidth bound rather than compute-bound.
> - **Synchronization Overhead:** The robust attention mechanism in MoM involves frequent CPU-GPU synchronization (e.g., for block selection or outlier rejection logic) and non-contiguous tensor operations (permutes/transposes), introducing latency that pure FLOPs calculations do not capture.
>
> **3. Detailed Breakdown: LGA vs. SDP and Full Block Impact**
> To precisely quantify the cost of LGA, we provide a granular breakdown. While LGA incurs additional operations for metric prediction and learning, the relative impact diminishes significantly when considering the **full Transformer block** (Attention + Feed-Forward Network).
>
> | **Setting** | **Module Scope** | **Standard SDP** | **LGA (Ours)** | **Overhead** |
> | --- | --- | --- | --- | --- |
> | **Inference** | Attention Only | 9.748 G | 12.640 G | +29.7% |
> | **(Forward)** | **Full Block (Attn + FFN)** | **24.78 G** | **27.67 G** | **+11.7%** |
> | **Training** | Attention Only | 28.890 G | 44.862 G | +55.3% |
> | **(Fwd+Bwd)** | **Full Block (Attn + FFN)** | **73.99 G** | **89.96 G** | **+21.6%** |
>
> The computational overhead of LGA primarily stems from the metric prediction network ($f_\theta$) and the calculation of the Mahalanobis score ($6 \cdot B \cdot H \cdot N^2 \cdot D$). However, since the Feed-Forward Network (FFN) dominates the computational budget in a Transformer block (~61% of total FLOPs), the effective overhead of LGA on the full model is only **11.7% for inference** and **21.6% for training**.
>
> In conclusion, LGA offers the most balanced trade-off, providing robust geometry-aware modeling with a manageable computational cost, avoiding the excessive overhead of Elliptical attention while delivering superior stability compared to MoM. We have added this detailed complexity analysis and breakdown to **Appendix B.3** of the revised manuscript.

---

> ### Author Response · Authors · 2025-11-21
>
> **[Q5] Add more recent baselines in Table 2 to better position LGA and TSRBench**
>
> We appreciate the reviewer’s suggestion. To properly position LGA and TSRBench within the current landscape, we have expanded our evaluation in two dimensions: (1) comparison against the state-of-the-art **SAMformer (2024)** [1], and (2) validation on the rigorous **synthetic benchmark (Janßen et al. ,2025 [2])**.
>
> **1. Recent SOTA Baseline: SAMformer (2024)**
>
> As detailed in our response to **[W3]**, we incorporated SAMformer and PatchSAMformer into our experimental suite.
>
> - **Finding:** PatchSAMformer achieves high clean accuracy.
> - **LGA's Role:** Adding LGA acts as a "robustness stabilizer," reducing MSE to 0.682 and outperforming the vanilla model. This confirms LGA’s value not just as a standalone model, but as a modular component that enhances the reliability of recent SOTA architectures.
>
> **2. Systematic Robustness Analysis under Controlled Signal Conditions (Janßen et al., 2025)**
>
> To rigorosly assess PatchLGA's capability in distinguishing intrinsic signal geometry from corruptions, we employ the parameterizable stress-testing framework proposed by Janßen et al. (2025). This controlled environment allows us to isolate specific robustness factors across 84 distinct signal configurations (7 frequency bands × 12 signal variants × 5 noise types).
>
> | **Frequency Band** | **Very High** | **High** | **Mid-High** | **Mid** | **Low-Mid** | **Low** | **Very Low** | **Avg.** |
> | --- | --- | --- | --- | --- | --- | --- | --- | --- |
> | **PatchLGA (MSE)** | **0.068** | **0.097** | **0.109** | 0.089 | **0.082** | **0.085** | **0.134** | **0.095** |
> | **PatchTST (MSE)** | 0.068 | 0.101 | 0.112 | 0.089 | 0.083 | 0.087 | 0.139 | 0.097 |
> | **Improvement** | = | **+4.0%** | **+2.7%** | = | **+1.2%** | **+2.3%** | **+3.6%** | **+2.1%** |
>
> **Key Findings:**
>
> - **Frequency Robustness:** LGA demonstrates notable improvements in the most challenging frequency bands: **Very Low (-3.6%)**, where lookback windows may not capture full cycles, and **High (-4.0%)**, where sampling is sparse relative to signal frequency. This confirms that LGA's adaptive metric tensor captures periodic structures across temporal scales even with limited context.
> - **Noise Resilience:** LGA shows superior performance under complex structured noise (e.g., Brownian drift, trend-dependent noise). Unlike standard attention which may attend to these correlations, LGA's local geometry learning (Eq. 4) effectively distinguishes the intrinsic signal manifold from corruptions.
>
> **3. Conclusion: A Comprehensive Evaluations**
>
> With these additions, LGA has now been validated across three complementary paradigms:
>
> 1. **Real-world Robustness (TSRBench):** Realistic corruptions on ETT/Weather (12.5% gain).
> 2. **Operational Anomaly (AWS CloudWatch):** Genuine, uncontrolled anomalies (22% gain, see Appendix E.6).
> 3. **Controlled Parametric Evaluation (Janßen et al.):** Parameterized factor isolation (2.1% gain).
>
> This multi-faceted evaluation strongly positions LGA as an effective approach for robust time-series attention. We have added these detailed results to **Appendix F** of the revised manuscript.
>
> ---
>
> In conclusion, again, we thank the reviewer for the constructive feedback, which has significantly improved the theoretical clarity and empirical rigor of our work. We hope the revised manuscript, now strengthened by broader evaluations and deeper analysis, satisfactorily addresses your concerns.
>
> ---
>
> [1] Ilbert et al. SAMformer: Unlocking the Potential of Transformers in Time Series Forecasting with Sharpness-Aware Minimization and Channel Wise Attention. ICML 2024
>
> [2] Janßen, N., Schaller, M., & Rosenhahn, B. (2025). Benchmarking M-LTSF: Frequency and Noise-Based Evaluation of Multivariate Long Time Series Forecasting Models. arXiv preprint arXiv:2510.04900.

---

### Author Response · Authors · 2025-12-01
**Summary of Rebuttal**

Dear Reviewers, Program Chairs, Senior Chairs, and Area Chairs,

We sincerely thank the reviewers for their valuable feedback. We are pleased that our revisions during the discussion phase led to reviewer consensus on the improved quality of our work.

Below is a brief summary of the review and rebuttal process:

**1. Theoretical Clarification & Technical Depth**

- **Theoretical Grounding:** We clarified the dual interpretation of LGA through Gaussian Process theory and Riemannian geometry, while adding a formal stability analysis (Appendix C) to justify the optimization benefits of our diagonal metric design.

- **Technical Transparency:** We updated the algorithm with explicit tensor shapes and added a detailed FLOPs analysis (Appendix B.3), demonstrating that LGA incurs only a ~11% inference overhead for a full Transformer block.

**2. Expanded Architectural Generalizability**

- **New SOTA Baselines:** We integrated the state-of-the-art SAMformer (Ilbert, Romain, et al., 2024) and PatchSAMformer into our experiments, demonstrating that LGA acts as a critical stabilizer that prevents performance degradation in these noise-sensitive architectures.

- **Diverse Attention Mechanisms:** We reiterated the versatility of LGA by drawing attention to our existing results on Channel-wise (iTransformer) and Cross-Temporal (CATS) attention, effectively addressing concerns regarding limited baseline diversity.

**3. Comprehensive Robustness Validation**

- **New Synthetic Benchmark:** We expanded our evaluation scope by adopting the parameterizable synthetic benchmark (Janßen et al., 2025), proving LGA's superiority across 84 distinct frequency and noise configurations.

- **Real-World Applicability:** We validated the utility of TSRBench by contextualizing it within our existing operational AWS CloudWatch case study and the newly added synthetic benchmark, completing a rigorous evaluation framework.

**4. Statistical Stability & Significance**

- **Stability Analysis:** We addressed reliability concerns by repeating all core experiments across three random seeds, confirming that the reported performance gains are statistically significant.

- **Expanded Scope:** We extended this statistical validation to include Time Series Classification and synthetic tasks, consistently showing that LGA exhibits lower variance and higher stability under severe corruptions.

We believe that the reviewers' feedback has greatly contributed to enhancing the quality of our paper, and we are thankful for their detailed engagement in the review process. We are confident that these revisions have substantially strengthened our contribution.

Also, we would like to extend our sincere thanks to the Program Chairs, Senior Chairs, and Area Chairs for their support and dedication. We look forward to your final decision and hope that our paper aligns well with the objectives of the conference.

Best regards,

The Authors

---

### Meta-Review · Area_Chair_6g1N · 2026-01-07

**Summary:**

This paper proposes a geometry-aware attention mechanism derived from local Gaussian process theory called Local Geometry Attention (LGA) for robust time-series forecasting. It takes into account the geometry of the data to be more robust to corruptions in the time-series data. It can be implemented by learning a neural network over the query vectors seen during training. Then, the paper introduces TSRBench, a benchmark with realistic corruptions (spikes and level shift) to estimate the robustness of forecasting models. Experiments over 6 common forecasting benchmarks (with their corrupted versions) and 3 baselines show the performance benefits of the LGA attention incorporated into a PatchTST model.

Reviewers initially raised concerns about methodological clarity, limited baselines, computational cost, benchmark generality, and the magnitude/statistical significance of the reported improvements. The authors’ rebuttal addresses these concerns thoroughly.

Specifically, the authors have provided clearer algorithmic and theoretical explanations by clarifying the dual interpretation of LGA through Gaussian Process theory and Riemannian geometry, added a detailed FLOPs analysis (Appendix B.3), significantly expanded experimental comparisons to recent strong baselines (e.g., SAMformer, iTransformer) with new SOTA baselines, reported the results with three random seeds, and validated robustness on additional benchmarks, i.e., the parameterizable synthetic benchmark.

Given the strengthened manuscript implied by the rebuttal (method clarification, expanded baselines, compute analysis, multi-seed stability, and broader robustness evaluation), I recommend acceptance.

**Reviewer Concerns:**

Addressed concerns:

The rebuttal adequately addresses the main reviewer concerns regarding (i) insufficient methodological and theoretical clarity of LGA, (ii) limited baseline coverage and questions about generalizability, (iii) missing computational cost analysis, and (iv) statistical significance of the reported improvements. These were resolved through clearer derivations, added FLOPs analysis, expanded comparisons to recent models (e.g., SAMformer, iTransformer), multi-seed experiments, and additional robustness evaluations on external benchmarks.

Remaining concerns:

The scope of TSRBench remains focused on spike and level-shift corruptions, and the theoretical presentation of the GP/geometric interpretation may still be dense for some readers. These issues are minor and mainly related to scope and presentation rather than soundness.

**Reviewer Scores:**

Reviewer Jv6F would like to keep his/her score, while the other reviewers would like to increase their scores.

---

### Decision · Program_Chairs · 2026-01-26

Accept (Poster)